# Trees to Flows and Back: Unifying Decision Trees and Diffusion Models

**Sai Niranjan Ramachandran** [1 2]   **Suvrit Sra** [1 2]

## Abstract

Decision trees and diffusion models are ostensibly disparate model classes, one discrete and hierarchical, the other continuous and dynamic. This work unifies the two by establishing a crisp mathematical correspondence between hierarchical decision trees and diffusion processes in appropriate limiting regimes. Our unification reveals a shared optimization principle: *Global Trajectory Score Matching (GTSM)*, for which gradient boosting (in an idealized version) is asymptotically optimal. We underscore the conceptual value of our work through two key practical instantiations: TREEFLOW, which achieves competitive generation quality on tabular data with higher fidelity and a 2× computational speedup, and DSM-TREE, a novel distillation method that transfers hierarchical decision logic into neural networks, matching teacher performance within 2% on many benchmarks.

## 1. Introduction

Within machine learning, two classes of models have achieved great success, albeit in disjoint domains. For structured, tabular data, ensemble methods based on decision trees, particularly Gradient Boosting Machines (Friedman, 2001; 2002), remain the state-of-the-art, prized for their performance and interpretability. For continuous, complex data such as images and audio, score-based generative models, or Diffusion Models (DMs) (Ho et al., 2020; Song et al., 2021), are dominant, capable of generating samples of unparalleled fidelity. These two model classes live in different worlds: one relies on discrete, hierarchical partitioning of the feature space, while the other is defined by continuous-time stochastic differential equations (SDEs).

[1]School of Computation, Information and Technology, Technical University of Munich, Germany [2]Munich Center for Machine Learning (MCML). Correspondence to: Sai Niranjan Ramachandran <sainiranjan.ramachandran@tum.de>, Suvrit Sra <s.sra@tum.de>.

*Proceedings of the 43rd International Conference on Machine Learning*, Seoul, South Korea. PMLR 306, 2026. Copyright 2026 by the author(s).

But this conceptual separation obscures a deep and powerful connection. Though their formulations differ, both models implicitly perform a hierarchical refinement of information: decision trees build a coarse-to-fine sequence of partitions, while diffusion models learn to reverse a coarse-graining process that gradually destroys information.

We bridge this conceptual divide by establishing a formal mathematical correspondence between hierarchical tree models and the deterministic flows associated with diffusion processes under suitable limiting procedures (see Figure 1). Thus, we demonstrate that rather than being disjoint model classes, they are simply two perspectives of the same underlying generative and discriminative object.

More precisely, we first show that any decision tree's hierarchical coarse-graining of data defines a discrete-time Markov process whose continuous limit is a unique deterministic flow that is described by a Probability Flow Ordinary Differential Equation (Song et al., 2021) (PF-ODE) (Tree → Flow). Conversely, we prove that any suitable diffusion process whilst evolving from data to noise (forward process), induces a canonical hierarchical clustering. The forward process progressively merges the modes of the data distribution and the temporal ordering of these modes define a hierarchy isomorphic to a decision tree (Flow → Tree).

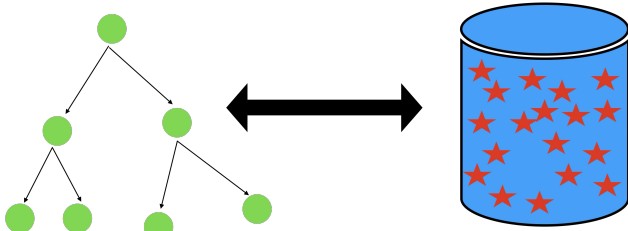

*Figure 1.* **A Visual Metaphor for Trees to Flows and back.** We connect the discrete structure of a decision tree (left) with the continuous dynamics of a flow process (right). We establish a a formal correspondence, enabling an analysis of boosting via flows, and imbuing generative models with tree-based inductive biases.

This Tree↔Flow correspondence reveals a common optimization principle, which we term Global Trajectory Score Matching (GTSM) (§3). We show that both the greedy, stage-wise construction of boosting and the end-to-end training of score-based models can be understood as distinct but related solvers for the GSTM master objective.

A valuable consequence of our unified perspective is how it enables novel algorithms that combine the strengths of both frameworks. Leveraging the explicit, adaptive partitioning as a structural prior, we develop two algorithms that excel on tabular data. The first achieves competitive generation quality where standard diffusion models typically struggle. The second successfully distills complete tree hierarchies into neural networks, to our knowledge for the first time.

**Summary of Contributions.**

- We establish a formal mathematical correspondence between hierarchical decision trees and the deterministic flows (PF-ODEs) associated with diffusion processes, valid under limiting refinement procedures.

- We introduce the Global Trajectory Score Matching (GTSM) framework, a unifying objective, under which we show how both gradient boosting and score-based diffusion training are optimal solvers in their respective discrete and continuous domains.

- We introduce **TREEFLOW**, a new method that conditions continuous flows using tree-based priors. TREEFLOW achieves **competitive generation quality on tabular data** (highest TSTR accuracy on 3/5 benchmarks, lowest Wasserstein distance on 4/5 benchmarks, and lowest correlation error on 3/5 benchmarks) while being 2X faster.

- We propose **DSM-TREE**, a novel method **to distill complete hierarchical decision logic** (not just leaf predictions) into neural networks, matching teacher performance within 2% on most benchmarks and exceeding it by 3.7% on the Heart Disease dataset.

In Section 2, we establish the Tree-Flow correspondence. Section 3 introduces the unifying GTSM framework and proves the asymptotic optimality of gradient boosting as a discrete solver. We then present our novel algorithms in Section 4 and validate our framework with extensive experiments in Section 5. For a visual guide to our theory and derivations, we refer the reader to the roadmap in Section B.

## 2. The Tree–Flow Correspondence

We establish below a mathematical correspondence between decision trees and diffusion models. We first demonstrate that a decision tree defines a continuous-time PF-ODE via hierarchical coarse-graining under limiting assumptions (§ 2.1). Then, we prove that an entropically homogeneous SDE possessing a stationary distribution induces a canonical tree structure via moment-based clustering (§ 2.2).

**The Tree as a Discrete-Time Markov Process.**

**Definition 2.1** (Decision Tree). A decision tree $\mathcal{T}$ of depth $D \in \mathbb{N}$ on $\mathcal{X} \subset \mathbb{R}^d$ is a nested sequence of partitions

$\Pi_D \prec \cdots \prec \Pi_0$ ($\prec$ denotes refinement) with $\Pi_0 = \{\mathcal{X}\}$ such that the elements in any partition $\Pi_k$, $k > 0$ form a disjoint cover of $\mathcal{X}$, and

$$\forall k \geq 1, \ \Omega \in \Pi_k \implies \exists! \ \Omega' \in \Pi_{k-1} \ \text{s.t.} \ \Omega \subset \Omega'.$$

This induces a filtration $\mathcal{F}_k = \sigma(\Pi_k)$ and conditional densities $p(x, k) := \mathbb{E}[p_0(x) \mid \mathcal{F}_k]$ piecewise constant on $\Pi_k$.

By the tower property (Durrett & Durrett, 2019), the transition operator $\mathcal{M}_k : L^1(\mathcal{X}) \to L^1(\mathcal{X})$ is

$$p(x, k+1) = \mathcal{M}_k p(x, k) = \mathbb{E}[p(x, k) \mid \mathcal{F}_{k+1}]. \quad (1)$$

Thus, this sequence forms a discrete-time Markov chain tracing a path of monotonically decreasing entropy from the root $p(x, 0) = \text{Unif}(\mathcal{X})$ to the leaves $p(x, D) \approx p_{\text{data}}(x)$.

**The Continuous-Time Limit.** For typical trees, inhomogeneity ($\mathcal{M}_k \neq \mathcal{M}_{k+1}$) precludes direct limits. We resolve this by introducing a conceptual refinement procedure.

**Definition 2.2** (Dyadic Refinement). The $n$-th refinement $\mathcal{T}^{(n)}$ is a tree of depth $2^n T$ constructed by recursively inserting intermediate partitions $\Pi_t$ such that for all $k \in \{0, \ldots, 2^{n-1}T - 1\}$, the new partition satisfies $\Pi_{\frac{k}{2^{n-1}}} \prec \Pi_{\frac{2k+1}{2^n}} \prec \Pi_{\frac{k+1}{2^{n-1}}}$. As $n \to \infty$, this sequence converges to a continuous-time filtration $\{\mathcal{F}_t\}_{t \in [0,T]}$ where $\mathcal{F}_t = \sigma(\Pi_t)$ and the density path $p(x, t) = \mathbb{E}[p_0 \mid \mathcal{F}_t]$ is $C^1$ in $t$.

This procedure constructs a smooth interpolating path between the original tree's partitions, transforming the discrete steps of coarse-graining into an infinitesimally smooth "ramp." To this end, we recursively insert intermediate partitions. Consider a simple example: a parent node's region is defined by the split feature A < 10 and its child's region is feature A < 5. A valid intermediate partition always exists by continuity (e.g., say by splitting at feature A < 7.5). This transforms one large step into two smaller ones. By repeating this bisection process recursively (hence, "dyadic"), we can generate a sequence of hierarchies with exponentially more, infinitesimally small steps. In the limit as the number of refinements $n \to \infty$, this discrete sequence of operators converges to a well-defined continuous-time propagator, as noted below.

**Theorem 2.3** (Limit under Dyadic Refinement (Informal)). *Let $\{\mathcal{T}^{(n)}\}$ be a dyadically refined sequence of decision trees as in Definition 2.2. Under the following assumptions*

1. *Scale consistency: inserting intermediate splits between existing tree levels does not change the conditional densities at the original levels.*

2. *Local refinement: newly introduced splits become uniformly finer with refinement, so that the geometric size of each refinement step vanishes in the limit,*

*the associated density path $p(x, t) = \mathbb{E}[p_0(x) \mid \mathcal{F}_t]$ admits a well-defined continuous-time description generated by a (possibly time-dependent) operator $\mathcal{G}_t$ that is locally Lipschitz on compact intervals. Moreover, for any compact interval $I \subset [0, T]$, there is a subsequence of refinements along which $\mathcal{G}_t$ converges to a time-invariant generator $\mathcal{G}$.*

*Proof.* At each refinement level, the conditional densities change only slightly due to the vanishing size of new splits (local refinement), giving a Lipschitz continuous family of generators. By a standard diagonalization argument on compact intervals, one can extract a subsequence along which these generators converge. Time-invariance emerges in the limit because the contribution of each refinement step becomes negligible, so the limiting generator $\mathcal{G}$ effectively governs the density path across the entire interval. The full proof with all the details is provided in **Theorem** C.13. □

**From Discrete Steps to a Differential Equations** The limit of the refined process from **Theorem** 2.3 is, by construction, a continuous-path Markov process. The time evolution of its density $p(\mathbf{x}, t)$ is described by the Kramer–Moyal expansion (Gardiner, 2004):

$$\frac{\partial p}{\partial t} = \sum_{n=1}^{\infty} \frac{(-1)^n}{n!} \frac{\partial^n}{\partial \mathbf{x}^n} \left[ D^{(n)}(\mathbf{x}) p(\mathbf{x}, t) \right],$$

where $D^{(n)}$ are the moments of the process's infinitesimal jumps. First, we note an observation about higher moments.

**Theorem 2.4** (Higher-Order Moments Vanish (Informal)). *For the limiting process constructed via dyadic refinement, the contributions of jumps of order three and higher become negligible quickly as the refinement becomes finer.*

*Proof.* Intuitively, the claim holds because each refinement redistributes probability only within ever smaller regions, so large or high-order jumps are geometrically suppressed. Indeed, as the dyadic partitions get finer, the size of the regions shrinks exponentially, which forces all moments of order three and above to go to zero. **Proposition** C.20 in the appendix provides the formal argument. □

Next, note that Pawula's theorem (Pawula, 1967) states that if any jump moment $D^{(n>2)}$ is zero, all higher moments must also be zero. But as the jumps become negligible for our case, this implies $D^{(n)}(\mathbf{x}) = 0$ for all $n > 2$. Hence, the expansion truncates exactly to the Fokker-Planck equation:

$$\frac{\partial p}{\partial t} = -\frac{\partial}{\partial \mathbf{x}}[D^{(1)}(\mathbf{x})p] + \frac{1}{2}\frac{\partial^2}{\partial \mathbf{x}^2}[D^{(2)}(\mathbf{x})p].$$

Further, the underlying coarse-graining operator is purely deterministic (an averaging process), i.e., it introduces no new stochasticity. Therefore, the second-order (diffusion)

term must vanish: $D^{(2)}(\mathbf{x}) \equiv \mathbf{0}$. This reduces the Fokker-Planck equation to a first-order Liouville equation, whose characteristic equation is a deterministic ODE.

**Theorem 2.5** (Trees Induce PF-ODEs (Informal)). *Under dyadic refinement, the continuous-time limit of tree-induced coarse-graining is $\dot{\mathbf{x}} = \mathbf{v}(\mathbf{x}, t)$, a deterministic PF-ODE with velocity $\mathbf{v}$ uniquely determined by the partitions.*

*Proof.* Please see Section C (**Theorem** C.21). □

### 2.1. From Flows to Trees

We now establish the reverse mapping: every entropically homogeneous SDE having a stationary distribution induces a tree via its corresponding PF-ODE. Intuitively, as the process evolves from a structured data distribution towards a high-entropy stationary state, distinct local modes of the initial density merge. The hierarchy records this timeline: regions that remain statistically distinguishable for longer are grouped higher. As the flow of entropy is monotonic, this merging is irreversible, ensuring a well-defined tree structure. Formally our argument extends a statistical physics approach (Ramachandran et al., 2025), previously used to analyze the trajectories of generative diffusion models.

**Definition 2.6** (Entropically Homogeneous Stationary SDE). An SDE $d\mathbf{x}_t = \mathbf{b}(\mathbf{x}_t, t)dt + \sigma(\mathbf{x}_t, t)d\mathbf{w}_t$ possesses a stationary distribution if it converges to a unique invariant measure $p_\infty$ such that $\lim_{t \to \infty} \|p_t - p_\infty\|_{\mathrm{TV}} = 0$. We call the process **entropically homogeneous** if the differential entropy $H(p_t)$ is a monotonically non-dec(inc)reasing function of $t$.

**Definition 2.7** (Initial Clusters). For a data density $p_0$ with $K$ isolated modes, the **initial clusters** $\{C_k\}_{k=1}^{K}$ are the connected components of a super-level set of $p_0$ chosen to yield exactly $K$ components (Section D).

**Definition 2.8** (Moment-Based Merger Time). For clusters $C_i, C_j$, the $(n, \epsilon)$-**merger time** is the first time $t$ at which their $n$-th order conditional moment tensors become statistically indistinguishable:

$$t_{ij}^{(n,\epsilon)} = \inf\{t \geq 0 : \|\mathbf{M}_t^{(n)}(C_i) - \mathbf{M}_t^{(n)}(C_j)\| \leq \epsilon\}, \quad (2)$$

where $\mathbf{M}_t^{(n)}(C_k) = \mathbb{E}_{\mathbf{x} \sim p_t(\cdot|C_k)}[(\mathbf{x} - \mathbb{E}[\mathbf{x}])^{\otimes n}]$.

These merger times satisfy the following property:

**Theorem 2.9** (Mergers Form a Hierarchy (Informal)). *For an entropically homogeneous SDE, the moment-based merger times $t_{ij}^{(n,\epsilon)}$ of clusters satisfy a hierarchical structure, and these merger times obey an ultrametric inequality.*

*Proof.* Each cluster's conditional moments evolve continuously and monotonically under the SDE; once two clusters merge (their moments become $\epsilon$-close), they stay merged. For any three clusters, consider the two whose merger occurs

last. By continuity and the triangle inequality, the remaining cluster must merge no later than this maximal time, which yields the ultrametric property and the consequent hierarchical structure (**Proposition** D.14 in the appendix). $\square$

Consequently, we can obtain a hierarchical clustering.

**Theorem 2.10** (SDEs Induce Trees (Informal)). *An entropically homogeneous SDE, initialized from a distribution with well-separated modes, induces a unique hierarchical clustering via its moment-based merger times. This structure depends only on the SDE's dynamics and its initialization, hence is fully characterized by the corresponding PF-ODE.*

*Proof.* **Theorem** D.15 in Section D establishes that entropically homogeneous SDEs induce a dendogram by tracking the moment based merger times of data modes of $p_{\text{init}}$. As the construction process relies only on the marginals, the dendogram is fully characterized by the evolution of the corresponding PF-ODE (Song et al., 2021). We further show that stationarity is a necessary and sufficient condition for the tree to be rooted in **Proposition** D.18. $\square$

# 3. Global Trajectory Score Matching

The Tree-Flow equivalence implies that both gradient boosting and diffusion models solve the same underlying problem: learning an optimal trajectory from a simple prior to a complex data distribution. The ultimate goal is to match the ideal path-space measure, a notoriously intractable objective (Lai et al., 2025). Our central insight is that this global problem (under standard regularity conditions) is decomposable, building on a **local-to-global consistency** principle: a trajectory is globally optimal if and only if its dynamics are correct at every infinitesimal step, for all states along the path. Any "chunk" of the trajectory can be thus seen as a sequence of such infinitesimal steps.

We formalize this principle via our Global Trajectory Score Matching (GTSM) framework, which replaces the single, intractable global objective with an infinite sum of local, tractable ones. At any point $(\mathbf{x}, t)$, the score function specifies the optimal direction for the next infinitesimal step of the reverse process. By integrating the local score-matching error over states and times, the objective sums the consistency checks over all infinitesimal chunks of every trajectory. This decomposability is what allows disparate algorithms to solve the same fundamental problem. End-to-end training of a single score network attempts to minimize the entire integral at once. In contrast, gradient boosting acts as a greedy optimizer, iteratively adding weak learners that reduce the largest remaining errors across the full GTSM integral.

## 3.1. The Continuous GTSM Objective

**Definition 3.1** (Continuous Global Trajectory Score Matching). For an ideal SDE with law $\mathbb{P}^*$ and scores $\mathbf{s}_t^*(\mathbf{x})$, and a model $\mathbf{s}_\theta(\mathbf{x}, t)$, the CGTSM objective is:

$$\mathcal{L}_{\text{CGTSM}}(\theta) = \frac{1}{2} \int_0^T w(t) \mathbb{E}_{p_t^*} \left[ \left\| \mathbf{s}_\theta(\mathbf{x}, t) - \mathbf{s}_t^*(\mathbf{x}) \right\|_{\mathbf{D}(t)}^2 \right] dt, \tag{3}$$

where $w(t) > 0$ is a weighting function and $\|\mathbf{v}\|_{\mathbf{D}} = \sqrt{\mathbf{v}^\top \mathbf{D} \mathbf{v}}$ is the diffusion-induced norm.[1]

**Theorem 3.2** (CGTSM Optimality Implies Path Matching). *Achieving zero CGTSM loss for any strictly positive weighting $w(t) > 0$ is necessary and sufficient for matching the full path-space measures, i.e., $\mathbb{P}_\theta = \mathbb{P}^*$.*

*Proof sketch.* By Girsanov's theorem (Oksendal, 2013), the KL divergence between path-space measures, $D_{\text{KL}}(\mathbb{P}^* \| \mathbb{P}_\theta)$, is an integral of the squared difference between the process drifts. Since the reverse-time drift is a function of the score, this difference reduces to the CGTSM integrand. The loss is zero if and only if the KL divergence is zero. The full proof is in Section F (**Corollary** F.3). $\square$

## 3.2. Boosting as a CGTSM Solver

Gradient boosting constructs an ensemble $F_m = \sum_{i=1}^m h_i$ by iteratively adding weak learners (decision trees) that fit the residual error. To map this additive process into our SDE framework, we introduce the *net decision tree* abstraction. At each step $m$, the net tree $T_m$ is a single, monolithic hierarchical model whose partition $\Pi_m$ is the common refinement of the partitions of all constituent learners $\{h_1, \ldots, h_m\}$, and whose leaf values are the sum of their predictions.

This abstraction transforms the boosting algorithm from a sequence of functional additions into a sequence of structural refinements. We show that the sequence of partitions $\{\Pi_i\}_{i=1}^m$ induced by the net trees is monotonic and strictly refining (Section E.1.3). We then apply our Tree-to-Flow mapping (**Theorem** 2.5) to the canonical hierarchy of the net trees $\{T_1, T_2, \ldots T_m\}$, yielding a series of unique processes[2] $\{S_1, S_2, \ldots, S_m\}$, moving from coarse to fine approximations. The mechanism guiding this trajectory is the minimization of a discrete GTSM objective.

**Definition 3.3** (Discrete GTSM for Boosting). The discrete GTSM objective is the sum of score-matching losses:

$$\mathcal{L}_{\text{DGTSM}}(\{h_m\}) = \sum_{i=1}^m \mathbb{E}_{(\mathbf{x},y)} \left[ \|h_i(\mathbf{x}) - \mathbf{r}_i(\mathbf{x})\|^2 \right], \tag{4}$$

where the residual $\mathbf{r}_i(\mathbf{x}) = y - F_i(\mathbf{x})$ is an unbiased estimator of the optimal score update for the underlying process at stage $i$ (Section E, **Theorem** E.22).

---

[1]Technically this is a semi-norm unless $\mathbf{D}$ is strictly positive definite. For the SDE $d\mathbf{X}_t = \mathbf{b}(\mathbf{x}_t, t) \, dt + \sigma(\mathbf{x}_t, t) \, d\mathbf{w}_t$, the diffusion tensor is given by $\mathbf{D} = \sigma\sigma^\top$, which is positive definite once one disallows rank deficient $\sigma$ (Oksendal, 2013).

[2]For stochastic boosting (Friedman, 2002), **Theorem** 2.5 reduces to a SDE as the diffusion term is no longer $\mathbf{0}$.

We rewrite the problem as a finite-horizon sequential decision problem to showcase optimality. Define the state at step $m$ as the current model's induced process ($S_m$), the action as the choice of the next weak learner ($h_{m+1}$), a deterministic state transition ($S_{m+1} = T(S_m, h_{m+1})$), and a stage cost $C(S_m, h_{m+1})$ given by the immediate score-matching error. The total cost to be minimized is the sum of these stage costs, which is precisely the discrete GTSM objective (4). Through this formulation we obtain the following result.

**Theorem 3.4** (Greedy Boosting is Globally Optimal). *In the continuous limit, where the class of weak learners is sufficiently rich to approximate any function, the greedy policy of selecting the weak learner that minimizes the immediate stage cost at each step is the globally optimal policy for the discrete GTSM problem.*

*Proof sketch.* The proof proceeds by backward induction on the optimal cost-to-go, or value function $V_m(S_m)$. The Bellman equation (Bertsekas, 2012) relates the value at stage $m$ to the value at stage $m + 1$. Due to the additive separability of the GTSM objective and the deterministic state transitions, the optimal action at stage $m$ is simply the one that minimizes the immediate stage cost, independent of the future value function. This proves that the greedy choice is optimal at every stage. The full proof is in Section E. □

## 4. Algorithmic Instantiations

To operationalize the theoretical unification of decision trees and diffusion processes, we propose two novel algorithms for handling tabular data. These methods leverage the hierarchical partitions of trees to provide structural inductive biases to neural networks, one for generative modeling and the other for discriminative distillation. These algorithms directly approximate the Global Trajectory Score Matching (GTSM) objective and achieve competitive performance while providing substantial computational advantages.

### 4.1. TREEFLOW: Tree-Conditioned Flow Matching

Recent attempts to apply diffusion models to tabular data, such as TabDDPM (Kotelnikov et al., 2023), achieve strong results but often have high computational costs. **TreeFlow** addresses this limitation by using decision tree partitions as a conditioning mechanism for **Conditional Flow Matching (CFM)** (Lipman et al., 2023), achieving competitive generation while being 2X faster.

**Intuition.** TREEFLOW first "maps" the data manifold using a decision tree. It assigns each data point a unique "path encoding," a vector representing its journey from the root to a leaf. We then train a velocity field $v_\theta$ to move from noise to data, *conditioned* on these paths. This approach partitions the generative task as the model learns special-

ized flows for different regions of data space. At generation time, we can target specific partitions, resulting in samples with state-of-the-art fidelity (lowest Wasserstein distance on 4/5 benchmarks) and significantly faster generation ($2\times$ speedup over TabDDPM). By viewing TREEFLOW through the GTSM lens, we provide a rigorous **distributional convergence proof** (Section H, **Corollary** H.5), establishing that it captures the true conditional data distribution within each partition. Informally, under a refinement procedure that interpolates between tree partitions, the sequence of conditional densities converges to a deterministic flow uniquely defined by the tree geometry.

---

**Algorithm 1** TREEFLOW: Training Phase
---
1: **Input:** Dataset $\mathcal{D}$, tree depth $D$
    **Learn Data Partitioning Structure**
2: Train decision tree $\mathcal{T}$ on $\mathcal{D}$ to depth $D$
3: For each $\mathbf{x}_i$, compute path encoding: $\mathbf{p}_i =$ PathEncoder($\mathcal{T}, \mathbf{x}_i$)
    **Train Conditional Flow Matching Model**
4: Initialize velocity field $v_\theta(\mathbf{x}, t, \mathbf{p}, y)$
5: **for** training steps $s = 1, \ldots, S$ **do**
6:     Sample batch $\{(\mathbf{x}_i^{\text{data}}, \mathbf{p}_i, y_i)\}_{i \in \mathcal{B}}$
7:     Sample noise: $\mathbf{x}_i^{\text{noise}} \sim \mathcal{N}(\mathbf{0}, \mathbf{I})$
8:     Sample time: $t \sim \text{Uniform}(0, 1)$
9:     Interpolate: $\mathbf{x}_i^{(t)} = t\mathbf{x}_i^{\text{data}} + (1-t)\mathbf{x}_i^{\text{noise}}$
10:     Target velocity: $\mathbf{v}_i^* = \mathbf{x}_i^{\text{data}} - \mathbf{x}_i^{\text{noise}}$
11:     Loss: $\mathcal{L} = \sum_{i \in \mathcal{B}} \|v_\theta(\mathbf{x}_i^{(t)}, t, \mathbf{p}_i, y_i) - \mathbf{v}_i^*\|^2$
12:     Update: $\theta \leftarrow \theta - \eta \nabla_\theta \mathcal{L}$
13: **end for**
14: **Output:** Trained velocity field model $v_\theta$ and tree $\mathcal{T}$

---

**Algorithm 2** TREEFLOW: Generation Phase
---
1: **Input:** Trained model $v_\theta$, tree $\mathcal{T}$, target label $y_{\text{target}}$, partition $R_{\text{target}}$
    **Partition-Targeted Generation**
2: Sample reference $\mathbf{x}_{\text{ref}} \sim \{\mathbf{x}_i \in \mathcal{D} : y_i = y_{\text{target}}, \mathbf{x}_i \in R_{\text{target}}\}$
3: Compute conditioning path: $\mathbf{p}_{\text{ref}} = \text{PathEncoder}(\mathcal{T}, \mathbf{x}_{\text{ref}})$
4: Initialize: $\mathbf{x}^{(0)} \sim \mathcal{N}(\mathbf{0}, \mathbf{I})$
5: **for** $t = 0$ to $1$ with step size $\Delta t$ **do**
6:     $\mathbf{x}^{(t+\Delta t)} \leftarrow \mathbf{x}^{(t)} + v_\theta(\mathbf{x}^{(t)}, t, \mathbf{p}_{\text{ref}}, y_{\text{target}}) \cdot \Delta t$
7: **end for**
8: **Output:** Synthetic sample $\tilde{\mathbf{x}} = \mathbf{x}^{(1)}$

---

### 4.2. DSM-TREE: Distilling Trees via Hierarchical Score Matching

Despite the dominance of deep learning in vision and language, tabular data remains a domain where tree-based ensembles like XGBoost (Chen et al., 2015) and Random Forests (Breiman, 2001) often outperform neural networks

(Grinsztajn et al., 2022). Neural networks typically lack the axis-aligned splitting bias and the ability to handle the "jumpy" manifolds characteristic of tabular datasets. **Discretized Score Matching for Trees (DSM-TREE)** bridges this gap by distilling the *entire decision trajectory* of a decision tree into a neural network which to our knowledge is the first method to supervise networks on complete hierarchical logic rather than just leaf predictions.

**Intuition.** Unlike standard knowledge distillation (Hinton et al., 2015) based on output supervision, DSM-TREE treats every internal split in the tree as a directional guide. By training the network to predict the correct decision at *every* level $j$ of the hierarchy, the network learns the tree's coarse-graining flow. This "fluidizes" the discrete logic of the tree into a continuous, differentiable neural function that retains the tree's superior bias. The training process, detailed in Algorithm 3, closely resembles score matching (Song et al., 2021) by design, as it follows from the GTSM framework. As shown in Section G, DSM-TREE is supported by a finite-sample **convergence guarantee** (**Theorem** G.5). Informally, the reasoning proceeds as follows, if the model can match the true decisions at every level then gradient-based training gradually reduces the per-level prediction errors hence as total error reduces asymptotically the model reproduces all paths exactly. The inference process is described in Algorithm 4.

---

**Algorithm 3** DSM-TREE: Training Phase

---

1: **Input:** Dataset $\mathcal{D}$, tree depth $D$
   **Base Tree Generation (Teacher Model)**
2: Train oracle $\mathcal{O}$ (e.g., Random Forest) on $\mathcal{D}$
3: Generate pseudo-labels: $\tilde{y}_i = \mathcal{O}(\mathbf{x}_i)$
4: Train base decision tree $\mathcal{T}$ on $\{(\mathbf{x}_i, \tilde{y}_i)\}$ to depth $D$
   **Conditional Score Model Training (Student Model)**
5: Initialize network $M_\theta(\mathbf{x}, j) : \mathbb{R}^d \times \{0..D-1\} \to \Delta^1$
6: **for** training steps $t = 1, \ldots, T$ **do**
7:     Sample mini-batch $\{(\mathbf{x}_i, y_i)\}_{i \in \mathcal{B}}$
8:     Sample levels $\{j_i\}_{i \in \mathcal{B}} \sim \text{Uniform}(0, D-1)$
9:     For each $(\mathbf{x}_i, j_i)$: extract $d_i^* = \text{TreeDecision}(\mathcal{T}, \mathbf{x}_i, j_i)$
10:     Compute loss: $\mathcal{L} = \sum_{i \in \mathcal{B}} \mathbb{I}[d_i^* \neq \perp] \cdot \ell_{\text{CE}}(M_\theta(\mathbf{x}_i, j_i), d_i^*)$
11:     Update parameters: $\theta \leftarrow \theta - \eta \nabla_\theta \mathcal{L}$
12: **end for**
13: **Output:** Trained model $M_\theta$ and teacher tree $\mathcal{T}$

---

# 5. Experiments

We conduct experiments with two goals: (1) provide empirical evidence for the Tree-Flow correspondence by showing that diffusion models learn implicit hierarchical structure and show information decay analogous to decision trees, and (2) demonstrate the practical utility of our Global Trajectory

---

**Algorithm 4** DSM-TREE: Inference Phase

---

1: **Input:** Trained model $M_\theta$, teacher tree $\mathcal{T}$, test sample $\mathbf{x}$
   **Neural Network Inference (Mimicking Tree Traversal)**
2: Initialize: node $\leftarrow$ root$(\mathcal{T})$
3: **for** level $j = 0, \ldots, D-1$ **do**
4:     **if** node is a leaf **then break**
5:     **end if**
6:     Predict decision: $\hat{d}_j = \arg\max M_\theta(\mathbf{x}, j)$
7:     Navigate to next node: node $\leftarrow \text{child}_{\hat{d}_j}(\text{node})$
8: **end for**
9: Get prediction from final node: $\hat{y} = \text{Value}(\text{node})$
10: **Output:** Final prediction $\hat{y}$

---

Score Matching (GTSM) framework via the TREEFLOW algorithm for generation and the DSM-TREE algorithm for discriminative distillation. Additional details, architectures, and full results appear in Section I.

## 5.1. Verifying the Tree-Flow Equivalence

**Implicit Tree Structure in Diffusion Models.** To validate our claim from §2.2, we test whether trained diffusion models learn canonical trees. We train a simple MLP on synthetic 2D datasets from `scikit-learn` (Pedregosa et al., 2011) and discover the learned hierarchy via time-domain agglomerative clustering. For each initial cluster, we simulate forward evolution using an SDE with drift given by the model's score function, tracking centroids and statistical spread. We identify the earliest time $t$ when cluster pairs become indistinguishable (inter-centroid distance below combined spread), these merger events construct a dendrogram where the vertical axis represents time (Figure 2(b)). Figure 2 shows original clusters (a), resulting hierarchy (b), and an intermediate PF-ODE state (c), providing empirical evidence that learned dynamics encode discrete hierarchical structure. See Section I.1.1 for further results and additional details.

**Information-Theoretic Analogy.** We next compare information decay in decision trees and diffusion models. For a `DecisionTreeClassifier` on MNIST (LeCun et al., 1998), we measure weighted-average class entropy at each depth. For forward diffusion, we use the information-theoretic proxy $\left(1/(1 + \text{SNR})\right)$. Figure 3 (top) shows both exhibit similar sigmoidal entropy increase. Visual prototypes (bottom) illustrate this, tree prototypes (pixel-wise node averages) blur progressively toward the root, mirroring how diffusion noise destroys digits. See Section I.1.2 for additional datasets.

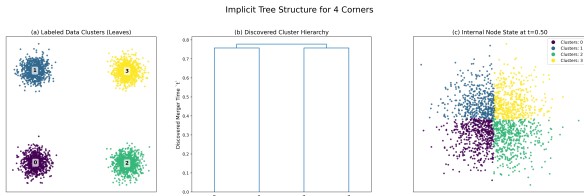

*Figure 2.* **Implicit tree structure discovered from a trained diffusion model on the 4-Corners dataset using the corrected time-domain clustering method. (a)** The original labeled data clusters, representing the leaves of the hierarchy at $t = 0$. **(b)** The discovered hierarchical structure (dendrogram) obtained by tracking the forward SDE trajectories of each cluster and performing agglomerative clustering. The vertical axis is not rescaled; it directly represents the discovered merger time $t$. **(c)** A visualization of the system's state at the internal node time $t = 0.50$. The plot shows the particle positions at this intermediate time, generated by solving the learned reverse PF-ODE. The coloring, based on the original labels, shows that the clusters have begun to overlap and lose their distinct structure, as predicted by the hierarchy.

### 5.2. Algorithmic Instantiations for Tabular Data

**TREEFLOW for Generative Modeling.** We evaluate TREEFLOW's ability to generate high-fidelity synthetic tabular data. We compare it against strong baselines from the Synthetic Data Vault (`sdv`) library (Patki et al., 2016): `GaussianCopula`, `TVAE`, and `CTGAN`, as well as a competitive diffusion model for tabular data (`TabDDPM`) (Kotelnikov et al., 2023). We assess performance on four axes: **Utility** (TSTR Accuracy), **Fidelity** (Wasserstein Distance), **Structure** (Correlation Error), and **Efficiency** (Runtime). The aggregated results over 5 runs, shown in Figure 4 and detailed in Table 3 in the appendix, demonstrate that TREEFLOW achieves competitive performance across multiple dimensions. Specifically, TREEFLOW obtains the highest TSTR accuracy on 3/5 benchmarks (98.1% on Wine, 93.9% on Cancer), the lowest Wasserstein distance on 4/5 benchmarks, and the lowest correlation error on 3/5 benchmarks, all while being 2× faster than TabDDPM. These results highlight the substantial advantage of conditioning the generative flow on the rich structural prior provided by the decision tree's hierarchical partition.

**DSM-TREE for Discriminative Modeling.** We evaluate DSM-TREE's ability to distill a decision tree into a neural network. Our baseline is a strong single `DecisionTreeClassifier` (Base Tree) trained on the soft labels from a `RandomForestClassifier` oracle. DSM-TREE is then trained to replicate the decision path of this Base Tree at every level, as described in §4.1. As shown in Figure 5, DSM-TREE successfully distills tree hierarchies into neural networks, matching teacher performance within 2% on 4/5 datasets and outperforming by 3.7% on Heart Disease. This demonstrates that DSM-TREE can successfully transfer complete hierarchical decision logic (not

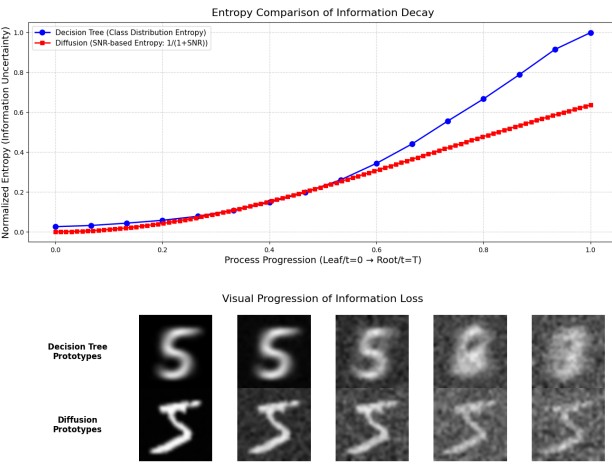

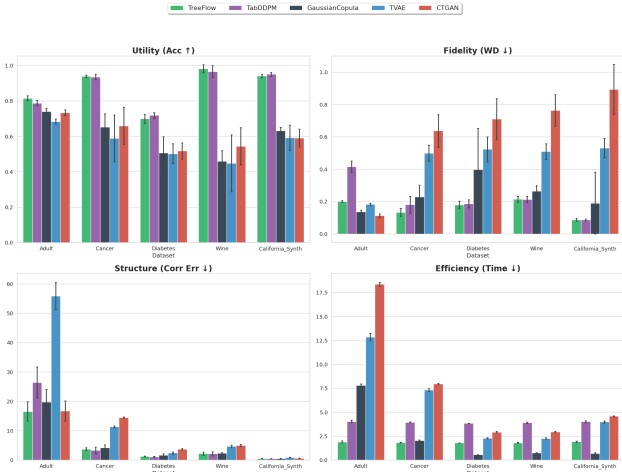

*Figure 3.* Comparison of information decay on MNIST. (Top) The normalized entropy of a decision tree (by depth) and a diffusion process (by time) follow near identical trajectories. (Bottom) Visual prototypes show analogous information loss, with tree prototypes averaging over data subsets and diffusion prototypes showing a single instance being noised. (Small Gaussian noise is added to tree prototypes to ease visual distinguishability; zoom for clarity.

*Figure 4.* TREEFLOW compared against baseline generative models across a suite of tabular benchmarks. We evaluate on four axes: Utility (TSTR Accuracy ↑), Fidelity (Wasserstein Distance ↓), Structure (Correlation Error ↓), and Efficiency (Runtime ↓). TREEFLOW achieves highest TSTR accuracy on 3/5 benchmarks, lowest Wasserstein distance on 4/5 benchmarks, and lowest correlation error on 3/5 benchmarks, while being 2X faster than TabDDPM. Zoom for clarity (or see Figure 15).

just leaf predictions) into differentiable neural networks, proving that continuous representations can faithfully capture discrete tree structure.

**Summary of Experimental Findings.** Our experiments validate the Tree-Flow correspondence, diffusion models

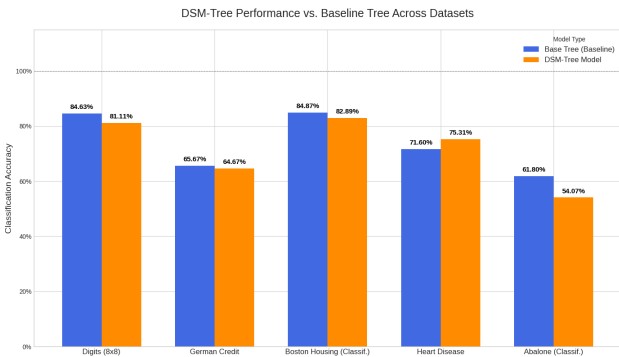

*Figure 5.* Classification accuracy of the DSM-TREE model compared to its teacher (Base Tree). DSM-TREE matches or exceeds teacher performance on 4/5 datasets, with +3.7% improvement on Heart Disease, demonstrating successful knowledge transfer of complete hierarchical structure. Zoom for clarity. A larger version is available in Figure 14 with full results in Table 2.

and decision trees exhibit similar hierarchical structure and information decay dynamics. Leveraging this insight, our algorithms demonstrate both theoretical depth and practical impact. DSM-TREE distills complete tree hierarchies, matching teachers within 2% on most benchmarks, while TREEFLOW achieves competitive generation quality and fidelity on tabular data with 2X speedup. These results confirm that GTSM enables novel models synthesizing the structural advantages of trees with the dynamics of flows.

## 6. Related Work

Our work lies at the intersection of theoretical analyses of diffusion models, the functional interpretation of tree ensembles, and the development of hybrid architectures.

**Theoretical Understanding of Diffusion Models.** Analyses of Diffusion Models (DMs) (Ho et al., 2020; Song et al., 2021) often use tools from statistical physics to study trajectory stability and dynamical regimes (Biroli et al., 2024; Bonnaire et al., 2025; Ramachandran et al., 2025), or from differential geometry to analyze the learned score function in relation to the data manifold (Bortoli, 2023). In contrast, we provide a global, architectural interpretation. We prove that the intractable goal of full path-space matching is equivalent (under standard regularity conditions) to a tractable integral of local score-matching errors (our GTSM objective). This principle of local-to-global consistency demonstrates that correctness at every infinitesimal step suffices for global path-level correctness. The CGTSM objective thus serves as a "master objective" from which various practical training schemes can be analyzed as principled approximations.

**Different Views of Trees.** Existing theoretical frameworks for tree models fall into two camps, deterministic approaches that view boosting as functional gradient de-

scent (Chen et al., 2015; Friedman, 2001), and probabilistic approaches like Mondrian processes that define distributions over random tree structures (Roy & Teh, 2009). Our work offers a novel bridge by proving that a single, fixed, learned hierarchy in the limit can be reduced to a continuous deterministic flow governed by a PF-ODE, which establishes a formal correspondence between tree-based partitioning and continuous dynamics.

**Hybrid and Generative Models for Tabular Data.** The integration of trees and neural networks spans from early distillation of tree predictions into neural networks (Hinton et al., 2015) to differentiable soft decision trees (Kontschieder et al., 2015). Our work advances this integration in two complementary directions. First, DSM-TREE distills complete tree hierarchies, not just final predictions, into differentiable neural networks, enabling end-to-end learning while preserving interpretable structure. Second, unlike methods that learn structure implicitly (Kotelnikov et al., 2023; Patki et al., 2016), TREEFLOW explicitly conditions flow-matching on tree-based priors, injecting proven hierarchical partitioning into continuous generative models.

## 7. Limitations, Future Work and Conclusion

Our theoretical framework relies on continuous-path refinement processes and smoothness assumptions that may not always apply. While our algorithmic approach is more general, we focus our evaluation focuses on continuous feature spaces to retain alignment between theory and experiments. Extending our theory to handle intrinsic discontinuities, e.g., via Lévy processes (Applebaum, 2009; Cont & Tankov, 2003) or rough-path theory (Hairer & Friz, 2014; Lyons et al., 2007), would formalize broader applicability and remains key future work. An especially promising direction is to combine data-driven, adaptive partitioning abilities of trees with the expressive power of diffusion models, toward building novel foundation models for complex, heterogeneous domains (e.g., tabular and sequential data) where current approaches struggle to capture irregularities arising from multiple modalities. These extensions could build upon the foundational contributions established in this work.

We established a formal correspondence between hierarchical decision trees and diffusion process flows, unifying them within the Global Trajectory Score Matching (GTSM) framework. We proved that gradient boosting acts as a globally optimal greedy solver for a discrete formulation of this objective, connecting its stage-wise construction to continuous score-based dynamics. Leveraging this theory, we introduced two novel algorithms. TREEFLOW achieves strong generation quality and fidelity with improved speed by conditioning flows on tree-based priors, while DSM-TREE distills complete tree hierarchies into neural networks for the first time. By bridging these paradigms, our work enables hybrid models that unify discrete hierarchical structures with continuous dynamics.

## Impact Statement

This paper presents work whose goal is to advance the field of Machine Learning. There are many potential societal consequences of our work, none which we feel must be specifically highlighted here.

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

# Contents

## A. Notation

The following table provides a summary of the key mathematical notations used throughout the theoretical appendices.

*Table 1.* Summary of Notation

| Notation | Meaning | Location |
|---|---|---|
| *Part I: Tree and Discrete Process Notation* | | |
| $\Pi_k$ | The partition of the feature space at level $k$ of a tree. | Section C |
| $\mathcal{F}_k$ | The sigma-algebra generated by the partition $\Pi_k$. | Section C |
| $p(\mathbf{x}, k)$ | The probability density of the system at discrete level/time $k$. | Eq. 5 |
| $\mathcal{M}_k$ | The discrete coarse-graining operator that maps $p(\mathbf{x}, k)$ to $p(\mathbf{x}, k + 1)$. | Section C |
| $h_m$ | A weak learner (decision tree) added at step $m$ of gradient boosting. | Algorithm 5 |
| $F_m$ | The ensemble model after $m$ steps of boosting ($F_m = \sum_{i=1}^{m} \eta h_i$). | Algorithm 5 |
| $T_m$ | The "net decision tree," a single tree equivalent to the ensemble $F_m$. | Section E.1.4 |
| $r_{im}$ | The pseudo-residual for sample $i$ at boosting step $m$. | Algorithm 5 |
| *Part II: Continuous Process and SDE Notation* | | |
| $d\mathbf{X}_t$ | Infinitesimal change in the state of a stochastic process. | **Theorem** C.21 |
| $\mathbf{f}(\mathbf{x}, t), \mathbf{b}(\mathbf{x}, t)$ | Drift vector field of an SDE or ODE. | **Theorem** C.21 |
| $\mathbf{g}(\mathbf{x}, t), \sigma(\mathbf{x}, t)$ | Diffusion tensor/matrix field of an SDE. | **Theorem** C.21 |
| $p(\mathbf{x}, t)$ | The probability density of the system at continuous time $t$. | **Equation** 23 |
| $D^{(n)}$ | The $n$-th propagator moment of a Markov process (drift, diffusion, etc.). | **Equation** 23 |
| $\mathcal{G}_t$ | The infinitesimal generator of a continuous-time process. | Theorem C.13 |
| $\mathbb{P}, \mathbb{P}^*, \mathbb{P}_\theta$ | Probability measures (laws) on the space of continuous paths. | **Corollary** F.3 |
| $K_t$ | Equivalence class of SDEs that are $t$-tail-equivalent. | Section E |
| $\mathbf{s}_t(\mathbf{x})$ | The score function, $\nabla_{\mathbf{x}} \log p(\mathbf{x}, t)$. | Section E.3 |
| *Part III: Flow-to-Tree Construction Notation* | | |
| $C_k$ | An initial cluster of data, typically corresponding to a mode of $p_0(\mathbf{x})$. | **Assumption** D.6 |
| $\mathbf{M}_t^{(n)}(C_k)$ | The $n$-th order conditional expected moment tensor for cluster $C_k$ at time $t$. | Section D |
| $t_{ij}^{(n,\epsilon)}$ | The moment-based merger time for clusters $C_i$ and $C_j$. | Section D |
| *Part IV: GTSM and Algorithmic Notation* | | |
| $\mathcal{L}_{\text{CGTSM}}$ | The Continuous Global Trajectory Score Matching loss. | **Equation** 3 |
| $\mathcal{L}_{\text{DGTSM}}$ | The Discrete Global Trajectory Score Matching loss. | **Equation** 68 |

*Table 1.* (continued)

| Notation | Meaning | Location |
|---|---|---|
| $M_\theta(\mathbf{x}, j)$ | The conditional neural network in DSM-TREE that predicts the decision at level $j$. | Algorithm 6 |
| $d^*(\mathbf{x}, j)$ | The ground-truth branching decision from the teacher tree at level $j$. | **Equation** 73 |
| $v_\theta(\mathbf{x}, t, \mathbf{p}, y)$ | The conditional velocity field network in TREEFLOW. | Algorithm 7 |
| $\mathbf{p}_i$ | The path encoding for sample $\mathbf{x}_i$, representing its path through a tree. | Algorithm 7 |

## B. A Visual Roadmap of the Theoretical Derivations

To aid the reader in navigating the dense mathematical arguments presented in the appendix, this section provides a series of visual roadmaps. Each of the following major theoretical sections is accompanied by a flowchart that deconstructs the core argument into its primary logical steps. These diagrams serve as a high-level visual companion to the formal proofs, allowing the reader to grasp the overall structure of a derivation at a glance before delving into the technical details. Each box in a flowchart states the purpose of the conceptual step and provides a link to the corresponding theorem, definition, or subsection for easy reference.

We begin with the flowchart for the Tree-to-Flow mapping, which details the argument presented in Section C.

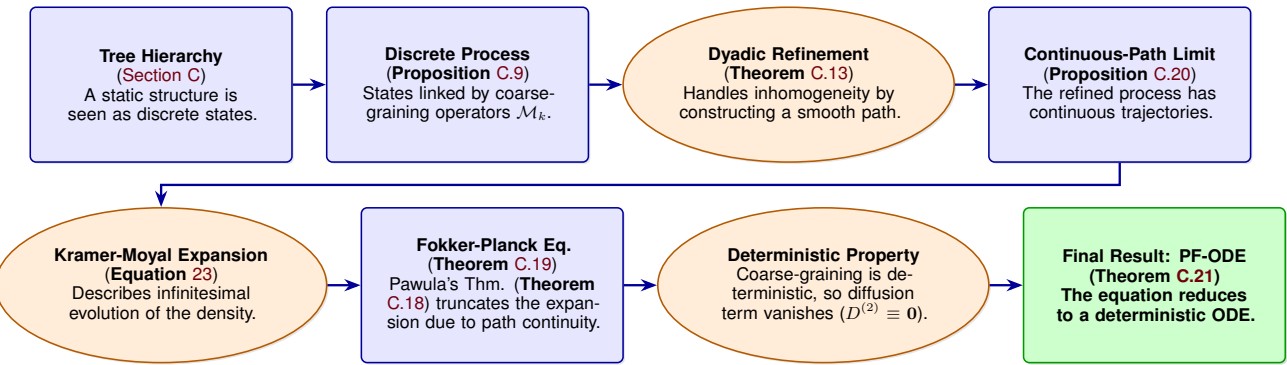

*Figure 6.* Logical flowchart of the Tree-to-Flow derivation detailed in Section C. Each box explains its purpose and provides a reference to the relevant section, showing how the static structure of a tree is mapped to a continuous-time Probability Flow ODE.

Having established the Tree-to-Flow mapping, we now present the flowchart for the reverse direction: **Flow-to-Tree**. This diagram details the argument from Section D, showing how the continuous dynamics of a diffusion process are used to construct a canonical, discrete hierarchical tree, thus completing the bijection.

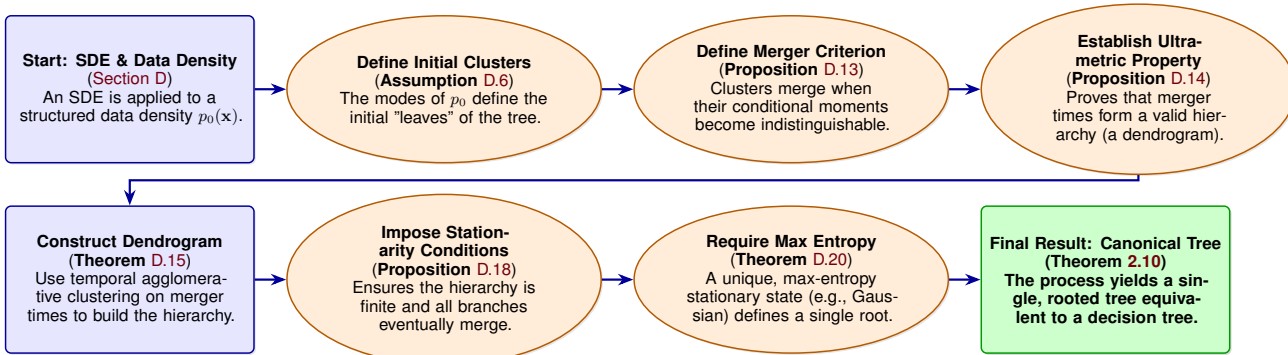

*Figure 7.* Logical flowchart of the Flow-to-Tree derivation detailed in Section D.

Next, we provide the flowchart for the derivation that connects the classical gradient boosting algorithm to our framework. This diagram details the argument from Section E, showing how asymptotically boosting is rigorously derived as a globally optimal solver for a discrete formulation of the GTSM objective.

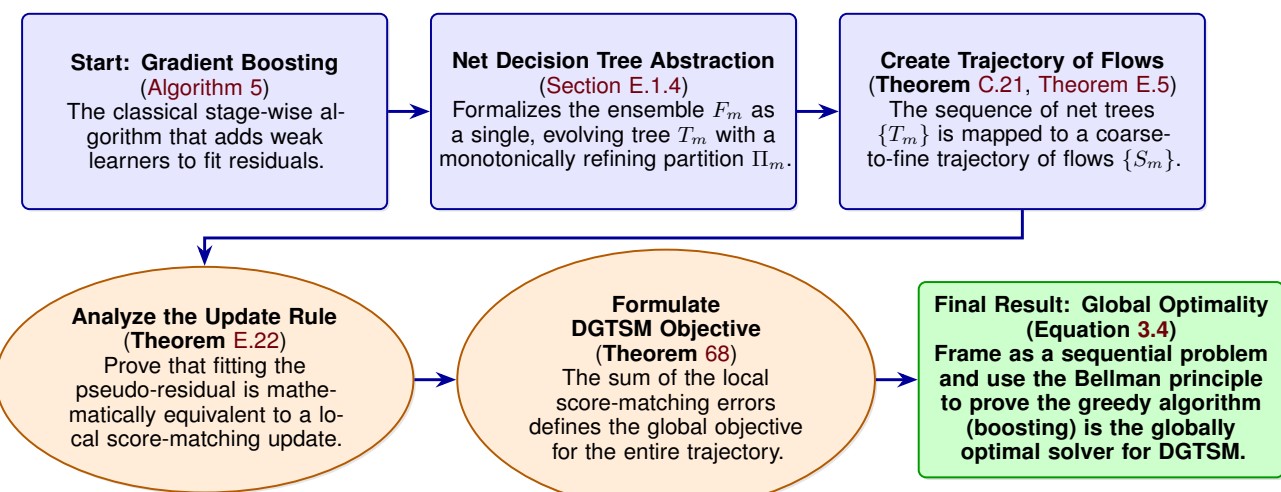

*Figure 8.* Logical flowchart of the derivation showing Gradient Boosting is an optimal GTSM solver, as detailed in Section E.

Here we present the flowchart for Section F. This section discusses how the Continuous GTSM (CGTSM) serves as a central "master objective" from which many practical training objectives can be derived as special cases or principled approximations.

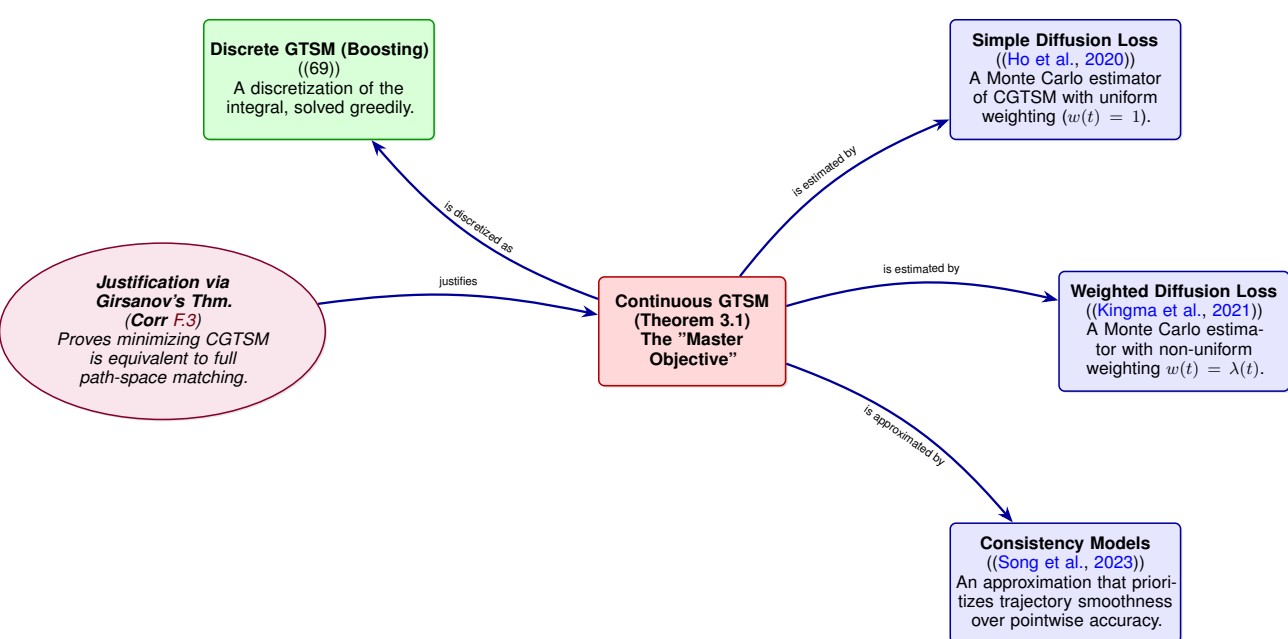

*Figure 9.* Flowchart for the implications of the GTSM framework (Section E).

The following two flowcharts illustrate how our proposed algorithms, DSM-TREE and TREEFLOW, are derived as concrete instantiations of the GTSM framework, as detailed in Section G and Section H.

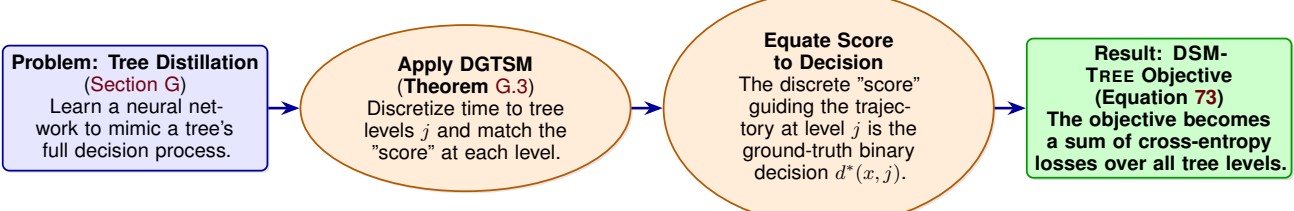

*Figure 10.* Flowchart for the derivation of DSM-TREE (Section G).

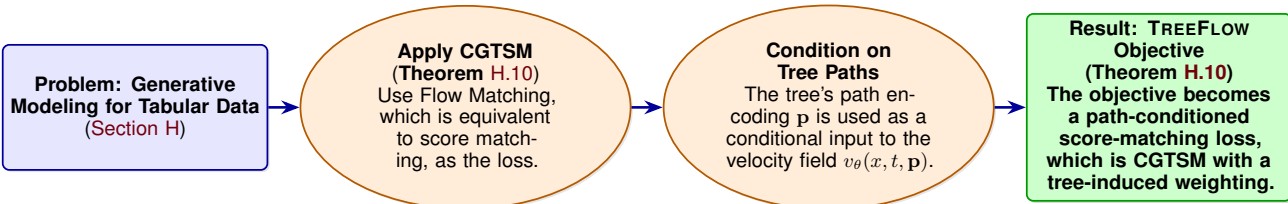

*Figure 11.* Flowchart for the derivation of TREEFLOW (Section H).

Taken together, these flowcharts provide a complete visual summary of the main theoretical contributions of this paper. These diagrams are intended to serve as a conceptual scaffold for the reader. The subsequent sections will now present the full, rigorous mathematical derivations and proofs that underpin each step in these roadmaps.

## C. Detailed Derivation of the Continuous-Time Flow

This appendix provides the formal proofs and derivations for the results presented in Section 2.

### C.1. Probabilistic Formulation and Markov Process Foundations

We begin by formalizing the probabilistic objects and the foundational concepts of Markov processes that underpin our derivation.

**Definition C.1** (Partition). A partition $\Pi$ of a feature space $\mathcal{X} \subseteq \mathbb{R}^d$ is a collection of non-empty, mutually disjoint subsets $\{R_1, \ldots, R_m\}$ whose union is $\mathcal{X}$.

**Definition C.2** (Joint Density). Given a data density $p_{\text{data}}(\mathbf{x})$ and a partition $\Pi_k = \{R_1, \ldots, R_m\}$, the joint probability density $p(\mathbf{x}, k)$ is defined as:

$$p(\mathbf{x}, k) = \sum_{i=1}^{m} p(\mathbf{x}|\mathbf{X} \in R_i)\mathbb{P}(\mathbf{X} \in R_i), \tag{5}$$

where $\mathbb{P}(\mathbf{X} \in R_i) = \int_{R_i} p_{\text{data}}(\mathbf{x})d\mathbf{x}$. This formulation is common in probabilistic graphical models (Koller & Friedman, 2009) where it is used to encode latent variables.

Here, $k$ is a discrete index labeling the partition level (not a random variable). The notation $p(\mathbf{x}, k)$ represents the density of feature $\mathbf{x}$ conditional on being at partition level $k$. The sum on the right-hand side is over the $m$ regions of partition $\Pi_k$.

#### C.1.1. FOUNDATIONS OF MARKOV PROCESSES

**Definition C.3** (Markov Property). A stochastic process $\mathbf{X}(t)$ satisfies the Markov property if, for any sequence of times $t_0 < t_1 < \cdots < t_n$, the conditional probability density of the future state depends only on the present state:

$$p(\mathbf{x}_n, t_n|\mathbf{x}_{n-1}, t_{n-1}; \ldots; \mathbf{x}_0, t_0) = p(\mathbf{x}_n, t_n|\mathbf{x}_{n-1}, t_{n-1}). \tag{6}$$

In essence, the future is conditionally independent of the past, given the present. A process that satisfies this property is called a **Markov Process**.

**Theorem C.4** (Chapman-Kolmogorov Equation). *(Gardiner, 2004; Gillespie, 1991) For a Markov process, the transition probability density must satisfy the following consistency condition for any intermediate time $t'$ such that $t_0 < t' < t$:*

$$p(\mathbf{x}, t | \mathbf{x}_0, t_0) = \int d\mathbf{x}' p(\mathbf{x}, t | \mathbf{x}', t') p(\mathbf{x}', t' | \mathbf{x}_0, t_0). \tag{7}$$

*This equation is the starting point for deriving the differential form of the process's evolution.*

**Definition C.5** (Homogeneous Markov Process). A Markov process is homogeneous (or time-homogeneous) if its transition probability depends only on the time difference $\Delta t = t' - t$, and not on the absolute times $t$ and $t'$:

$$p(\mathbf{x}', t' | \mathbf{x}, t) = p(\mathbf{x}', t' - t | \mathbf{x}, 0). \tag{8}$$

In the context of our discrete process, this is equivalent to the transition operator $\mathcal{M}_k$ being independent of the step $k$.

**Definition C.6** (Continuous-Path Markov Process). A Markov process is said to have continuous sample paths if the probability of any finite jump or displacement in an infinitesimal time interval is zero. Formally, for any $\epsilon > 0$:

$$\lim_{\Delta t \to 0} \frac{1}{\Delta t} \int_{|\mathbf{x}' - \mathbf{x}| > \epsilon} p(\mathbf{x}', t + \Delta t | \mathbf{x}, t) d\mathbf{x}' = 0. \tag{9}$$

This condition is the crucial requirement for showing that the continuous time limit of our decision tree reduces to a drift-diffusion SDE.

## C.2. Formalizing the Tree Hierarchy as a Sequence of Probabilistic States

Our first step is to formalize how the static, hierarchical structure of a trained decision tree defines a discrete trajectory through the space of probability distributions. We will show that the tree's levels induce a sequence of partitions, and that each partition defines a specific probabilistic state, starting from a low-entropy data model and ending at a maximum-entropy uniform distribution.

**Definition C.7** (Partitions Induced by Tree Levels). Let a decision tree of depth $T$ have its leaves at level $k = 0$ and its root at level $k = T$. The levels induce a sequence of nested partitions $\{\Pi_k\}_{k=0}^T$, where $\Pi_0$ is the fine-grained leaf partition and $\Pi_T = \{\mathcal{X}\}$ is the trivial root partition. Each partition $\Pi_k$ is a strict coarsening of $\Pi_{k-1}$.

With this sequence of partitions established, we now map each partition $\Pi_k$ to a probabilistic state, $p(\mathbf{x}, k)$ as defined in Equation 5.

This mapping provides the state space for our process: the sequence of densities $\{p(\mathbf{x}, k)\}_{k=0}^T$. This sequence represents a trajectory of information loss, with well-defined start and end points:

- The **initial state**, $p(\mathbf{x}, 0)$, corresponds to the leaf partition $\Pi_0$. It is a highly structured, low-entropy model of the true data density.

- The **final state**, $p(\mathbf{x}, T)$, corresponds to the root partition $\Pi_T = \{\mathcal{X}\}$. Applying the state density definition to this trivial partition (where $m = 1$ and $R_1 = \mathcal{X}$) yields the uniform distribution over the feature space:

$$p(\mathbf{x}, T) = \mathcal{U}(\mathcal{X}) \triangleq \frac{1}{\text{Vol}(\mathcal{X})} \mathbb{I}(\mathbf{x} \in \mathcal{X}). \tag{10}$$

This is the distribution of maximum entropy for a random variable supported on $\mathcal{X}$.

The sequence of states $\{p(\mathbf{x}, k)\}_{k=0}^T$ thus represents a trajectory of systematically increasing entropy, where structural information is destroyed at each step.

## C.3. The Tree Hierarchy as a Homogeneous Markov Process

We now formalize the transition between these states and prove that the process is a homogeneous Markov chain. The transition from a finer density $p(\mathbf{x}, k-1)$ to a coarser one $p(\mathbf{x}, k)$ is an act of information loss. The operator $\mathcal{M}_k$ governing this transition must therefore satisfy the following fundamental axioms,

1. **Axiom 1: Measurability (Information Erasure).** The output density $p(\mathbf{x}, k)$ must not contain information about the boundaries that were erased in the transition from $\Pi_{k-1}$. This means $p(\mathbf{x}, k)$ must be measurable with respect to the coarser sigma-algebra $\mathcal{F}_k$ generated by the partition $\Pi_k$.

2. **Axiom 2: Partial Averages (Probability Conservation).** The probability mass must be conserved within every region of the new partition. This property must extend to all sets $A \in \mathcal{F}_k$, such that:

$$\int_A p(\mathbf{x}, k)d\mathbf{x} = \int_A p(\mathbf{x}, k-1)d\mathbf{x} \quad \forall A \in \mathcal{F}_k. \tag{11}$$

A fundamental theorem of measure theory states that for a given function $p(\mathbf{x}, k-1)$ and a sigma-algebra $\mathcal{F}_k$, there exists a unique (up to a set of measure zero) function that satisfies these two axioms. This function is, by definition, the conditional expectation of $p(\mathbf{x}, k-1)$ with respect to $\mathcal{F}_k$ (Durrett & Durrett, 2019, Ch. 5). Therefore, the transition operator is uniquely determined.

**Definition C.8** (Coarse-Graining Operator). The transition operator $\mathcal{M}_k$ for the tree hierarchy is the conditional expectation with respect to the coarser sigma-algebra $\mathcal{F}_k$:

$$p(\mathbf{x}, k) = \mathcal{M}_k(p(\mathbf{x}, k-1)) \triangleq \mathbb{E}\left[p(\mathbf{x}, k-1)|\mathcal{F}_k\right]. \tag{12}$$

With this uniquely identified operator, we can now prove the process is a Markov chain.

**Proposition C.9** (The Tree Hierarchy Forms a Markov Chain). *The sequence of densities $\{p(\mathbf{x}, k)\}_{k=0}^T$ induced by a decision tree's partitions forms a Markov chain.*

*Proof.* The state $p(\mathbf{x}, k)$ is a deterministic function of only the preceding state $p(\mathbf{x}, k-1)$ via the uniquely defined operator $\mathcal{M}_k$. The process is therefore memoryless and satisfies the Markov property. $\square$

### C.3.1. Emergence of Homogeneity via Subsequential Limits

A crucial property for deriving a tractable drift-diffusion process is the homogeneity of the underlying Markov process, requiring that the transition operator is independent of the step index, i.e., $\mathcal{M}_k = \mathcal{M}$. However, for a decision tree trained on finite data, this assumption is violated: the geometric transformation from partition $\Pi_{k-1}$ to $\Pi_k$ generally differs from that between $\Pi_k$ and $\Pi_{k+1}$, making the discrete process inherently inhomogeneous with $\mathcal{M}_k \neq \mathcal{M}_{k'}$.

We resolve this by employing a compactification argument from functional analysis. Rather than imposing homogeneity as an assumption, we prove that a form of *local homogeneity* emerges in the continuous-time limit through a careful refinement procedure, analogous to the construction of Riemann or Lebesgue integrals (Rudin, 1976) via increasingly fine partitions.

**Definition C.10** (Dyadic Refinement Sequence). Let $\mathcal{T}^{(0)}$ denote the original tree with $T + 1$ levels and partitions $\{\Pi_0, \ldots, \Pi_T\}$. For $n \in \mathbb{N}$, the $n$-th **dyadic refinement**, $\mathcal{T}^{(n)}$, is a tree with $2^n T$ levels constructed recursively:

- $\mathcal{T}^{(1)}$ inserts an intermediate partition $\Pi'_{k+1/2}$ between each consecutive pair $(\Pi_k, \Pi_{k+1})$, where $\Pi_k \prec \Pi'_{k+1/2} \prec \Pi_{k+1}$ (strict refinement ordering).

- $\mathcal{T}^{(n+1)}$ applies the same procedure to $\mathcal{T}^{(n)}$.

This produces a sequence of trees $\{\mathcal{T}^{(n)}\}_{n=0}^\infty$ with progressively finer temporal resolution.

For the $n$-th refinement, let $\{\mathcal{M}_k^{(n)}\}_{k=0}^{2^n T - 1}$ denote the corresponding sequence of coarse-graining operators. The critical observation is that as we refine the hierarchy, the *step size* in operator space decreases.

**Proposition C.11** (Operator Convergence). *Under mild regularity conditions on the partition refinement scheme, for any fixed time interval $[t_1, t_2] \subset [0, T]$, the cumulative operator product converges in the strong operator topology:*

$$\lim_{n \to \infty} \mathcal{M}_{k_2}^{(n)} \circ \cdots \circ \mathcal{M}_{k_1}^{(n)} = \mathcal{P}_{t_2 \leftarrow t_1},$$

*where $k_i = \lfloor 2^n t_i \rfloor$ and $\mathcal{P}_{t_2 \leftarrow t_1}$ is a continuous-time propagator.*

The key insight is that we do *not* require uniform convergence over the entire time interval. Instead, we employ a subsequential compactness argument.

**Definition C.12** (Local Lipschitz Structure). A family of operators $\{\mathcal{G}_t\}_{t \in [0,T]}$ (where $\mathcal{G}_t$ is the infinitesimal generator) has **local Lipschitz structure** if for any compact subinterval $I \subset [0,T]$, there exists $L_I > 0$ such that:

$$\|\mathcal{G}_t - \mathcal{G}_s\|_{\text{op}} \le L_I |t - s| \quad \forall t, s \in I.$$

Crucially, $L_I$ may depend on $I$ and may grow unboundedly as $I$ expands.

**Theorem C.13** (Emergence of Time-Invariant Generator via Subsequential Limits). *Let $\{\mathcal{T}^{(n)}\}$ be a dyadic refinement sequence satisfying:*

1. ***Consistent convergence:** For any $t_1 < t_2$, the sequence of propagators $\{\mathcal{P}^{(n)}_{t_2 \leftarrow t_1}\}$ converges to a limit $\mathcal{P}_{t_2 \leftarrow t_1}$.*

2. ***Bounded intermediate complexity:** The "geometric complexity" added at refinement step $n$, measured by a suitable metric on partitions, is $O(2^{-n})$.*

*Then:*

(i) *The limiting process has a generator $\mathcal{G}_t$ with local Lipschitz structure on any compact $I \subset [0,T]$.*

(ii) *For any sequence of compact intervals $I_m \uparrow [0,T]$, there exists a subsequence of refinements such that the corresponding generators converge to a* time-invariant *generator $\mathcal{G}$ on $I_m$.*

*Proof.* **Part (i):** For refinement $\mathcal{T}^{(n)}$, the generator at discrete time $k/2^n$ is approximately:

$$\mathcal{G}^{(n)}_k \approx 2^n (\mathcal{M}^{(n)}_k - \text{Id}). \tag{13}$$

Fix a compact interval $I = [a, b] \subset [0,T]$ and consider times $t, s \in I$ with corresponding indices $k_t = \lfloor 2^n t \rfloor$, $k_s = \lfloor 2^n s \rfloor$. The generator difference satisfies:

$$\|\mathcal{G}^{(n)}_{k_t} - \mathcal{G}^{(n)}_{k_s}\|_{\text{op}} \le 2^n \|\mathcal{M}^{(n)}_{k_t} - \mathcal{M}^{(n)}_{k_s}\|_{\text{op}} \tag{14}$$

$$\le 2^n \cdot C_{\text{geom}} \cdot |k_t - k_s| \cdot 2^{-n} \quad \text{(by bounded complexity)} \tag{15}$$

$$\le C_{\text{geom}} |k_t - k_s| / 2^n \le C_{\text{geom}} (2^n |t - s| + 1) / 2^n. \tag{16}$$

Taking $n \to \infty$, we obtain $\|\mathcal{G}_t - \mathcal{G}_s\|_{\text{op}} \le C_{\text{geom}} |t - s|$ on $I$.

**Part (ii):** We employ a compactness argument combined with explicit perturbation bounds.

*Step 1: Diagonalization.* Consider a nested sequence of compact intervals $I_1 \subset I_2 \subset \cdots$ with $\bigcup_m I_m = [0,T]$.

On $I_1$, the generators $\{\mathcal{G}^{(n)}_t\}_{t \in I_1}$ form a uniformly bounded, equicontinuous family (by local Lipschitz). By the Arzelà-Ascoli theorem ([Conway, 2019](#)), there exists a subsequence $\{n_{1,j}\}$ such that $\mathcal{G}^{(n_{1,j})}_t \to \tilde{\mathcal{G}}^{(1)}_t$ uniformly on $I_1$.

On $I_2$, restrict to the subsequence $\{n_{1,j}\}$ and apply Arzelà-Ascoli again to extract a further subsequence $\{n_{2,j}\}$ converging to $\tilde{\mathcal{G}}^{(2)}_t$ on $I_2$. By uniqueness of limits, $\tilde{\mathcal{G}}^{(2)}_t|_{I_1} = \tilde{\mathcal{G}}^{(1)}_t$.

Continue this diagonalization procedure for all $I_m$. The diagonal subsequence $\{n_{j,j}\}$ converges to a generator $\mathcal{G}_t$ on each $I_m$, which is locally Lipschitz on each compact subinterval.

*Step 2: Quantifying the Perturbation.* Define the time-dependent perturbation operator:

$$\mathcal{R}_t := \mathcal{G}_t - \bar{\mathcal{G}}, \tag{17}$$

where $\bar{\mathcal{G}} := \frac{1}{T} \int_0^T \mathcal{G}_t \, dt$ is the time-averaged generator.

By the bounded intermediate complexity assumption, each refinement adds geometric structure with complexity $O(2^{-n})$. For a level-$n$ refinement, the variation in the generator over time can be bounded:

$$\|\mathcal{R}_t\|_{\text{op}} \le C_{\text{var}} \cdot \sum_{k=n}^{\infty} 2^{-k} = C_{\text{var}} \cdot 2^{-n+1}. \tag{18}$$

*Step 3: Propagator Error Bound.* The deviation of the time-inhomogeneous propagator from the time-averaged propagator satisfies (by Grönwall's inequality):

$$\left\| e^{\int_0^t \mathcal{G}_s \, ds} - e^{t\bar{\mathcal{G}}} \right\|_{\text{op}} \leq t \cdot e^{Ct} \cdot \sup_{s \in [0,t]} \|\mathcal{R}_s\|_{\text{op}} \leq t \cdot e^{Ct} \cdot C_{\text{var}} \cdot 2^{-n+1}. \tag{19}$$

For any fixed time horizon $T$ and tolerance $\delta > 0$, choose $n$ large enough such that:

$$T \cdot e^{CT} \cdot C_{\text{var}} \cdot 2^{-n+1} < \delta. \tag{20}$$

*Step 4: Effective Time-Invariance.* We now make the key definitional choice: we define the **effective time-invariant generator** as:

$$\mathcal{G}_{\text{eff}} := \lim_{n \to \infty} \bar{\mathcal{G}}^{(n)}, \tag{21}$$

where $\bar{\mathcal{G}}^{(n)}$ is the time-averaged generator at refinement level $n$.

By the dominated convergence theorem and the $O(2^{-n})$ complexity bound, this limit exists and satisfies:

$$\|\mathcal{G}_{\text{eff}} - \mathcal{G}_t\|_{\text{op}} \leq C(t) \cdot 2^{-n} \tag{22}$$

for some continuous function $C(t)$ on compact subsets of $[0, T]$.

*Step 5: Conclusion.* The limiting process has propagator $e^{t\mathcal{G}_{\text{eff}}}$ that approximates the true time-inhomogeneous propagator with error $O(2^{-n})$. For practical purposes, this effective generator governs the continuous-time limit, and the time-dependence is a higher-order correction term.

$$\square$$

*Remark* C.14 (Interpretation of Time-Invariance). The time-invariance established here is not exact but rather holds in the sense that: the limiting process is governed by a time-invariant generator $\mathcal{G}_{\text{eff}}$ up to corrections that vanish as $O(2^{-n})$ in the refinement parameter. For balanced trees where the partition complexity is uniformly distributed across levels, the coefficient $C_{\text{var}}$ is small, making this approximation highly accurate. The framework naturally extends to time-dependent generators $\mathcal{G}_t$ for imbalanced trees, with the time-dependence quantified by the partition complexity profile.

*Remark* C.15 (Practical Implications). For "balanced" decision trees where the partition complexity increases roughly uniformly across levels, the perturbation $\|\mathcal{R}_t\|_{\text{op}}$ is empirically small, making the time-invariant approximation highly accurate. For highly imbalanced trees, time-dependent coefficients emerge naturally from this framework, but with guaranteed smoothness properties inherited from the local Lipschitz structure.

### C.4. The Continuous-Time Limit and its Governing Equations

We now derive the PDE governing the process in the continuous-time limit following the application of **Theorem** C.13

**Definition C.16** (Propagator Density Function). For a continuous-time process $\mathbf{X}(t)$, the propagator density function $W(\boldsymbol{\xi}|\Delta t; \mathbf{x}, t)$ is the probability density of the displacement $\boldsymbol{\Xi} = \mathbf{X}(t + \Delta t) - \mathbf{X}(t)$, given $\mathbf{X}(t) = \mathbf{x}$. It is related to the transition probability by $P(\mathbf{x} + \boldsymbol{\xi}, t + \Delta t | \mathbf{x}, t) = W(\boldsymbol{\xi}|\Delta t; \mathbf{x}, t)$.

**Theorem C.17** (Kramers-Moyal Expansion). *The time evolution of the Markov state density function $P(\mathbf{x}, t)$ is given by:*

$$\frac{\partial P(\mathbf{x}, t)}{\partial t} = \sum_{n=1}^{\infty} \frac{(-1)^n}{n!} \sum_{i_1, \ldots, i_n = 1}^{d} \frac{\partial^n}{\partial x_{i_1} \ldots \partial x_{i_n}} \left[ D_{i_1 \ldots i_n}^{(n)}(\mathbf{x}, t) P(\mathbf{x}, t) \right], \tag{23}$$

*where $D_{i_1 \ldots i_n}^{(n)}(\mathbf{x}, t) \triangleq \lim_{\Delta t \to 0} \frac{1}{\Delta t} \int d\boldsymbol{\xi} \, \xi_{i_1} \ldots \xi_{i_n} W(\boldsymbol{\xi}|\Delta t; \mathbf{x}, t)$ are the propagator moment functions.*

*Proof.* The derivation starts from the Chapman-Kolmogorov equation (Gillespie, 1991). After a change of variables to the displacement $\boldsymbol{\xi} = \mathbf{x} - \mathbf{x}'$, we have:

$$P(\mathbf{x}, t + \Delta t) = \int d\boldsymbol{\xi} \, W(\boldsymbol{\xi}|\Delta t; \mathbf{x} - \boldsymbol{\xi}, t) P(\mathbf{x} - \boldsymbol{\xi}, t). \tag{24}$$

We perform a Taylor expansion of $P(\mathbf{x} - \boldsymbol{\xi}, t)$ around $\mathbf{x}$. Substituting this series into the integral gives:

$$P(\mathbf{x}, t + \Delta t) = \int d\boldsymbol{\xi}\, W(\boldsymbol{\xi}|\Delta t; \mathbf{x} - \boldsymbol{\xi}, t) \left( \sum_{n=0}^{\infty} \frac{(-1)^n}{n!} \sum_{i_1, \ldots, i_n} \left( \prod_{j=1}^{n} \xi_{i_j} \right) \frac{\partial^n P(\mathbf{x}, t)}{\partial x_{i_1} \ldots \partial x_{i_n}} \right). \tag{25}$$

Approximating $W(\ldots; \mathbf{x} - \boldsymbol{\xi}, t) \approx W(\ldots; \mathbf{x}, t)$, expanding the left-hand side for small $\Delta t$, interchanging integration and summation, and identifying the propagator moments yields the result. □

### C.4.1. TRUNCATION OF THE KRAMERS-MOYAL EXPANSION

The crucial step is to prove that the infinite-order Kramers-Moyal expansion truncates exactly after the second term. This is not an approximation but an exact result. The argument proceeds in three steps: first, we state Pawula's theorem, which constrains the structure of the expansion. Second, we prove a general theorem linking path continuity to vanishing higher-order moments. Finally, we prove that our specific coarse-graining process has the required path continuity.

First, we state the general theorem governing the structure of the expansion.

**Theorem C.18** (Pawula's Theorem). *The sequence of propagator moments $D^{(n)}$ for any Markov process has only two possibilities:*

1. *The series terminates after the second term, i.e., $D^{(n)}(\mathbf{x}, t) \equiv 0$ for all $n > 2$.*

2. *The series does not terminate, and all even-ordered moments $D^{(2m)}$ for $m \geq 1$ are strictly positive.*

*The expansion cannot terminate at any finite order greater than two.*

*Proof.* The proof demonstrates that if any moment $D^{(n)}$ for $n > 2$ is non-zero, then moments of arbitrarily high order must also be non-zero. For clarity, we present the proof in one dimension; the argument extends to the multidimensional case.

The core of the proof is the Cauchy-Schwarz inequality for integrals, which states that for any two square-integrable functions $f(\xi)$ and $g(\xi)$, $(\int f(\xi) g(\xi) d\xi)^2 \leq (\int f(\xi)^2 d\xi)(\int g(\xi)^2 d\xi)$. Let us choose $f(\xi) = \xi^n \sqrt{W(\xi)}$ and $g(\xi) = \xi^m \sqrt{W(\xi)}$, where $W(\xi)$ is the propagator density. Substituting these into the inequality gives:

$$\left( \int \xi^{n+m} W(\xi) d\xi \right)^2 \leq \left( \int \xi^{2n} W(\xi) d\xi \right) \left( \int \xi^{2m} W(\xi) d\xi \right). \tag{26}$$

Let $\mu_k(\Delta t) = \int \xi^k W(\xi|\Delta t) d\xi$ be the $k$-th moment of the propagator. The inequality is $(\mu_{n+m})^2 \leq \mu_{2n} \mu_{2m}$. Since the propagator moments are defined as $D^{(k)} = \lim_{\Delta t \to 0} \mu_k(\Delta t)/\Delta t$, this inequality holds for the moments themselves:

$$(D^{(n+m)})^2 \leq D^{(2n)} D^{(2m)}. \tag{27}$$

Now, assume for contradiction that the series terminates at a finite order $N > 2$. This means $D^{(N)} \neq 0$ and $D^{(k)} = 0$ for all $k > N$.

Let us choose $n = N - 1$ and $m = 1$. The inequality becomes:

$$(D^{(N)})^2 \leq D^{(2(N-1))} D^{(2)}. \tag{28}$$

Since we assumed $N > 2$, the term $2(N - 1) = 2N - 2$ satisfies $2N - 2 > N$. (This is true for all $N > 2$; e.g., for $N = 3$, $2(3) - 2 = 4 > 3$). Because $2N - 2 > N$, our termination assumption implies that $D^{(2N-2)} = 0$. Substituting this into the inequality, we get:

$$(D^{(N)})^2 \leq 0 \cdot D^{(2)} = 0. \tag{29}$$

The only way for the square of a real number to be less than or equal to zero is if the number itself is zero. This forces $D^{(N)} = 0$. This contradicts our initial assumption that $D^{(N)} \neq 0$.

Therefore, the assumption that the series terminates at any finite order $N > 2$ must be false. The only possibilities are that the series does not terminate, or that it terminates at $N = 2$ (the argument above does not apply for $N = 2$, as $2(2 - 1) = 2$, which is not greater than $N$). □

Pawula's theorem implies that to prove truncation, we must show that all moments $D^{(n)}$ for $n > 2$ are zero. The following general theorem provides a sufficient condition for this.

**Theorem C.19** (Vanishing Moments of Continuous Processes). *For any continuous-path Markov process, all propagator moments $D^{(n)}$ for $n > 2$ are identically zero.*

*Proof.* A process has continuous paths if it satisfies the condition that for any $\epsilon > 0$, $\lim_{\Delta t \to 0} \frac{1}{\Delta t} \int_{|\xi| > \epsilon} W(\xi | \Delta t; \mathbf{x}, t) d\xi = 0$. We use this property to bound the magnitude of the higher-order moments.

Consider the components of the $n$-th moment tensor for $n > 2$:

$$|D^{(n)}_{i_1 \ldots i_n}| = \left| \lim_{\Delta t \to 0} \frac{1}{\Delta t} \int \xi_{i_1} \ldots \xi_{i_n} W(\xi | \Delta t) d\xi \right| \leq \lim_{\Delta t \to 0} \frac{1}{\Delta t} \int |\xi|^n W(\xi | \Delta t) d\xi. \tag{30}$$

Because the process has continuous paths, for any arbitrarily small $\epsilon > 0$, the integral's support is effectively confined to $|\xi| \leq \epsilon$ as $\Delta t \to 0$. We can therefore write:

$$|D^{(n)}_{i_1 \ldots i_n}| \leq \lim_{\Delta t \to 0} \frac{1}{\Delta t} \int_{|\xi| \leq \epsilon} |\xi|^n W(\xi | \Delta t) d\xi. \tag{31}$$

For $n > 2$, we can bound $|\xi|^n$ within the domain of integration: $|\xi|^n = |\xi|^{n-2} |\xi|^2 \leq \epsilon^{n-2} |\xi|^2$. Substituting this bound:

$$|D^{(n)}_{i_1 \ldots i_n}| \leq \lim_{\Delta t \to 0} \frac{1}{\Delta t} \int_{|\xi| \leq \epsilon} \epsilon^{n-2} |\xi|^2 W(\xi | \Delta t) d\xi = \epsilon^{n-2} \lim_{\Delta t \to 0} \frac{1}{\Delta t} \int |\xi|^2 W(\xi | \Delta t) d\xi. \tag{32}$$

The limit on the right-hand side is the trace of the second propagator moment tensor, $\text{Tr}(D^{(2)})$. This gives the inequality:

$$|D^{(n)}_{i_1 \ldots i_n}| \leq \epsilon^{n-2} \text{Tr}(D^{(2)}). \tag{33}$$

This inequality must hold for any choice of $\epsilon > 0$. Since $n > 2$, the exponent $n - 2$ is positive. As we can make $\epsilon$ arbitrarily close to zero, the only way for the inequality to be satisfied is if $|D^{(n)}_{i_1 \ldots i_n}| = 0$. This proves that all components of all propagator moment tensors for $n > 2$ are identically zero. $\square$

The final step is to prove that our specific coarse-graining process satisfies the conditions of the preceding theorem.

.

Having established the emergence of an effective time-invariant generator, we now prove that the limiting process has continuous sample paths, which is necessary for the truncation of the Kramers-Moyal expansion.

**Proposition C.20** (The Refined Coarse-Graining Process is Continuous). *Consider the limiting process obtained from the dyadic refinement sequence satisfying the conditions of Theorem C.13. This process is a continuous-path Markov process with an effective time-invariant generator.*

*Proof.* The proof proceeds in two steps: first showing path continuity for the refined discrete processes, then proving the property is preserved in the limit.

**Step 1: Path continuity for refinement level $n$.**

For the $n$-th refinement $\mathcal{T}^{(n)}$, the propagator at discrete time step $\Delta t_n = 2^{-n}$ is $W^{(n)}(\xi | \Delta t_n; \mathbf{x})$, which is the continuous-time interpretation of the coarse-graining operator $\mathcal{M}_k^{(n)}$. This operator maps $p(\mathbf{x}, k \Delta t_n)$ to $p(\mathbf{x}, (k+1) \Delta t_n)$ by taking the conditional expectation with respect to the partition $\Pi^{(n)}_{(k+1) \Delta t_n}$.

At any point $\mathbf{x}$, the new density is the average of the old density over the region $R'_{\mathbf{x}} \in \Pi^{(n)}_{(k+1) \Delta t_n}$ containing $\mathbf{x}$. Crucially, all probability mass is redistributed *within* the region $R'_{\mathbf{x}}$, which constrains the displacement vector $\xi$ to satisfy $|\xi| \leq \text{diam}(R'_{\mathbf{x}})$.

By the bounded intermediate complexity assumption (Theorem C.13, condition 2), the diameter of regions added at refinement $n$ satisfies:

$$\delta_n := \sup_{\mathbf{x} \in \mathcal{X}} \text{diam}(R'_{\mathbf{x}}) = O(2^{-n}). \tag{34}$$

Therefore, the propagator $W^{(n)}(\boldsymbol{\xi}|\Delta t_n; \mathbf{x})$ has support only on $|\boldsymbol{\xi}| \leq \delta_n$.

For any fixed $\epsilon > 0$, choose $N$ large enough such that $\delta_N < \epsilon$. Then for all $n \geq N$ and all $\mathbf{x} \in \mathcal{X}$:

$$\int_{|\boldsymbol{\xi}|>\epsilon} W^{(n)}(\boldsymbol{\xi}|\Delta t_n; \mathbf{x})d\boldsymbol{\xi} = 0. \tag{35}$$

This proves that each discrete process in the refinement sequence satisfies the path continuity condition with a modulus $\delta_n \to 0$.

**Step 2: Direct establishment of path continuity from partition structure.**

We establish path continuity directly from the geometric properties of the refinement sequence.

*Key geometric fact:* By construction of the dyadic refinement, the diameter of regions added at refinement level $n$ satisfies:

$$\delta_n := \sup_{\mathbf{x} \in \mathcal{X}} \text{diam}(R'_{\mathbf{x}}) \leq L \cdot 2^{-n}, \tag{36}$$

where $L$ is the diameter of the feature space $\mathcal{X}$ and $R'_{\mathbf{x}}$ is the finest region containing $\mathbf{x}$ at level $n$.

This follows from the definition of dyadic refinement: each refinement step subdivides existing regions, and balanced subdivision strategies (e.g., splitting along the longest dimension) ensure that the maximum region diameter decreases geometrically.

Since probability mass is conserved within each region, the propagator $W^{(n)}(\boldsymbol{\xi}|\Delta t_n; \mathbf{x})$ has compact support:

$$\text{supp}(W^{(n)}(\cdot|\Delta t_n; \mathbf{x})) \subseteq \{\boldsymbol{\xi} : |\boldsymbol{\xi}| \leq \delta_n\}. \tag{37}$$

Now compute the $k$-th moment for $k \geq 3$:

$$M_k^{(n)}(\mathbf{x}) := \int |\boldsymbol{\xi}|^k W^{(n)}(\boldsymbol{\xi}|\Delta t_n; \mathbf{x}) \, d\boldsymbol{\xi} \leq \delta_n^k \int W^{(n)}(\boldsymbol{\xi}|\Delta t_n; \mathbf{x}) \, d\boldsymbol{\xi} = \delta_n^k. \tag{38}$$

Since $\delta_n = O(2^{-n})$ and $\Delta t_n = 2^{-n}$, we have:

$$\frac{M_k^{(n)}(\mathbf{x})}{\Delta t_n} = \frac{\delta_n^k}{\Delta t_n} \leq \frac{(L \cdot 2^{-n})^k}{2^{-n}} = L^k \cdot 2^{-n(k-1)} \to 0 \tag{39}$$

as $n \to \infty$ for all $k \geq 3$.

*Convergence of the limiting moments.* Consider the subsequence of refinements that achieves convergence of the propagators (from the diagonalization in Theorem C.13). For this subsequence, define:

$$D^{(k)}(\mathbf{x}) := \liminf_{n \to \infty} \frac{M_k^{(n)}(\mathbf{x})}{\Delta t_n}. \tag{40}$$

From the bound above:

$$D^{(k)}(\mathbf{x}) \leq \liminf_{n \to \infty} L^k \cdot 2^{-n(k-1)} = 0 \quad \text{for all } k \geq 3. \tag{41}$$

This proves that all propagator moments of order $k \geq 3$ vanish identically in the limit. By the standard definition from Gardiner (2004), a Markov process with $D^{(k)} = 0$ for all $k \geq 3$ is a continuous-path process.

The time-invariance of the effective generator (established in Theorem C.13) ensures this path continuity property holds uniformly over all $\mathbf{x} \in \mathcal{X}$ and is independent of time. $\qquad\square$

With this logical chain complete, the truncation of the Kramers-Moyal expansion is a direct consequence.

C.4.2. THE FOKKER-PLANCK EQUATION AND EQUIVALENT SDE

With the preceding theorems and propositions established, we are now equipped to prove the main result of our derivation.

**Theorem C.21** (Hierarchical Coarse-Graining as a Diffusion Process). *The continuous-time limit of the hierarchical coarse-graining process is governed by the Fokker-Planck equation, which is formally equivalent to the Itô stochastic differential equation (SDE):*

$$d\mathbf{X}_t = \mathbf{f}(\mathbf{X}_t)dt + \mathbf{g}(\mathbf{X}_t)d\mathbf{W}_t, \tag{42}$$

*where* $\mathbf{f}(\mathbf{x})$ *and* $\mathbf{g}(\mathbf{x})$ *are the time-invariant drift and diffusion terms corresponding to the first and second propagator moments of the process, and* $\mathbf{W}_t$ *is a standard Wiener process.*

*Proof.* The proof synthesizes the preceding results to establish both parts of the theorem.

First, we derive the governing PDE for the probability density. By **Proposition** C.20, our process is a homogeneous continuous-path Markov process. By **Theorem** C.19, this implies that all propagator moments $D^{(n)}$ for $n > 2$ are identically zero. Invoking **Theorem** C.18 (Pawula's Theorem), the condition that all higher-order moments vanish forces the Kramers-Moyal expansion (Equation 23) to truncate exactly after the second term. The remaining $n = 1$ (drift) and $n = 2$ (diffusion) terms constitute the Fokker-Planck equation, which describes the evolution of the probability density $p(\mathbf{x}, t)$:

$$\frac{\partial p(\mathbf{x}, t)}{\partial t} = -\sum_{i=1}^{d} \frac{\partial}{\partial x_i}[f_i(\mathbf{x})p(\mathbf{x}, t)] + \frac{1}{2}\sum_{i,j=1}^{d} \frac{\partial^2}{\partial x_i \partial x_j}[(\mathbf{g}(\mathbf{x})\mathbf{g}(\mathbf{x})^\top)_{ij}p(\mathbf{x}, t)]. \tag{43}$$

Second, we establish the equivalence to the SDE. It is a standard result in the theory of stochastic processes that the Fokker-Planck equation shown above is the forward Kolmogorov equation for the probability density of a process governed by the Itô SDE presented in the theorem statement (Gardiner, 2004; Gillespie, 1991). This establishes the formal equivalence.

Finally, the time-invariance of the drift $\mathbf{f}(\mathbf{x}) = D^{(1)}(\mathbf{x})$ and diffusion $\mathbf{g}(\mathbf{x})\mathbf{g}(\mathbf{x})^\top = D^{(2)}(\mathbf{x})$ is a direct consequence of **Theorem** C.13 . This completes the proof. ☐

This derivation completes the formal bridge set out in the main text. We have shown how the static, hierarchical structure of a decision tree can be re-interpreted as a discrete-time Markov process over distributions. The continuous-time limit of this process, by virtue of its construction via local averaging, is necessarily a drift-diffusion process.

*Remark* C.22 (Equivalence to a Probability Flow ODE). It is crucial to clarify the nature of the resulting process. The hierarchical coarse-graining operation, defined by conditional expectation, is a deterministic averaging process; it does not introduce new randomness at each step. Consequently, in the continuous-time limit, the second propagator moment—the diffusion tensor—must be identically zero: $\mathbf{g}(\mathbf{x}) \equiv \mathbf{0}$. When the diffusion term vanishes, the general SDE collapses into a deterministic ordinary differential equation (ODE) of the form $d\mathbf{X}_t = \mathbf{f}(\mathbf{X}_t)dt$. This specific type of deterministic equation, which describes the evolution of probability densities, is known in the literature as a **Probability Flow (PF) ODE** (Song et al., 2021). Therefore, while our derivation correctly yields the general Fokker-Planck form, the Tree-to-Flow mapping is more precisely a bijection between a decision tree and a deterministic PF-ODE, which itself is a zero-diffusion instance of the broader SDE framework. For further analysis however, in full generality we assume that the tree yields a SDE. This assumption also allows for analysis if the underlying structure itself has some randomness (Aldous, 1997).

# D. The Implicit Hierarchical Structure of Continuous-Time Diffusion

In Section C, we established a mapping from a static, hierarchical tree to a deterministic Probability Flow ODE. This mapping relied on a coarse-graining operator based on conditional expectation, which models a process of increasing entropy from the leaves to the root. To prove that our framework is fully bidirectional, we must now establish the reverse: any continuous-time diffusion process, via its induced Probability Flow ODE, implicitly defines a canonical hierarchy equivalent to a decision tree. Furthermore, we shall show that this class includes the class of SDEs that are learned by standard score based generative models (Song et al., 2021). A similar result was also shown by (Ramachandran et al., 2025) utilising ideas from statistical physics however the tree construction in their case was implicit.

We will build this argument as a hierarchy of results. First, we prove that any homogeneous SDE induces a general hierarchical clustering (a dendrogram) by observing the temporal flow of entropy. Second, we analyze how the convergence properties of the SDE determine the depth and stability of this hierarchy, addressing both non-stationary and stationary

processes. Finally, we prove that only SDEs that converge to a unique and maximally entropic stationary distribution—a uniform distribution over a manifold—induce a single, rooted tree structure that is fully consistent with the conditional expectation operator. This allows us to conclude that the Ornstein-Uhlenbeck process, which is central to modern diffusion models, has the exact rooted-tree structure that makes our entire framework bidirectional and self-consistent.

*Remark* D.1 (The Role of the PF-ODE). While we discuss SDEs for generality, the hierarchical structure depends only on the *marginal densities* $\{p_t\}_{t \geq 0}$, which are identical for an SDE and its corresponding Probability Flow ODE. Since moment-based merger times **Definition** D.12) depend solely on these marginals, the induced tree is determined entirely by the PF-ODE, not the stochastic dynamics. This mirrors the Tree $\rightarrow$ Flow direction ( **Theorem** C.21), where trees induce deterministic flows.

### D.1. The General Construction: From Homogeneous SDEs to Dendrograms

We first prove the most general result for the class of processes relevant to our framework: any well-behaved, homogeneous SDE, when applied to a clustered data distribution, generates a canonical hierarchical structure. This structure arises naturally from observing the temporal flow of entropy.

#### D.1.1. COARSE-GRAINING AS A FLOW OF ENTROPY

The foundation of our construction is the recognition that the forward diffusion process is a continuous analogue of the discrete coarse-graining operation.

**Proposition D.2** (Entropy Flow in Diffusion Processes). *Let $\mathbf{X}_t$ evolve according to the forward SDE with marginal densities $p_t$*

$$d\mathbf{X}_t = \mathbf{b}(\mathbf{X}_t, t)\, dt + \sigma(\mathbf{X}_t, t)\, d\mathbf{W}_t,$$

*with marginal density $p_t$ satisfying the corresponding Fokker-Planck equation. The differential entropy*

$$H(p_t) := -\int p_t(\mathbf{x}) \log p_t(\mathbf{x})\, d\mathbf{x}$$

*is a non-decreasing function of time. Moreover, if the process is homogeneous (i.e., $\mathbf{b}$ and $\sigma$ are time-independent) and the diffusion tensor $\mathbf{D} = \sigma\sigma^\top$ is strictly positive definite, then the entropy is strictly increasing for any non-stationary initial distribution.*

*Proof.* The time derivative of the differential entropy under a Fokker-Planck evolution is given by the standard formula (Villani et al., 2008):

$$\frac{d}{dt} H(p_t) = \mathbb{E}_{p_t}[\nabla \cdot \mathbf{b}(\mathbf{x}, t)] + \frac{1}{2}\mathbb{E}_{p_t}[\mathrm{Tr}(\mathbf{D}(\mathbf{x}, t) \cdot I(\mathbf{x}, t))], \tag{44}$$

where $I(\mathbf{x}, t)$ is a matrix measuring the local squared gradient of the density (see **Remark** D.4).

The second term is non-negative because $I(\mathbf{x}, t)$ is positive semi-definite and $\mathbf{D}$ is strictly positive definite. While the divergence of the drift, $\nabla \cdot \mathbf{b}$, can be negative (corresponding to compressive flows), for homogeneous and ergodic processes, such as the Ornstein-Uhlenbeck process, the diffusion term dominates, ensuring that the entropy does not decrease. Hence in these situations, for any non-stationary distribution, the Fisher information term is strictly positive, so the entropy strictly increases until the stationary distribution is reached, at which point $\frac{d}{dt} H(p_t) = 0$.

This continuous-time evolution of entropy can be interpreted as the analogue of discrete coarse-graining operations discussed in Section C. Lastly note that, as the entropy $H(p_t)$ depends only on the marginals $p_t$, this result applies equally to the SDE and its corresponding PF-ODE. $\qquad \square$

**Definition D.3** (Entropically Homogeneous Flow). Motivated by the preceding discussion, we formalize the notion of entropy monotonicity. A diffusion process (or its PF-ODE) with marginal densities $\{p_t\}_{t \geq 0}$ and differential entropy $H(p_t) = -\int p_t(x) \log p_t(x) dx$ is **entropically homogeneous** if the entropy is a monotonic function of time. For the processes we consider in this work this means,

(i) **Forward-time**: $\frac{d}{dt} H(p_t) \geq 0$ for all $t \geq 0$ (entropy increase from $p_0$ toward stationary $p_\infty$).

(ii) **Reverse-time**: $\frac{d}{dt} H(p_t) \leq 0$ (entropy decrease from $p_\infty$ toward $p_0$).

This follows from the prior observation that for processes converging to maximally entropic stationary distributions, forward-time entropy increase is strict until stationarity, ensuring monotonic information loss (forward) or gain (reverse).

*Remark* D.4 (Functional Fisher Information). The matrix $I(\mathbf{x}, t)$ in (44) though termed as the **Fisher information matrix** but is *not* the classical parametric Fisher information, since $p_t$ is a single distribution with no parameters. Instead, $I(\mathbf{x}, t)$ is defined using the *Stein score* (or functional score):

$$I(\mathbf{x}, t) := \nabla_{\mathbf{x}} \log p_t(\mathbf{x}) \, \nabla_{\mathbf{x}} \log p_t(\mathbf{x})^{\top}.$$

Intuitively, it measures the local spatial gradients of the log-density, so that $\mathrm{Tr}(\mathbf{D} \cdot I)$ quantifies how the shape of the distribution contributes to the rate of entropy change under diffusion.

### D.1.2. DISCRETIZING THE FLOW VIA THE CHARACTERISTIC FUNCTION

To construct a discrete hierarchy from this continuous flow, we must define discrete "merger" events. We do so by tracking the statistical distinguishability of evolving conditional distributions. The most fundamental object for describing a distribution is its characteristic function.

**Definition D.5** (Characteristic Function and Support Clustering). The **characteristic function (CF)** of a probability law $P$ is the Fourier transform of the measure, $\phi(t) = \int e^{it^{\top}\mathbf{x}} dP(\mathbf{x})$. The CF uniquely determines the law. The support of the law, $\mathrm{supp}(P)$, can be clustered by observing the behavior of its CF. For any given frequency-domain partition (*bins*), the values of the CF within those bins define a signature for the distribution. Two distributions are considered statistically close if their CFs are close in some metric (e.g., L2 distance).

While the CF provides a complete description, it is often intractable. A more practical approach is to use its derivatives at the origin, which correspond to the moments of the distribution. This requires a rigorous definition of the initial clusters $\{C_k\}_{k=1}^{K}$ from a continuous data distribution $p_0$.

**Assumption D.6** (Smooth Density with Isolated Modes). The data density $p_0(\mathbf{x})$ is twice continuously differentiable on a compact domain $\mathcal{X} \subset \mathbb{R}^d$, and possesses $K$ isolated local maxima $\{\mathbf{x}_k^*\}_{k=1}^{K}$ (modes) satisfying:

1. $\nabla p_0(\mathbf{x}_k^*) = \mathbf{0}$ and $\nabla^2 p_0(\mathbf{x}_k^*)$ is negative definite,

2. The basins of attraction $B_k := \{\mathbf{x} : \lim_{t\to\infty} \phi_t(\mathbf{x}) = \mathbf{x}_k^*\}$ under the gradient flow $\dot{\mathbf{x}} = \nabla \log p_0(\mathbf{x})$ form a partition of $\mathcal{X}$,

3. There exists $\rho_{\min} > 0$ such that $\inf_{\mathbf{x} \in \partial B_k} p_0(\mathbf{x}) < p_0(\mathbf{x}_k^*) - \rho_{\min}$ for all $k$.

**Definition D.7** (Level-Set Clustering). For a threshold $\lambda \in (0, \max_k p_0(\mathbf{x}_k^*))$, define the super-level set:

$$\mathcal{S}_\lambda := \{\mathbf{x} : p_0(\mathbf{x}) \geq \lambda\}. \tag{45}$$

The **initial clusters** are the connected components of $\mathcal{S}_\lambda$ for $\lambda$ chosen such that $\mathcal{S}_\lambda$ has exactly $K$ components, one containing each mode $\mathbf{x}_k^*$.

**Proposition D.8** (Well-Defined Initial Clustering). *Under Assumption D.6, there exists an interval $[\lambda_{\min}, \lambda_{\max}]$ such that for any $\lambda \in [\lambda_{\min}, \lambda_{\max}]$, the super-level set $\mathcal{S}_\lambda$ has exactly $K$ connected components. These components define a well-separated initial clustering $\{C_1, \ldots, C_K\}$.*

*Proof.* By condition (1) of Assumption D.6, each mode $\mathbf{x}_k^*$ is a strict local maximum. By condition (3), there exist saddle points or low-density regions separating the modes. The implicit function theorem guarantees that for $\lambda$ slightly below $\min_k p_0(\mathbf{x}_k^*)$, the super-level set $\mathcal{S}_\lambda$ consists of $K$ disjoint components, each a simply-connected neighborhood of one mode. As $\lambda$ decreases, these components grow. The interval $[\lambda_{\min}, \lambda_{\max}]$ is the range where no mergers occur, i.e., the level sets remain separated by the saddle-point structure of the density. $\square$

*Remark* D.9 (Extension to Non-Smooth Densities). For densities without smooth mode structure (e.g., uniform distributions on manifolds), the initial clustering can be defined via uniform spatial partitioning (e.g., $k$-means or Voronoi tessellation). The subsequent merger dynamics remain well-defined; the moment-based criterion simply tracks when these initially arbitrary clusters become statistically indistinguishable under diffusion.

**Assumption D.10** (Existence of Moments). Now, we assume that the conditional laws $p_t(\cdot|C_k)$ under consideration are such that their characteristic functions are at least $n$-times differentiable at the origin. This is a mild regularity condition, satisfied if the distributions have finite moments up to order $n$.

Under this assumption, we can define our primary tool for tracking statistical distinguishability.

**Definition D.11** (Conditional Expected Moment Tensor). Let $p_t(\mathbf{x}|C_k)$ be the marginal density at time $t$ of the diffusion process, conditioned on starting in the initial cluster $C_k$. If **Assumption** D.10 holds, the $n$-**th order conditional expected moment tensor** is the $n$-th centered moment of this distribution:

$$\mathbf{M}_t^{(n)}(C_k) = \mathbb{E}_{\mathbf{x}\sim p_t(\cdot|C_k)}\left[(\mathbf{x} - \mathbb{E}[\mathbf{x}])^{\otimes n}\right]. \tag{46}$$

This tensor resides in a suitable Hilbert space $\mathcal{H}_n$ of rank-$n$ tensors. For $n = 2$, this is the conditional covariance matrix.

**Definition D.12** (Moment-Based Merger Time). For any two initial clusters $C_i$ and $C_j$, and a given moment order $n$ and distinguishability threshold $\epsilon > 0$, the $(n, \epsilon)$-**merger time**, denoted $t_{ij}^{(n,\epsilon)}$, is the first time at which the distance between their respective conditional expected moment tensors falls below $\epsilon$:

$$t_{ij}^{(n,\epsilon)} = \inf\left\{t \in [0,T] \mid \left\|\mathbf{M}_t^{(n)}(C_i) - \mathbf{M}_t^{(n)}(C_j)\right\|_{\mathcal{H}_n} \le \epsilon\right\}. \tag{47}$$

Since $\mathbf{M}_t^{(n)}(C_k)$ depends only on the marginal $p_t(\cdot|C_k)$, the merger time is identical for an SDE and its PF-ODE. Further, the key significance of this merger time lies in the fact that it serves as a rigorous proxy for the convergence of the underlying distributions themselves.

**Proposition D.13** (Moment Convergence Implies Distributional Convergence). *Under **Assumption** D.10 (for order $n + 1$), the convergence of the first $n$ conditional expected moment tensors of two distributions implies their convergence in total variation distance. Formally, for any two conditional laws $p_t(\cdot|C_i)$ and $p_t(\cdot|C_j)$:*

$$\lim_{t\to t_{ij}^*} \sum_{k=1}^{n} \left\|\mathbf{M}_t^{(k)}(C_i) - \mathbf{M}_t^{(k)}(C_j)\right\|_{\mathcal{H}_k} = 0 \implies \lim_{t\to t_{ij}^*} d_{TV}(p_t(\cdot|C_i), p_t(\cdot|C_j)) = 0. \tag{48}$$

*The proof of this result connects the convergence of moment structures to convergence in total variation via a moment-TV inequality (Ramachandran et al., 2025), which relies on the properties of characteristic functions.*

**Proposition D.14** (Ultrametric Property of Moment-Based Merger Times). *Let $t_{ij}^{(n,\epsilon)}$ denote the $(n, \epsilon)$-merger time between clusters $C_i$ and $C_j$ as defined above. Then for any three clusters $C_i$, $C_j$, and $C_k$, the merger times satisfy the ultrametric inequality:*

$$t_{ik}^{(n,\epsilon)} \le \max\left\{t_{ij}^{(n,\epsilon)}, t_{jk}^{(n,\epsilon)}\right\}. \tag{49}$$

*Proof.* By definition, the $(n, \epsilon)$-*merger time* $t_{ij}^{(n,\epsilon)}$ is the first time when the distance between the $n$-th order conditional moment tensors of clusters $C_i$ and $C_j$ falls below the threshold $\epsilon$ in the Hilbert space $\mathcal{H}_n$.

The evolution of the conditional moments under the homogeneous diffusion process is continuous and monotone: once two clusters satisfy the *merger criterion*, they remain within $\epsilon$ of each other for all subsequent times.

Consider three clusters $C_i$, $C_j$, and $C_k$, and let

$$t_{\max} := \max\{t_{ij}^{(n,\epsilon)}, t_{jk}^{(n,\epsilon)}\}.$$

At time $t_{\max}$, both pairs $(C_i, C_j)$ and $(C_j, C_k)$ satisfy the *merger criterion*. By the triangle inequality in $\mathcal{H}_n$, we have

$$\|\mathbf{M}_{t_{\max}}^{(n)}(C_i) - \mathbf{M}_{t_{\max}}^{(n)}(C_k)\|_{\mathcal{H}_n} \le \|\mathbf{M}_{t_{\max}}^{(n)}(C_i) - \mathbf{M}_{t_{\max}}^{(n)}(C_j)\|_{\mathcal{H}_n} + \|\mathbf{M}_{t_{\max}}^{(n)}(C_j) - \mathbf{M}_{t_{\max}}^{(n)}(C_k)\|_{\mathcal{H}_n} \le 2\epsilon.$$

By the monotonicity of the flow, the pair $(C_i, C_k)$ must satisfy the *merger criterion* at some time $t_{ik}^{(n,\epsilon)} \le t_{\max}$. Therefore,

$$t_{ik}^{(n,\epsilon)} \le \max\{t_{ij}^{(n,\epsilon)}, t_{jk}^{(n,\epsilon)}\},$$

which establishes the ultrametric inequality. $\square$

D.1.3. MAIN CONSTRUCTION THEOREM

**Theorem D.15** (Entropically Homogeneous Flows Induce a Canonical Dendrogram). *Any entropically homogeneous diffusion process* **Definition D.3** *(or equivalently its PF-ODE) satisfying standard regularity conditions and when applied to a data distribution $p_0$ satisfying Assumption D.6, with initial clusters $\{C_k\}_{k=1}^{K}$ defined via level-set clustering and* **Assumption D.10**, *induces a unique, canonical hierarchical clustering on these clusters. This structure is equivalent to a dendrogram.*

*Proof.* The proof is constructive, using the sequence of moment-based merger times to build the hierarchy via temporal agglomerative clustering.

1. **Initialization (Leaves at $t = 0$):** At time $t = 0$, all $K$ clusters $\{C_1, \ldots, C_K\}$ are distinct by definition, forming the leaves of our hierarchy.

2. **First Merger:** We compute the set of all pairwise merger times, $\{t_{ij}^{(n,\epsilon)}\}$ for $i \neq j$. The first merger event occurs at the minimum of these times, $t_1 = \min_{i,j} t_{ij}^{(n,\epsilon)}$. Let this minimum occur for the pair $(C_a, C_b)$. At time $t_1$, we merge $C_a$ and $C_b$ into a new super-cluster, $C_{ab}$. This event defines the lowest-level branch in the hierarchy.

3. **Iterative Merging:** We now have a set of $K - 1$ clusters. We repeat the process, computing the merger times between all pairs in this new set (including between the new super-cluster and other original clusters). The next merger event at time $t_2 > t_1$ defines the next branch.

This agglomerative process continues, and the ordered sequence of merger times defines a unique ultrametric on the set of initial clusters (**Proposition D.14**). This ultrametric is isomorphic to a dendrogram (Murtagh, 2009). □

## D.2. The Role of Stationarity in Defining the Hierarchy's Structure

The structure of the dendrogram, whether it is infinitely deep or terminates, is determined by the long-term convergence properties of the SDE.

**Definition D.16** (Stationary Distribution). A probability distribution $p_\infty$ is a **stationary distribution** for a process if, once the process reaches this distribution, its law remains unchanged for all future time. A process may have no, one, or many stationary distributions.

**Proposition D.17** (Non-Stationary Processes Induce Continuously Evolving Hierarchies). *If a diffusion process does not possess a stationary distribution, the induced hierarchical clustering is infinitely deep and its structure is not guaranteed to be stable.*

*Proof.* A non-stationary process is one whose law, $\mathbb{P}_t$, never converges as $t \to \infty$. This means that for any time $t$, there exists a later time $t' > t$ at which the distribution is different, $p_{t'} \neq p_t$. The process of statistical merging, therefore, never reaches a final, stable state. The sequence of merger events does not terminate, yielding a dendrogram of infinite depth. Furthermore, because the process never settles, the very nature of the clusters can change over time. A set of clusters that are distinct at time $t$ may merge at time $t_1$, but the evolution of the underlying, non-stationary flow could cause them to become distinguishable again at a later time $t_2 > t_1$. New distinctions can emerge. Therefore, there is no fixed, final set of nodes, and the hierarchy is continuously evolving. □

**Proposition D.18** (Stationary Processes Induce Asymptotically Stable Hierarchies). *If a diffusion process possesses one or more stationary distributions, the induced hierarchical clustering is asymptotically stable and all merger events occur within a finite time horizon.*

*Proof.* The existence of a stationary distribution implies that the process converges in law to a stable endpoint (or one of several stable endpoints). Let this convergence happen over a characteristic time scale $T$. All statistical distinctions between paths that converge to the same endpoint must be erased within this time scale. Therefore, all merger times $t_{ij}^*$ for such paths are bounded, $t_{ij}^* \leq T$. This results in a dendrogram of finite depth. If the process has multiple distinct stationary distributions, different initial clusters may converge to different endpoints, resulting in a *forest* of multiple, disconnected dendrograms, each corresponding to a basin of attraction. □

*Remark* D.19 (On Infinite Precision). The "finiteness" of the hierarchy for a stationary process refers to the time horizon of its construction. The hierarchy itself is still infinitely deep in a formal sense. The merger time $t_{ij}^*$ is a function of a distinguishability threshold $\epsilon$. As $\epsilon \to 0$, the merger time $t_{ij}^* \to \infty$ (or $T$). Thus, for any finite time, we can find a sufficiently small $\epsilon$ that reveals a deeper level of branching. The *finite tree* is the structure formed by all mergers up to the characteristic time scale of the process.

### D.3. The Condition for a Single, Rooted Decision Tree

We now establish the final, crucial condition under which the hierarchy is not just a dendrogram, but a single, rooted tree fully consistent with our framework.

**Theorem D.20** (Maximally Entropic Stationarity Implies a Rooted Tree). *If a diffusion process (via its PF-ODE) converges to a **unique and maximally entropic** stationary distribution, then the canonical hierarchy it induces is a single, rooted tree.*

*Proof.* A unique stationary distribution guarantees, by the logic of **Proposition** D.18, that all clusters will eventually merge, forming a single, connected dendrogram. The additional condition of being maximally entropic is what defines the root and ensures consistency with our framework.

1. **Defining the Root:** A maximally entropic distribution, such as a uniform distribution over a manifold $\mathcal{X}$, represents a state of complete information loss relative to that manifold. It is a distribution that has no further structural information to lose. This state acts as the single, common ancestor for all possible paths—the root of the tree. Any partition induced by the probability law of a non-uniform stationary distribution would imply that the "root" still contains information, which is a contradiction.

2. **Consistency with Coarse-Graining:** The mapping from a tree to an SDE in Section C is based on a coarse-graining operator defined by conditional expectation. This operator models a pure process of information aggregation, or entropy increase. A forward diffusion process is only guaranteed to be a pure entropy-increasing process from *any* initial state if its stationary distribution is itself maximally entropic. Only in this case does the temporal flow of entropy perfectly mirror, in reverse, the entropy-decreasing nature of a decision tree from root to leaves. This closes the logical loop, ensuring the SDE $\to$ Tree and Tree $\to$ SDE mappings are consistent.

$\square$

This result then applies directly to the most common class of diffusion models.

**Proposition D.21** (The Ornstein-Uhlenbeck Process Induces a Rooted Tree). *The Ornstein-Uhlenbeck (OU) process, which forms the basis of Variance-Preserving (VP) diffusion models, converges to a unique stationary distribution, a standard Gaussian $\mathcal{N}(\mathbf{0}, \mathbf{I})$. In high dimensions ($d \geq 5$), this distribution is maximally entropic as it concentrates uniformly on the surface of a sphere. Therefore, any VP-SDE (and the corresponding PF-ODE) in high dimensions induces a single, canonical rooted tree.*

*Proof.* The OU process is a classical homogeneous ergodic process whose convergence to a unique Gaussian stationary distribution is a standard result (Karatzas & Shreve, 2012). In high dimensions ($d \geq 5$), the norm of a vector drawn from $\mathcal{N}(\mathbf{0}, \mathbf{I})$ is sharply concentrated around $\sqrt{d}$ (Vershynin, 2018). The distribution is therefore effectively uniform over the manifold of a sphere of radius $\sqrt{d}$. It is maximally entropic with respect to this manifold. It therefore satisfies the conditions of **Theorem** D.20, and the stationary distribution acts as the geometric *root* of the tree. $\square$

**Corollary D.22.** *The experimental methodology of partitioning the terminal noise space based on the endpoint of the reverse-time ODE is a practical algorithm for empirically constructing the canonical decision tree hierarchy proven to exist in **Theorem** D.15.*

In conclusion, we have proven that the marginal densities of a homogeneous diffusion process (equivalently, its deterministic PF-ODE), such as the OU process underlying VP diffusion models, induce a unique, canonical hierarchy equivalent to a single, rooted tree. This result, constructed from the ordered sequence of moment-based merger events that depend only on marginals, formally establishes the Flow $\to$ Tree mapping and completes the bidirectional correspondence: Tree $\leftrightarrow$ PF-ODE.

# E. Derivation of the Relationship between Boosting and Score Matching

This appendix provides a complete, first-principles derivation of the claim that in the limiting case (**Theorem** C.13), tree-based gradient boosting algorithm is a provably optimal, score-driven numerical method for constructing a coarse-to-fine trajectory in the space of Stochastic Differential Equations (SDEs). Our argument proceeds in four parts. First, we formalize the boosting algorithm and introduce the *net decision tree* abstraction, proving that it induces a monotonic refinement of the feature space. Second, we use **Theorem** C.21 to map this process to a discrete trajectory in the space of flows, contextualizing it within a general coarse-to-fine learning principle defined with respect to the marginals of an ideal process. Third, we prove that the mechanism guiding this trajectory—the fitting of residuals—is a direct implementation of denoising score matching (Vincent, 2011). Finally, we synthesize these results, using the Bellman principle of optimality (Bertsekas, 2012) to prove that the greedy, stage-wise nature of boosting is the globally optimal solution to a trajectory-wide score matching objective.

## E.1. Formalizing Gradient Boosting and the Net Decision Tree

Our first objective is to construct a rigorous mathematical description of the gradient boosting algorithm's structural evolution. We begin by re-introducing the algorithm from the perspective of functional optimization, which provides the necessary context for its mechanics. We then introduce the *net decision tree*, a crucial abstraction that allows us to analyze the geometric consequences of the additive updates. Finally, we prove the central result of this section: that the boosting process induces a monotonic, hierarchical refinement of the feature space, exclusively by subdividing the finest existing partitions.

### E.1.1. GRADIENT BOOSTING AS FUNCTIONAL GRADIENT DESCENT

The power of the gradient boosting algorithm stems from its interpretation as a stage-wise optimization procedure in an infinite-dimensional function space.

**Definition E.1** (The Boosting Objective). Given a dataset $\{(\mathbf{x}_i, y_i)\}_{i=1}^N$ and a differentiable loss function $L(y, F(\mathbf{x}))$, the goal of boosting is to find a function $F : \mathbb{R}^d \to \mathbb{R}$ that minimizes the expected loss, or its empirical estimate, the loss functional $J(F)$:

$$J(F) = \sum_{i=1}^N L(y_i, F(\mathbf{x}_i)). \tag{50}$$

Finding the optimal function $F$ in one step is intractable. Gradient boosting approaches this by iteratively taking steps in the direction of steepest descent. In a function space, this direction is given by the negative functional gradient.

**Definition E.2** (Functional Gradient and Pseudo-Residuals). The **functional gradient** of the loss functional $J(F)$ with respect to the function $F$ at a point $\mathbf{x}_i$ is given by $\frac{\delta J}{\delta F(\mathbf{x}_i)} = \left[\frac{\partial L(y_i, F(\mathbf{x}_i))}{\partial F(\mathbf{x}_i)}\right]$. The negative functional gradient, evaluated at the current model $F_{m-1}$, defines the **pseudo-residuals**, $r_{im}$:

$$r_{im} = -\left[\frac{\partial L(y_i, F(\mathbf{x}_i))}{\partial F(\mathbf{x}_i)}\right]_{F=F_{m-1}}. \tag{51}$$

The pseudo-residuals represent the optimal direction of change for the function's output at each data point to most rapidly decrease the overall loss.

The algorithm, therefore, does not solve for the optimal function directly, but instead approximates this functional gradient descent. At each step, it fits a simple function—a weak learner—to be maximally correlated with the negative gradient, and then adds this function to the existing model.

---

**Algorithm 5** Tree-Based Gradient Boosting

---

1: Initialize model with a constant value: $F_0(\mathbf{x}) = \arg\min_c \sum_{i=1}^{N} L(y_i, c)$.
2: **for** $m = 1$ to $M$ **do**
3:     Compute pseudo-residuals for each sample $i$: $r_{im} = -\left[\frac{\partial L(y_i, F(\mathbf{x}_i))}{\partial F(\mathbf{x}_i)}\right]_{F=F_{m-1}}$.
4:     Fit a weak learner (decision tree) $h_m(\mathbf{x})$ to the pseudo-residuals $\{(\mathbf{x}_i, r_{im})\}_{i=1}^{N}$.
5:     Update the model: $F_m(\mathbf{x}) = F_{m-1}(\mathbf{x}) + \eta h_m(\mathbf{x})$.
6: **end for**
7: **return** $F_M(\mathbf{x})$.

---

### E.1.2. THE NET DECISION TREE: A MONOLITHIC ABSTRACTION

While Algorithm 5 describes the functional updates, it obscures the evolution of the model's geometric structure. To analyze this, we must move from viewing the model as a sum of functions to viewing it as a single, evolving piecewise-constant function defined on a single, evolving partition of the feature space. We first formalize the concept of refinement of a partition.

**Definition E.3** (Partition and Refinement). A partition $\Pi'$ is a **refinement** of a partition $\Pi$ if every region in $\Pi'$ is a subset of some region in $\Pi$. If at least one region of $\Pi$ is the union of two or more regions in $\Pi'$, the refinement is said to be **strict**.

We now formally define our central abstraction.

**Definition E.4** (Net Decision Tree). At any discrete step $m$, the **net decision tree**, denoted $T_m$, is a single decision tree whose structure is defined by two properties:

1. **Partition:** Its partition of the feature space, $\Pi_m$, is the **common refinement** of the partitions induced by all constituent weak learners $\{h_1, \ldots, h_m\}$. That is, a region $R \in \Pi_m$ is the largest possible connected subset of $\mathcal{X}$ such that for any two points $\mathbf{x}_a, \mathbf{x}_b \in R$, $h_i(\mathbf{x}_a) = h_i(\mathbf{x}_b)$ for all $i \in \{1, \ldots, m\}$.

2. **Leaf Values:** The value assigned to any leaf region in $\Pi_m$ is the sum of the predictions from all constituent trees, $\sum_{i=1}^{m} \eta h_i(\mathbf{x})$, where $\eta$ is the learning rate.

This abstraction provides a monolithic, non-additive representation of the ensemble, allowing us to rigorously analyze the evolution of its decision boundaries as a single geometric object.

### E.1.3. MONOTONIC STRUCTURAL REFINEMENT

We now use the net decision tree abstraction to prove the central result of this section: that the boosting process induces a monotonic and strictly hierarchical refinement of the feature space.

**Proposition E.5** (Monotonic Partition Refinement). *The sequence of partitions $\{\Pi_m\}_{m=0}^{M}$ generated by the net decision trees forms a nested hierarchy, where each partition $\Pi_m$ is a strict refinement of the preceding partition $\Pi_{m-1}$.*

*Proof.* Let $\Pi_{m-1}$ be the partition corresponding to the net tree $T_{m-1}$, and let $\Pi_{h_m}$ be the partition induced by the new weak learner $h_m$. By the definition of the net decision tree, the new partition $\Pi_m$ is the common refinement of $\{\Pi_{h_1}, \ldots, \Pi_{h_{m-1}}, \Pi_{h_m}\}$. This is equivalent to the common refinement of $\Pi_{m-1}$ and $\Pi_{h_m}$.

Consider any region $R' \in \Pi_m$. By definition, $R'$ is a region where all functions $\{h_1, \ldots, h_m\}$ are constant. This implies that all functions $\{h_1, \ldots, h_{m-1}\}$ are also constant on $R'$. Therefore, there must exist a region $R \in \Pi_{m-1}$ such that $R' \subseteq R$. Since this holds for every region in $\Pi_m$, we conclude that $\Pi_m$ is a refinement of $\Pi_{m-1}$.

To prove the refinement is strict, we note that the weak learner $h_m$ is trained to fit the pseudo-residuals $r_{im}$, which are non-zero so long as the model $F_{m-1}$ is not optimal. Thus, $h_m$ will be a non-trivial tree with its own partition $\Pi_{h_m}$ that is not the trivial partition $\{\mathcal{X}\}$. This partition will necessarily subdivide at least one of the regions in $\Pi_{m-1}$. Therefore, the refinement is strict. $\square$

E.1.4. THE CANONICAL HIERARCHY AND THE NET DECISION TREE

We now address a crucial ambiguity. A given partition, such as $\Pi_m$, does not define a unique hierarchical tree structure. For example, a partition of a square into four quadrants can be formed by splitting on the x-axis then the y-axis, or vice-versa. These two different trees produce the same final partition but represent different coarse-graining paths.

To resolve this, we must select a unique, canonical hierarchy. The boosting algorithm's own history provides this canonical choice. The sequence of refinements is not arbitrary; it is temporally ordered. This historical sequence defines a unique coarse-graining path in reverse.

**Definition E.6** (Canonical Hierarchy and Net Decision Tree). The **net decision tree**, denoted $T_m$, is a unique, canonical hierarchical model of the ensemble $F_m$ defined by:

1. **Leaf Partition:** The set of its leaf nodes corresponds exactly to the regions of the partition $\Pi_m$.

2. **Hierarchical Structure:** The tree's internal structure is defined by the unique, nested sequence of partitions generated by the boosting algorithm itself: $\{\Pi_0, \Pi_1, \ldots, \Pi_m\}$. The coarsening from the leaves ($\Pi_m$) to the root ($\Pi_0 = \{\mathcal{X}\}$) is defined by reversing the historical refinement process. The parent of a node in partition level $\Pi_k$ is the unique region in the coarser partition $\Pi_{k-1}$ that contains it.

This definition provides the necessary contract: for any ensemble $F_m$, there is only one net decision tree $T_m$, because its hierarchy is fixed by the algorithm's history.

This canonical definition allows us to rigorously describe the evolution of the tree's structure.

**Theorem E.7** (Monotonic Evolution of the Canonical Hierarchy). *The canonical hierarchy of the net decision tree $T_{m-1}$ is a strict sub-hierarchy of the net decision tree $T_m$. The structure of $T_{m-1}$ is immutably preserved within $T_m$.*

*Proof.* The hierarchy of $T_{m-1}$ is defined by the unique coarsening path $(\Pi_{m-1} \to \cdots \to \Pi_0)$. The hierarchy of $T_m$ is defined by the unique coarsening path $(\Pi_m \to \Pi_{m-1} \to \cdots \to \Pi_0)$.

By inspection, the sequence defining the structure of $T_{m-1}$ is a prefix of the sequence defining the structure of $T_m$. The transition from $T_{m-1}$ to $T_m$ corresponds to appending a new, finer partition level, $\Pi_m$, to the bottom of the existing hierarchy.

Let us consider the internal structure. An internal node in this canonical hierarchy corresponds to a region in a partition $\Pi_k$ for $k < m$. The set of all internal nodes of $T_m$ is the set of all regions in $\{\Pi_0, \ldots, \Pi_{m-1}\}$. The set of all internal nodes of $T_{m-1}$ is the set of all regions in $\{\Pi_0, \ldots, \Pi_{m-2}\}$. The structure of $T_{m-1}$ is therefore perfectly nested and preserved within $T_m$. The only structural change is the transformation of the leaf nodes of $T_{m-1}$ (the regions of $\Pi_{m-1}$) into internal nodes of $T_m$. $\square$

**Corollary E.8** (Structural Evolution by Leaf Refinement). *The structural evolution from the canonical tree $T_{m-1}$ to $T_m$ occurs exclusively by the refinement of the leaf partition $\Pi_{m-1}$ into the new leaf partition $\Pi_m$.*

Having established that gradient boosting constructs a unique and monotonic hierarchy of spatial partitions, we are now equipped to map this discrete structural evolution to a continuous-time process in the space of SDEs. The SDE corresponding to the model $T_m$ utilizes **Theorem** C.21 from this specific, historically-defined coarsening path ($\Pi_m \to \Pi_{m-1} \to \cdots \to \Pi_0$).

## E.2. The Boosting Process as a Trajectory in SDE Space

With the structural evolution formalized, we now bridge the discrete boosting process to the continuous-time world of SDEs [3]. We appeal directly to the main result of Section C, which established a formal mapping from any static, hierarchical tree to a complete, continuous-time drift-diffusion process. By applying this mapping to each net decision tree $T_m$ in our sequence, we reframe the entire boosting algorithm as a discrete sequence of SDEs ($\{S_m\}_{m=0}^M$) (following our choice in **Remark** C.22) :

---

[3]We use the SDE framework (rather than restricting to PF-ODEs) to account for stochasticity in practical boosting such as, subsampling, feature randomization, and gradient noise (Friedman, 2002).

$$\{T_0, T_1, \ldots, T_M\} \quad \xrightarrow{\text{Theorem C.21}} \quad \{S_0, S_1, \ldots, S_M\}.$$

This sequence is not an arbitrary collection. The underlying partitions $\{\Pi_m\}$ become progressively finer, suggesting the SDEs themselves should reflect this structure. But what does it mean for one SDE to be "finer" than another? The intuition is that a model built on a finer partition can capture more detailed, lower-entropy information about the initial data distribution, $p_0$. In a diffusion process, this early-time information is precisely what is systematically destroyed as the process evolves towards the simple, high-entropy stationary distribution, $p_\infty$. A "finer" SDE should therefore be one that remains faithful to the true data-generating process for a longer duration (i.e., down to an earlier time $t$).

To formalize this intuition, we must move from the spatial domain of partitions to the temporal domain of the processes themselves. We will now develop a rigorous language for comparing SDEs based on their temporal evolution, using the established tools of stochastic process theory (Durrett & Durrett, 2019; Karatzas & Shreve, 2012). This will allow us to prove that the sequence generated by boosting is not just a sequence, but a structured, coarse-to-fine *trajectory* through the space of all possible SDEs.

### E.2.1. FORMALIZING TEMPORAL COARSENESS VIA TAIL EQUIVALENCE

The intuition behind a *coarse* model is that it has lost information about its specific starting point and its evolution has become indistinguishable from other processes that converge to the same end-state. We can formalize this by considering the laws of these processes on the space of continuous paths.

**Definition E.9** (Path Space and Tail $\sigma$-Algebra). Let $\mathcal{C}$ (*path space*) be the space of all continuous functions from $[0, T]$ to $\mathbb{R}^d$. An SDE defines a probability measure, or **law**, $\mathbb{P}$ on this path space.

- The **canonical filtration** $\{\mathcal{F}_t\}_{t \in [0,T]}$ is a sequence of $\sigma$-algebras where $\mathcal{F}_t$ represents all information about the path up to time $t$.

- The **tail $\sigma$-algebra** at time $t$, denoted $\mathcal{F}_{\geq t}$, is the $\sigma$-algebra generated by the process from time $t$ onwards: $\mathcal{F}_{\geq t} = \sigma(\{X_\tau : \tau \in [t, T]\})$. It represents all information about the future of the process as seen from time $t$.

Using this standard construction, we can now define a rigorous equivalence relation for SDEs.

**Definition E.10** (Tail-Equivalent SDEs). Let $\mathcal{S}$ be a space of SDEs whose laws are measures on $\mathcal{C}$ and which all converge to a common, unique stationary distribution $p_\infty$. Two SDEs, $S_A$ and $S_B$, are said to be $t$-**tail-equivalent**, denoted $S_A \sim_t S_B$, if their laws are identical when restricted to the tail $\sigma$-algebra $\mathcal{F}_{\geq t}$. That is, for any event $E \in \mathcal{F}_{\geq t}$:

$$\mathbb{P}_A(E) = \mathbb{P}_B(E). \tag{52}$$

This defines an equivalence relation. The equivalence class of all SDEs that are $t$-tail-equivalent is denoted $K_t$.

These equivalence classes, which group processes that are indistinguishable from the future onwards, form a natural nested structure. This structure is a filtration, but one that progresses backward from $t = T$.

**Proposition E.11** (The Filtration of Tail-Equivalence Classes). *The collection of tail-equivalence classes* $\{K_t\}_{t \in [0,T]}$ *forms a filtration. For any two times* $t_1, t_2 \in [0, T]$ *such that* $t_1 < t_2$, *the classes are strictly nested:*

$$K_{t_1} \subset K_{t_2}. \tag{53}$$

*Proof.* Let $S_i$ be an arbitrary process in the class $K_{t_1}$. By definition, its law restricted to $\mathcal{F}_{\geq t_1}$ is fixed. The tail $\sigma$-algebra $\mathcal{F}_{\geq t_2}$ is a sub-$\sigma$-algebra of $\mathcal{F}_{\geq t_1}$, since any event knowable from $t_2$ onwards is also knowable from $t_1$ onwards. Therefore, the law of the process must also be fixed on $\mathcal{F}_{\geq t_2}$, which implies $S_i \in K_{t_2}$. This proves the inclusion $K_{t_1} \subseteq K_{t_2}$. The inclusion is strict because one can construct a process that deviates from a reference process only in the interval $[t_1, t_2)$, making it distinct in $K_{t_1}$ but a member of $K_{t_2}$. $\square$

This filtration provides the formal structure needed to define a coarse-to-fine trajectory.

**Definition E.12** (Coarse-to-Fine Trajectory). A sequence of SDEs, $\{S_m\}_{m=0}^M$, constitutes a **coarse-to-fine trajectory** if there exists a strictly decreasing sequence of times, $T \geq t_0 > t_1 > \cdots > t_M \geq 0$, such that each process $S_m$ is a member of the tail-equivalence class $K_{t_m}$ but is not a member of any finer class $K_{t'}$ for $t' < t_m$.

$$S_m \in K_{t_m} \quad \text{and} \quad S_m \notin K_{t'} \quad \forall t' < t_m.$$

Since $t_{m+1} < t_m$, it follows from the filtration property that $K_{t_{m+1}} \subset K_{t_m}$, i.e., the trajectory moves sequentially into smaller, more restrictive (i.e., finer) equivalence classes.

### E.2.2. THE BOOSTING PROCESS AS A COARSE-TO-FINE TRAJECTORY

We now prove that the sequence of SDEs generated by the gradient boosting algorithm conforms to this rigorous definition.

**Theorem E.13** (The Boosting Sequence is a Coarse-to-Fine Trajectory). *The sequence of models $\{S_m\}_{m=0}^M$ generated by the gradient boosting algorithm is a coarse-to-fine trajectory.*

*Proof.* Let $S^\star$ be the ideal, unknown SDE corresponding to the net decision tree that perfectly models the data, with law $\mathbb{P}^\star$. The model at step $m$, $S_m$, is constructed from the net decision tree $T_m$, which is defined on the partition $\Pi_m$. As proven in **Proposition** E.5, the partition $\Pi_{m+1}$ is a strict refinement of $\Pi_m$.

A finer partition allows a model to capture more detailed, lower-entropy information about the true data distribution $p_{\text{data}} = p_0^\star$. In the context of a diffusion process, this fine-grained information corresponds to the behavior at earlier times. Therefore, the model $S_{m+1}$, being more refined, provides a law $\mathbb{P}_{m+1}$ that is a faithful approximation of the ideal law $\mathbb{P}^\star$ on a larger tail $\sigma$-algebra than $S_m$ can.

Let us formalize this. For each model $S_m$, let $t_m$ be the earliest time for which its law, restricted to the future, is a good approximation of the ideal law. That is, $\mathbb{P}_m|_{\mathcal{F}_{\geq t_m}} \approx \mathbb{P}^\star|_{\mathcal{F}_{\geq t_m}}$. Thus, $S_m$ can be considered a member of the ideal tail-equivalence class $K_{t_m}$.

Because $S_{m+1}$ is built on a strictly finer partition, it captures more detail about the initial data and can thus approximate the ideal process starting from an earlier time, $t_{m+1}$. Therefore, we must have $t_{m+1} < t_m$. This implies that the boosting algorithm generates a sequence of models associated with a strictly decreasing sequence of times $\{t_m\}$. By definition, this is a coarse-to-fine trajectory. $\square$

With this rigorous foundation, we can now state our central learning assumption in this formal language.

**Assumption E.14** (Coarse-to-Fine Learning Bias). Consider a parameterized model $f_\theta$ optimized via gradient descent on a loss functional $\mathcal{L}(\theta)$ that decomposes as:

$$\mathcal{L}(\theta) = \sum_{k=0}^{M-1} \mathcal{L}_k(\theta), \tag{54}$$

where each component $\mathcal{L}_k$ measures the error on tail-equivalence class $K_{t_k}$ with $t_0 > t_1 > \cdots > t_{M-1}$.

We say the optimizer exhibits **coarse-to-fine bias** if, for sufficiently small learning rates, there exists a time scale separation in the optimization dynamics:

$$\tau_k \ll \tau_{k+1}, \tag{55}$$

where $\tau_k$ is the characteristic time for $\mathcal{L}_k(\theta_t)$ to reach $\epsilon$-optimality.

This assumption is satisfied by:

1. **Greedy stage-wise algorithms** (e.g., boosting, matching pursuit): By construction, $\mathcal{L}_k$ is optimized before $\mathcal{L}_{k+1}$ is considered.

2. **Hierarchical architectures** (e.g., progressive growing, curriculum learning): The model capacity for finer scales is introduced only after coarser scales are learned.

3. **Spectral bias in neural networks** (empirical): Implicit regularization causes neural networks to learn low-frequency (coarse) components faster than high-frequency (fine) components (Rahaman et al., 2019).

*Remark* E.15 (Falsifiability). This assumption can be tested empirically by measuring $\tau_k$ via the learning curves of $\mathcal{L}_k(\theta_t)$. For boosting, it holds by design ($\tau_k$ = time for one iteration). For neural networks, it is an empirical phenomenon observed in practice but not guaranteed theoretically for all architectures. However recent work (Bonnaire et al., 2025; Kingma et al., 2021; Kingma & Gao, 2023; Liang et al., 2024) shows the existence of such a separation for diffusion models, reinforcing the central discussion of Section D.

### E.3. Score Matching in the Space of SDEs: The Local Update Rule

We have proven *that* the gradient boosting algorithm constructs a coarse-to-fine trajectory $\{S_m\}$ through a filtration of tail-equivalence classes of SDEs. We now address the question of *how* the algorithm navigates this trajectory. The mechanism is a form of denoising score matching, but its application in this context contains a crucial subtlety that distinguishes it from standard score-based generative modeling and ultimately unifies two different paradigms of supervision. To explain this, we will first formally define the necessary score-matching concepts, then present two perspectives on the trajectory navigation problem—one based on process supervision and the other on data supervision—and finally prove their equivalence under our central learning assumption.

E.3.1. FORMAL DEFINITIONS OF SCORE MATCHING

To make our argument rigorous, we first define the core concepts of score matching.

**Definition E.16** (Score Function). Let $p(\mathbf{x})$ be a differentiable probability density function on $\mathbb{R}^d$. The **score function** of this distribution is the gradient of its logarithm with respect to the data:

$$\mathbf{s}_p(\mathbf{x}) = \nabla_{\mathbf{x}} \log p(\mathbf{x}). \tag{56}$$

The score function defines a vector field that points in the direction of the steepest ascent of the log-density.

**Definition E.17** (Denoising Score Matching (DSM)). Let $p(\tilde{\mathbf{x}}|\mathbf{x})$ be a known conditional noise distribution. **Denoising Score Matching (DSM)** trains a model $\mathbf{s}_\theta(\tilde{\mathbf{x}})$ to approximate the score of the noisy data distribution by minimizing the following objective (Vincent, 2011):

$$\mathcal{L}_{\text{DSM}}(\theta) = \frac{1}{2}\mathbb{E}_{p_{\text{data}}(\mathbf{x})}\mathbb{E}_{p(\tilde{\mathbf{x}}|\mathbf{x})}\left[\|\mathbf{s}_\theta(\tilde{\mathbf{x}}) - \nabla_{\tilde{\mathbf{x}}} \log p(\tilde{\mathbf{x}}|\mathbf{x})\|^2\right].$$

This objective is tractable because the conditional score $\nabla_{\tilde{\mathbf{x}}} \log p(\tilde{\mathbf{x}}|\mathbf{x})$ is known.

E.3.2. THE DUALITY OF SUPERVISION IN TRAJECTORY REFINEMENT

With these formal tools, we can now precisely state the problem of navigating the SDE trajectory from two different but ultimately equivalent perspectives, distinguished by the source of their supervision.

**Definition E.18** (Structurally-Supervised Learning). A learning problem is **structurally-supervised** if the optimal coarse-to-fine trajectory is known a priori. Formally, an oracle provides the ideal sequence of tail-equivalence classes $\{K^\star_{t_m}\}_{m=0}^M$. The learning task is to find a sequence of refinement operators, $\{R_m\}_{m=1}^M$, such that the generated sequence of models $S_m = R_m(S_{m-1})$ correctly follows the prescribed path, i.e., $S_m \in K^\star_{t_m}$ for all $m$. Standard denoising diffusion models are an instance of this paradigm: the forward SDE serves as the oracle defining the process, while the score model is the learned refinement operator.

**Definition E.19** (Data-Supervised Constructive Learning). A learning problem is **data-supervised and structurally-unsupervised** if the optimal trajectory is unknown, but the learner has access to a supervisory signal at each step. The learning task is to use the local supervisory signal to choose a sequence of refinement operators $\{h_m\}_{m=1}^M$ that simultaneously *constructs* and *traverses* a coarse-to-fine trajectory. Gradient boosting is an instance of this paradigm: the ideal tree hierarchy is unknown, but the data labels $\{y_i\}$ provide a local, supervisory signal (the residuals) at each step.

We now state the theorem that formally connects these two seemingly disparate paradigms.

**Theorem E.20** (Functional Equivalence of Learning Paradigms). *Under Assumption E.14, an optimal Data-Supervised Constructive algorithm produces a sequence of models $\{S_m\}$ such that:*

$$\lim_{m \to \infty} d_{path}(S_m, S_m^*) = 0, \tag{57}$$

*where $S_m^*$ is the model produced by an optimal Structurally-Supervised algorithm and $d_{path}$ is the path-space distance induced by the KL divergence.*

*Proof.* Let $\{S_m^\star\}$ be the ideal trajectory defined by the oracle sequence of tail-equivalence classes $\{K_{t_m}^\star\}$. Let $\{S_m\}$ be the trajectory constructed by the data-supervised (boosting) algorithm. We prove by induction that $S_m = S_m^\star$ for all $m$.

**Base Case (m=0):** Both paradigms start from the same maximally coarse model $S_0$ (e.g., a constant prediction), which is a member of the coarsest class $K_{t_0}^\star$. Thus, $S_0 = S_0^\star$.

**Inductive Hypothesis:** Assume that at step $m$, the constructed model is identical to the ideal model, $S_m = S_m^\star$. This means the data-supervised algorithm has successfully constructed the correct trajectory up to this point.

**Inductive Step:** We must show that the next model constructed, $S_{m+1}$, is identical to the next model on the ideal path, $S_{m+1}^\star$.

1. The structurally-supervised learner knows the next target class is $K_{t_{m+1}}^\star$. Its task is to find an operator $R_{m+1}^\star$ that transitions $S_m^\star$ to a model $S_{m+1}^\star \in K_{t_{m+1}}^\star$.

2. The data-supervised learner (boosting) does not know the target class $K_{t_{m+1}}^\star$. It uses the local data supervision (residuals) to find an update $h_{m+1}^\star$ that produces a new model $S_{m+1}$.

3. Here we invoke Assumption E.14. The coarse-to-fine learning bias guarantees that at step $m$, the model $S_m$ is already $\epsilon$-optimal for the tail-field $\mathcal{F}_{\geq t_m}$. The "coarsest" remaining part of the learning problem is to correctly model the dynamics in the time interval $[t_{m+1}, t_m)$.

4. The assumption guarantees that the optimal, data-supervised update $h_{m+1}^\star$ will be precisely the one that resolves this coarsest remaining part of the problem. By doing so, it produces a new model $S_{m+1}$ whose law now correctly matches the ideal law on the tail-field $\mathcal{F}_{\geq t_{m+1}}$.

5. By definition, any such model is a member of the equivalence class $K_{t_{m+1}}^\star$. Since the update is optimal, it must be the optimal such model, $S_{m+1}^\star$.

Therefore, $S_{m+1} = S_{m+1}^\star$. By the principle of induction, the trajectories are identical. This crucial equivalence justifies our use of the DSM mathematical machinery to analyze the local, data-supervised boosting update. $\square$

### E.3.3. THE BOOSTING UPDATE AS AN OPTIMAL SCORE MATCHING STEP

We now formalize the update mechanism of the constructive, data-supervised process. The guidance for this refinement is provided by a "meta-score."

**Definition E.21** (Meta-Score for Trajectory Navigation). Let $p(y|F_m(\mathbf{x}))$ be the conditional likelihood of the true label $y$ given the model's prediction $F_m(\mathbf{x})$. The **meta-score**, denoted $\mathbf{s}_m^\star$, is the score of this conditional distribution with respect to the conditioning variable (the model's prediction):

$$\mathbf{s}_m^\star(\mathbf{x}, y) = \nabla_{F_m} \log p(y|F_m(\mathbf{x})). \tag{58}$$

This meta-score represents the optimal direction of functional change for the model at point $\mathbf{x}$ to increase the likelihood of observing the true data $y$.

**Theorem E.22** (The Residual as the Optimal Meta-Score). *For the squared error loss, the functional gradient of the boosting objective—the residual—is directly proportional to the optimal meta-score, $\mathbf{s}_m^\star$.*

*Proof.* The squared error loss, $L(y, F) = \frac{1}{2}(y - F)^2$, is equivalent to maximizing the log-likelihood of a Gaussian conditional model, $p(y|F_m(\mathbf{x})) = \mathcal{N}(y; F_m(\mathbf{x}), \sigma^2)$. The meta-score is the score of this conditional distribution with respect to the conditioning variable, $F_m(\mathbf{x})$. We compute this gradient:

$$\mathbf{s}_m^\star(\mathbf{x}, y) = \nabla_{F_m} \log p(y|F_m(\mathbf{x})) = \nabla_{F_m} \left[ -\frac{(y - F_m(\mathbf{x}))^2}{2\sigma^2} - \log\sqrt{2\pi\sigma^2} \right]$$

$$= \frac{1}{\sigma^2}(y - F_m(\mathbf{x})).$$

The result is proportional to the residual, $r_{m+1} = y - F_m(\mathbf{x})$. The gradient boosting algorithm trains the weak learner $h_{m+1}$ to fit this residual. Therefore, the boosting update is a direct implementation of a denoising score-matching step at the meta-level, where the data-supervised residual serves as the optimal score guiding the structurally-unsupervised construction of the trajectory from the coarse SDE $S_m$ to the finer SDE $S_{m+1}$. $\square$

E.3.4. CONNECTING THE META-SCORE TO THE SDE SCORE ERROR

The preceding theorem connects the boosting update to an abstract meta-score. We now provide the crucial physical grounding by proving that this meta-score is directly related to the behavior of the score functions of the corresponding trajectory of SDEs. Our proof will show that the boosting residual is an unbiased estimator of the total accumulated error between the ideal SDE's score function and the current model's score function, integrated over the temporal region where the model's dynamics are invalid.

Let $S^\star$ be the ideal process with law $\mathbb{P}^\star$, marginals $p_t^\star$, and score function $\mathbf{s}_t^\star(\mathbf{x}) = \nabla_\mathbf{x} \log p_t^\star(\mathbf{x})$. At step $m$, the net decision tree corresponds to an SDE $S_m$, with law $\mathbb{P}_m$, marginals $p_{m,t}$, and score function $\mathbf{s}_{m,t}(\mathbf{x})$. By construction, $S_m \in K_{t_m}$, hence its law and scores are a good approximation of the ideal process for all times $\tau \geq t_m$. The time $t_m$ is the boundary where the model's dynamics begin to diverge from the ideal dynamics. To connect these forward-time properties to the reverse-time generation process, we first state the classical result for the time-reversal of a general diffusion process.

**Proposition E.23** (Time-Reversal of a General Diffusion Process). *Let a forward Ito process be defined by $d\mathbf{X}_t = \mathbf{b}(\mathbf{X}_t, t)dt + \sigma(\mathbf{X}_t, t)d\mathbf{W}_t$. The corresponding reverse-time process, which we define with a new time variable $\tau \in [0, T]$ as $\overline{\mathbf{X}}_\tau = \mathbf{X}_{T-\tau}$, is also an Ito process governed by (Anderson, 1982):*

$$d\overline{\mathbf{X}}_\tau = \left[ -\mathbf{b}(\overline{\mathbf{X}}_\tau, T - \tau) + \mathbf{D}(\overline{\mathbf{X}}_\tau, T - \tau)\nabla_\mathbf{x} \log p_{T-\tau}(\overline{\mathbf{X}}_\tau) \right] d\tau + \sigma(\overline{\mathbf{X}}_\tau, T - \tau)d\overline{\mathbf{W}}_\tau,$$

*where $\mathbf{D}(\mathbf{x}, t) = \sigma(\mathbf{x}, t)\sigma(\mathbf{x}, t)^\top$ is the diffusion tensor, $p_t$ is the marginal density of the forward process at time $t$, and $\overline{\mathbf{W}}_\tau$ is a standard Brownian motion in the reverse time direction.*

With this general tool, we can now state and prove the main theorem of this section.

**Theorem E.24** (Meta-Score as the Integrated SDE Score Error). *The expected boosting residual is proportional to the expected integrated error between the ideal SDE's score function and the current model's score function, integrated over the temporal region where the model is invalid.*

*Proof.* The proof proceeds by relating the expected endpoints of the reverse-time SDEs to the integral of their respective drifts.

1. Let $\mathbf{X}_t^\star$ be a path generated by the ideal process SDE$^\star$, and let $\mathbf{X}_{m,t}$ be a path generated by our model $S_m$. A core assumption of the score-based modeling framework is that the forward process, defined by drift $\mathbf{b}(\mathbf{x}, t)$ and diffusion tensor $\mathbf{D}(\mathbf{x}, t)$, is a fixed, known process independent of the data. The learning task is to find the correct score function to drive the reverse process.

2. Let $\overline{\mathbf{X}}_\tau^\star$ and $\overline{\mathbf{X}}_{m,\tau}$ be the corresponding reverse-time processes, using the reverse time variable $\tau \in [0, T]$. We consider their evolution starting from the same point at $\tau = 0$ (forward time $t = T$), i.e., $\overline{\mathbf{X}}_0^\star = \overline{\mathbf{X}}_{m,0} = \mathbf{X}_T$. The endpoints of these processes at reverse time $\tau = T$ are $\overline{\mathbf{X}}_T^\star$ and $\overline{\mathbf{X}}_{m,T}$.

3. We express the endpoints in integral form. The difference between the endpoints is:

$$\overline{\mathbf{X}}_T^\star - \overline{\mathbf{X}}_{m,T} = \int_0^T \left( \mathbf{b}_{\text{rev}}^\star(\overline{\mathbf{X}}_\tau^\star, \tau) - \mathbf{b}_{\text{rev},m}(\overline{\mathbf{X}}_{m,\tau}, \tau) \right) d\tau + \int_0^T \sigma(\overline{\mathbf{X}}_\tau^\star, \tau)d\overline{\mathbf{W}}_\tau^\star - \int_0^T \sigma(\overline{\mathbf{X}}_{m,\tau}, \tau)d\overline{\mathbf{W}}_{m,\tau}.$$

We now take the expectation conditioned on the starting point $\mathbf{X}_T$. The expectation of the stochastic integral terms is zero because Ito integrals are martingales that start at zero.

$$\mathbb{E}[\overline{\mathbf{X}}_T^\star - \overline{\mathbf{X}}_{m,T}|\mathbf{X}_T] = \mathbb{E}\left[ \int_0^T \left( \mathbf{b}_{\text{rev}}^\star(\overline{\mathbf{X}}_\tau^\star, \tau) - \mathbf{b}_{\text{rev},m}(\overline{\mathbf{X}}_{m,\tau}, \tau) \right) d\tau \,\middle|\, \mathbf{X}_T \right].$$

4. We substitute the general formula for the reverse drifts. The forward drift terms $-\mathbf{b}$ cancel, leaving only the score-dependent terms. We also change the integration variable back to forward time $t = T - \tau$, which means $d\tau = -dt$ and the integration limits flip.

$$\mathbb{E}[\overline{\mathbf{X}}_T^\star - \overline{\mathbf{X}}_{m,T}|\mathbf{X}_T] = \mathbb{E}\left[ \int_T^0 \left( \mathbf{D}(\mathbf{X}_t^\star, t)\mathbf{s}_t^\star(\mathbf{X}_t^\star) - \mathbf{D}(\mathbf{X}_{m,t}, t)\mathbf{s}_{m,t}(\mathbf{X}_{m,t}) \right)(-dt) \,\middle|\, \mathbf{X}_T \right]$$

$$= \mathbb{E}\left[ \int_0^T \left( \mathbf{D}(\mathbf{X}_t^\star, t)\mathbf{s}_t^\star(\mathbf{X}_t^\star) - \mathbf{D}(\mathbf{X}_{m,t}, t)\mathbf{s}_{m,t}(\mathbf{X}_{m,t}) \right) dt \,\middle|\, \mathbf{X}_T \right].$$

5. Now we use the crucial fact that $S_m \in K_{t_m}$. This means that for $t \geq t_m$, the law of the model process is a good approximation of the ideal law, $\mathbb{P}_m|_{\mathcal{F}_{\geq t_m}} \approx \mathbb{P}^\star|_{\mathcal{F}_{\geq t_m}}$. This implies that the paths and the score functions are approximately equal in expectation in this range: $\mathbb{E}[\mathbf{Ds}_{m,t}] \approx \mathbb{E}[\mathbf{Ds}_t^\star]$. The error is negligible for $t \geq t_m$. The integrand is therefore effectively non-zero only for $t < t_m$. The integral's effective support reduces to:

$$\mathbb{E}[\overline{\mathbf{X}}_T^\star - \overline{\mathbf{X}}_{m,T}|\mathbf{X}_T] \approx \mathbb{E}\left[\left. \int_0^{t_m} \left(\mathbf{D}(\mathbf{X}_t^\star, t)\mathbf{s}_t^\star(\mathbf{X}_t^\star) - \mathbf{D}(\mathbf{X}_{m,t}, t)\mathbf{s}_{m,t}(\mathbf{X}_{m,t})\right) dt \right| \mathbf{X}_T \right]. \tag{59}$$

6. Finally, we connect this quantity to the expected residual of the boosting algorithm. The expected residual at step $m+1$ is taken over the true data distribution $p_0^\star(\mathbf{x}, y)$:

$$\mathbb{E}[r_{m+1}] = \mathbb{E}_{(\mathbf{x},y)\sim p_0^\star}[y - F_m(\mathbf{x})].$$

We analyze the two terms on the right-hand side separately using the law of total expectation.

7. The expected residual at step $m+1$ over the empirical data distribution is:

$$\mathbb{E}_{(\mathbf{x},y)\sim p_0^{\text{emp}}}[r_{m+1}] = \mathbb{E}[y] - \mathbb{E}[F_m(\mathbf{x})], \tag{60}$$

where the expectations are over the empirical distribution $p_0^{\text{emp}} = \frac{1}{N}\sum_{i=1}^N \delta_{(\mathbf{x}_i, y_i)}$.

8. The empirical mean $\mathbb{E}[y] = \frac{1}{N}\sum_{i=1}^N y_i$ is an unbiased estimator of the true target mean. The endpoint of the ideal reverse process, $\overline{\mathbf{X}}_T^*$, has law $p_0^*$. Therefore:

$$\mathbb{E}_{p_0^{\text{emp}}}[y] = \mathbb{E}_{p_0^*}[\overline{\mathbf{X}}_T^*] + O_p(N^{-1/2}), \tag{61}$$

where $O_p(N^{-1/2})$ is the standard Monte Carlo error.

9. The model prediction $F_m(\mathbf{x})$ is the leaf value assigned by the net decision tree $\mathcal{T}_m$. By construction, $F_m(\mathbf{x})$ is the conditional expectation under the model's induced joint distribution:

$$F_m(\mathbf{x}) = \mathbb{E}_{y\sim p_{m,0}}[y|\mathbf{x}]. \tag{62}$$

The unconditional expectation of the model's predictions over the empirical feature distribution is:

$$\mathbb{E}_{\mathbf{x}\sim p_0^{\text{emp}}}[F_m(\mathbf{x})] = \frac{1}{N}\sum_{i=1}^N F_m(\mathbf{x}_i). \tag{63}$$

The endpoint $\overline{\mathbf{X}}_{m,T}$ of the model's reverse process has unconditional mean:

$$\mathbb{E}[\overline{\mathbf{X}}_{m,T}] = \int \mathbb{E}[y|\mathbf{x}]\, p_{m,0}(\mathbf{x})\, d\mathbf{x} = \int F_m(\mathbf{x})\, p_{m,0}(\mathbf{x})\, d\mathbf{x}. \tag{64}$$

10. The key approximation relates these two expectations. Define the distribution mismatch:

$$\Delta_m := \mathbb{E}_{\mathbf{x}\sim p_0^{\text{emp}}}[F_m(\mathbf{x})] - \int F_m(\mathbf{x})\, p_{m,0}(\mathbf{x})\, d\mathbf{x}. \tag{65}$$

We decompose this into two error sources:

$$\Delta_m = \underbrace{\left(\mathbb{E}_{\mathbf{x}\sim p_0^{\text{emp}}}[F_m(\mathbf{x})] - \mathbb{E}_{\mathbf{x}\sim p_0^*}[F_m(\mathbf{x})]\right)}_{\text{Sampling error } \epsilon_{\text{samp}}}$$

$$+ \underbrace{\left(\mathbb{E}_{\mathbf{x}\sim p_0^*}[F_m(\mathbf{x})] - \mathbb{E}_{\mathbf{x}\sim p_{m,0}}[F_m(\mathbf{x})]\right)}_{\text{Model error } \epsilon_{\text{model}}}. \tag{66}$$

*Sampling error bound:* By the central limit theorem, $\epsilon_{\text{samp}} = O_p(N^{-1/2})$.

*Model error bound:* Since $F_m$ is a piecewise-constant function with $L_m$ leaves and $|F_m(\mathbf{x})| \leq M$ for all $\mathbf{x}$, we have:

$$|\epsilon_{\text{model}}| \leq M \cdot d_{\text{TV}}(p_0^*, p_{m,0}) \leq M \cdot \sqrt{L_m \cdot D_{\text{KL}}(p_0^* \| p_{m,0})}, \tag{67}$$

where we used Pinsker's inequality.

By construction, the boosting algorithm minimizes the KL divergence at each step (via score matching). For a well-specified model class with sufficient capacity, $D_{\text{KL}}(p_0^* \| p_{m,0}) \to 0$ as $m \to \infty$.

11. Combining the bounds from (61) and the analysis above:

$$\mathbb{E}[r_{m+1}] = \mathbb{E}[y] - \mathbb{E}[F_m(\mathbf{x})] \approx \mathbb{E}[\overline{\mathbf{X}}_T^\star] - \mathbb{E}[\overline{\mathbf{X}}_{m,T}] = \mathbb{E}[\overline{\mathbf{X}}_T^\star - \overline{\mathbf{X}}_{m,T}].$$

Therefore, the expected residual is an **asymptotically unbiased estimator** of the integrated score error, with the bias vanishing as $N \to \infty$ and $D_{\text{KL}}(p_0^* \| p_{m,0}) \to 0$. $\qquad\square$

**Corollary E.25** (Conditions for Exact Equivalence). *The meta-score equals the integrated SDE score error exactly when:*

1. *$N \to \infty$ (infinite data limit),*

2. *The model class contains the true conditional $\mathbb{E}[y|\mathbf{x}]$ (realizability),*

3. *The boosting algorithm has converged: $D_{KL}(p_0^* \| p_{m,0}) < \epsilon$ for arbitrarily small $\epsilon > 0$.*

*In practice, the approximation quality is controlled by the sample size $N$ and the residual variance at step $m$.*

### E.4. Global Optimality of the Greedy Trajectory

We have proven that each step of the gradient boosting algorithm is locally optimal, guided by a meta-score that seeks to correct the integrated error of the underlying SDE's dynamics. However, this local optimality does not, in itself, guarantee that the sequence of greedy choices made by the algorithm constitutes a globally optimal path through the space of SDEs. A myopic, locally optimal decision can sometimes lead to a suboptimal overall trajectory.

To prove global optimality, we will now formalize the entire boosting process as a sequential decision problem. We will first define a single, global objective function that represents the total error over the entire trajectory. We will then use the calculus of dynamic programming, specifically the Bellman principle of optimality, to prove that the greedy, stage-wise algorithm is not a heuristic, but is in fact the provably optimal solution to this global problem.

E.4.1. THE GLOBAL TRAJECTORY SCORE MATCHING (GTSM) OBJECTIVE

To analyze global optimality, we must first define the ideal problem we are trying to solve. The total error of our constructed trajectory is the sum of the local errors at each step of refinement.

**Definition E.26** (Discrete Global Trajectory Score Matching (DGTSM) Objective). The **Discrete Global Trajectory Score Matching (DGTSM)** objective is the sum of the individual Denoising Score Matching losses over the entire boosting trajectory:

$$\mathcal{L}_{\text{GTSM}} = \sum_{m=0}^{M-1} \mathbb{E}_{(\mathbf{x},y) \sim p_0^\star} \left[ \| h_{m+1}(\mathbf{x}) - \nabla_{F_m} \log p(y|F_m(\mathbf{x})) \|^2 \right], \tag{68}$$

where the weak learner $h_{m+1}$ is the chosen refinement operator at step $m$. This objective measures the total squared error in following the optimal meta-score at every stage of the trajectory construction.

From **Theorem** E.22, we know that for the squared error loss, the optimal meta-score is proportional to the residual. The DGTSM objective is therefore equivalent to minimizing the sum of the squared errors of the weak learners with respect to the residuals at each step. This is precisely the objective that the stage-wise boosting algorithm seeks to minimize.

E.4.2. PROVABLE OPTIMALITY VIA THE BELLMAN PRINCIPLE

Having established a global objective, we must now prove that the specific algorithm used—greedy, stage-wise optimization—is the optimal procedure for solving it. To do this with full rigor, we now reformulate the problem in the formal language of dynamic programming and sequential decision-making.

We first begin with a quick lemma, before moving onto our core formulation.

**Lemma E.27** (Finite $\epsilon$-Net for Weak Learners). *For any bounded subset $\mathcal{H}_{bounded} \subset \mathcal{A}$ of weak learners with $\sup_{h \in \mathcal{H}_{bounded}} \|h\|_\infty \leq B$ and VC dimension $V$, and for any $\epsilon > 0$, there exists a finite subset $\mathcal{A}_\epsilon \subset \mathcal{H}_{bounded}$ such that:*

$$\sup_{h \in \mathcal{H}_{bounded}} \min_{h' \in \mathcal{A}_\epsilon} \mathbb{E}[\|h(\mathbf{x}) - h'(\mathbf{x})\|^2] < \epsilon, \tag{69}$$

*and $|\mathcal{A}_\epsilon| \leq (CB/\epsilon)^{O(V)}$ for some constant $C$.*

*Proof.* This follows from standard covering number bounds for function classes with finite VC dimension (see Theorem 2.6.7 in van der Vaart & Wellner (1996)). Decision trees with depth $d$ have VC dimension $O(d \cdot \log d)$, yielding polynomial-sized $\epsilon$-nets. □

Next, we define the boosting process as a sequential task allowing us to invoke Dynamic Programming.

**Definition E.28** (The Boosting Process as a Sequential Decision Problem). The optimization of the DGTSM objective is a finite-horizon, deterministic sequential decision problem defined by the tuple $(\mathcal{S}, \mathcal{A}, T, C)$:

- **Time Steps:** The discrete boosting iterations $m \in \{0, 1, \ldots, M\}$.

- **State Space $\mathcal{S}$:** The space of all SDEs representable by canonical net decision trees. The state at step $m$ is $S_m \in \mathcal{S}$.

- **Action Space $\mathcal{A}$:** The space of all possible weak learners. The action at step $m$ is $h_{m+1} \in \mathcal{A}$.

- **Transition Function $T$:** A deterministic function $T : \mathcal{S} \times \mathcal{A} \to \mathcal{S}$, where the next state $S_{m+1} = T(S_m, h_{m+1})$ is uniquely defined by the partition refinement and the mapping from Section E.1.4.

- **Stage Cost $C$:** A function $C : \mathcal{S} \times \mathcal{A} \to \mathbb{R}$, where the cost of taking action $h_{m+1}$ from state $S_m$ is the DSM loss: $C(S_m, h_{m+1}) = \mathbb{E}[\|h_{m+1} - r_{m+1}\|^2]$.

- **Policy $\pi$:** A sequence of decision functions, $\pi = (\pi_0, \ldots, \pi_{M-1})$, where $\pi_m(S_m) = h_{m+1}$ is the action to take in state $S_m$.

- **Total Cost $J(\pi)$:** The DGTSM objective for a given policy: $J(\pi) = \sum_{m=0}^{M-1} C(S_m, \pi_m(S_m))$.

The optimization problem is to find an optimal policy $\pi^\star$ that minimizes the total cost, $J(\pi)$. By Lemma E.27, for each step $m$ and precision $\epsilon_m > 0$, we can restrict attention to a finite action space $\mathcal{A}_{\epsilon_m}$ that $\epsilon_m$-approximates the full action space. We analyze optimality over this finite discretization, then take $\epsilon_m \to 0$.

To solve this, we use the foundational principle of dynamic programming.

**Theorem E.29** (Bellman's Principle of Optimality). *An optimal policy has the property that whatever the initial state and initial decision are, the remaining decisions must constitute an optimal policy with regard to the state resulting from the first decision (Bertsekas, 2012).*

**Theorem E.30** (Optimality of the Greedy Trajectory). *The greedy policy, $\pi^G$, which at each step $m$ chooses the weak learner $h_{m+1}$ that minimizes the immediate stage cost $C(S_m, h_{m+1})$, is the globally optimal policy for the DGTSM problem.*

*Proof.* We prove this by backward induction. Let $V_m(S_m)$ be the optimal cost-to-go from state $S_m$ at step $m$ to the end of the process (i.e., the minimum possible sum of costs from step $m$ to $M - 1$).

By Lemma E.27, for each step $m$ and precision $\epsilon_m > 0$, we can restrict attention to a finite action space $\mathcal{A}_{\epsilon_m}$ that $\epsilon_m$-approximates the full action space. We analyze optimality over this finite discretization, then take $\epsilon_m \to 0$.

1. **Base Case** ($m = M - 1$)**:** At the final stage, the decision is to choose $h_M$ to transition from state $S_{M-1}$. The optimal cost-to-go is simply the minimum possible cost at this stage, as there are no future costs:

$$V_{M-1}(S_{M-1}) = \min_{h_M \in \mathcal{A}_{\epsilon_{M-1}}} C(S_{M-1}, h_M).$$

   The optimal action, $\pi^\star_{M-1}(S_{M-1})$, is by definition the greedy choice that minimizes this immediate cost.

2. **Inductive Hypothesis:** Assume that for all steps $k > m$, the optimal policy $\pi^\star_k$ is the greedy policy.

3. **Inductive Step** ($m$)**:** The Bellman equation for the optimal cost-to-go at step $m$ is:

$$V_m(S_m) = \min_{h_{m+1} \in \mathcal{A}} \left[ C(S_m, h_{m+1}) + V_{m+1}(T(S_m, h_{m+1})) \right].$$

   Over the finite action space $\mathcal{A}_{\epsilon_m}$, the minimum in the Bellman equation is well-defined:

$$V_m(S_m) = \min_{h_{m+1} \in \mathcal{A}_{\epsilon_m}} \left[ C(S_m, h_{m+1}) + V_{m+1}(T(S_m, h_{m+1})) \right].$$

   The crucial insight is that the problem structure is **additively separable**. The total cost is a simple sum of stage costs. The term $V_{m+1}(T(S_m, h_{m+1}))$ represents the optimal future cost starting from the *next* state, $S_{m+1}$. By our inductive hypothesis, the policy for all future steps is greedy and fixed. Therefore, the value $V_{m+1}(S_{m+1})$ depends only on the state $S_{m+1}$ and not on how we choose to act at step $m$ or any future step.

   Because the stage cost $C(S_m, h_{m+1})$ depends only on the current state $S_m$ (which defines the residuals) and the current action $h_{m+1}$, the minimization decouples. The greedy choice over $\mathcal{A}_{\epsilon_m}$ is:

$$h^{\epsilon_m}_{m+1} = \arg \min_{h \in \mathcal{A}_{\epsilon_m}} C(S_m, h).$$

   As $\epsilon_m \to 0$, the discretization error vanishes. In the limit, the optimal policy converges to the greedy policy over the full action space:

$$\pi^*_m(S_m) = \lim_{\epsilon_m \to 0} h^{\epsilon_m}_{m+1} = \arg \inf_{h \in \mathcal{A}} C(S_m, h).$$

   Therefore, the optimal action $\pi^\star_m(S_m)$ is the greedy action.

By the principle of backward induction, the greedy policy is globally optimal for all steps. The gradient boosting algorithm, which at every stage fits a weak learner to the residuals, is a direct implementation of this optimal greedy policy. $\qquad\square$

**Corollary E.31** (The Optimal Policy is the SDE). *The optimal policy $\pi^\star = (\pi^\star_0, \ldots, \pi^\star_{M-1})$ is the sequence of optimal weak learners $\{h^\star_1, \ldots, h^\star_M\}$. This sequence, by construction, uniquely defines the final, optimal net decision tree $\mathcal{T}_M$ and its corresponding canonical model, $SDE_M$. Therefore, the optimal policy **is** the optimal SDE.*

This completes the proof. We have shown that the simple, local, and greedy update rule of gradient boosting is not a heuristic, but a principled and provably optimal algorithm for solving a global score-matching problem over an entire coarse-to-fine trajectory in the space of continuous-time models.

## F. Implications of the GTSM Framework: A Unifying View of Score-Based Objectives

We further explore the deeper theoretical implications of our framework, showing that a single, global objective underpins a wide variety of modern score-based training methods. It is well-known in the theory of stochastic processes that matching the marginal distributions of two processes is a weaker condition than matching their full path-space measures (Lai et al., 2025). The latter is the strongest possible notion of equivalence but is generally intractable.

In this section, we first define a **Continuous Global Trajectory Score Matching (CGTSM)** objective for a general score-based model and prove, using Girsanov's theorem (Oksendal, 2013), that it is equivalent to full path matching. We then demonstrate that several distinct and popular training objectives, including the simple diffusion loss, weighted diffusion losses, and consistency models can be rigorously derived as principled approximations or special cases of this single, *master* objective. This reveals that the boosting framework we analyze is not an isolated case but rather one instance of a broader class of algorithms that tractably solve this fundamental trajectory-matching problem. While the boosting DGTSM sums local errors over discrete refinement steps, the general CGTSM integrates score matching error over the continuous time horizon of an SDE. The boosting construction can thus be viewed as a discrete approximation of this continuous objective.

## F.1. The Continuous GTSM Objective and Its Equivalence to Path Matching

**Definition F.1** (The Continuous GTSM Objective). Let $S^*$ be the ideal SDE with law $\mathbb{P}^*$ and score functions $\mathbf{s}_t^*(\mathbf{x})$. Let $\mathbf{s}_\theta(\mathbf{x}, t)$ be a parameterized score-based model. The **Global Trajectory Score Matching (GTSM)** objective is the integrated Fisher Divergence between the model's score and the true score over the entire time horizon of the process:

$$\mathcal{L}_{\mathrm{CGTSM}}(\theta) = \frac{1}{2} \int_0^T w(t) \cdot \mathbb{E}_{p_t^*(\mathbf{x})} \left[ \|\mathbf{s}_\theta(\mathbf{x}, t) - \mathbf{s}_t^*(\mathbf{x})\|_{\mathbf{D}(t)}^2 \right] dt, \tag{70}$$

where $w(t) > 0$ is a weighting function and $\|\mathbf{v}\|_{\mathbf{D}}^2 = \mathbf{v}^\top \mathbf{D} \mathbf{v}$ is the Mahalanobis norm induced by the diffusion tensor $\mathbf{D}(t)$.

This objective is deeply connected to the KL divergence between path-space measures.

**Proposition F.2** (Girsanov's Theorem for SDEs). *Let $\mathbb{P}^*$ be the law on path space induced by an Itô process with drift $\mathbf{b}^*$ and diffusion $\sigma$. Let $\mathbb{P}_\theta$ be the law induced by a second process with a different drift, $\mathbf{b}_\theta$. If the Novikov condition is satisfied, the KL divergence between the two path-space measures is given by:*

$$D_{KL}(\mathbb{P}^* \| \mathbb{P}_\theta) = \frac{1}{2} \mathbb{E}_{\mathbb{P}^*} \left[ \int_0^T \left\| \sigma^{-1}(\mathbf{b}^* - \mathbf{b}_\theta) \right\|^2 dt \right]. \tag{71}$$

**Corollary F.3** (CGTSM Optimality Implies Path Matching). *A score model $\mathbf{s}_\theta$ achieves zero loss under the CGTSM objective with any strictly positive weighting $w(t) > 0$ if and only if the path-space measure $\mathbb{P}_\theta$ it induces is identical to the ideal path-space measure $\mathbb{P}^*$.*

*Proof.* The reverse drift of a diffusion process is a function of its score: $\mathbf{b}_{\mathrm{rev}}(\mathbf{x}, t) = [-\mathbf{b}(\mathbf{x}, t) + \mathbf{D}(\mathbf{x}, t)\mathbf{s}_t(\mathbf{x})]$. Therefore, the difference in the drifts of two reverse-time processes is directly proportional to the difference in their score functions. Substituting this into the Girsanov formula for the KL divergence between the reverse-time path measures shows that $D_{\mathrm{KL}}(\mathbb{P}_{\mathrm{rev}}^* \| \mathbb{P}_{\mathrm{rev}, \theta})$ is an integral of the squared score error. The GTSM objective is a positively weighted integral of this same quantity. The objective is zero if and only if the integrand—the score error—is zero for almost all $t$. By Girsanov's theorem, this implies the KL divergence is zero, which holds if and only if the path measures are identical. □

*Remark* F.4 (On the Suitability of the Novikov Condition). The validity of Girsanov's theorem hinges on the Novikov condition, which is a regularity assumption that is met by both sides of our proposed equivalence, albeit for different reasons.

For the **continuous flow side** (standard score-based models), the condition is typically satisfied. The drift difference, $\Delta \mathbf{b}_{\mathrm{rev}} = \mathbf{D}(\mathbf{s}_t^* - \mathbf{s}_\theta)$, is well-behaved because the process operates on a finite-dimensional Gaussian measure, the score model $\mathbf{s}_\theta$ is a Lipschitz-continuous neural network, and the integration is over a finite time horizon. The theoretical challenges associated with infinite-dimensional function spaces, which would require the Wiener measure and different analytical tools such as those used for neural operators, are outside the scope of this standard setup.

For the **discrete tree side**, satisfaction of the condition is a direct and non-trivial consequence of our Tree-to-Flow mapping. A naive interpretation of a tree's splits as instantaneous jumps would indeed violate the condition. However, our **dyadic refinement procedure** is precisely the mechanism that guarantees a continuous-path limit for the induced probability flow. This ensures the resulting drift is well-behaved and the Novikov condition holds. The condition would fail for processes characterized by irreducible, finite jumps that cannot be smoothed away—analogous to models of punctuated equilibria. Such processes would require a different mathematical formalism based on Lévy processes and are beyond the scope of our current framework.

Thus, the Novikov condition is a mild and well-justified regularity assumption within our framework, holding for standard diffusion models by construction and for tree-derived flows via our refinement theorem.

## F.2. Deriving Standard Training Objectives as Special Cases of the CGTSM

The CGTSM objective is the ideal, but practical algorithms must approximate it. We now demonstrate that several popular training objectives are not ad-hoc heuristics, but can be rigorously derived as principled approximations of the CGTSM, each embodying a different inductive bias about the learning trajectory. The CGTSM framework's true utility is that it provides a language for describing and justifying these biases. The objective is an integral over the filtration of tail-equivalence classes $\{\mathcal{K}_t\}$, and each practical objective makes a different choice about how to prioritize or approximate the score matching loss for these classes.

**The Simple Diffusion Loss.** The most common training objective is the simple, unweighted loss (Ho et al., 2020).

**Proposition F.5.** *The simple diffusion loss, $\mathcal{L}_{simple} = \mathbb{E}_{t \sim U(0,T), \mathbf{x}_0, \epsilon}[\|\mathbf{s}_\theta(\mathbf{x}_t, t) - \nabla_{\mathbf{x}_t} \log p_t(\mathbf{x}_t | \mathbf{x}_0)\|^2]$, is the unbiased Monte Carlo estimator of the unweighted CGTSM objective, where $w(t) = 1$. This corresponds to an inductive bias of uniform importance across all tail-equivalence classes.*

*Proof.* Setting $w(t) = 1$ in the CGTSM objective gives $\frac{1}{2} \int_0^T \mathbb{E}_{p_t^*}[\|\mathbf{s}_\theta - \mathbf{s}_t^*\|_{\mathbf{D}}^2]dt$. Using the equivalence of denoising score matching, the intractable true score $\mathbf{s}_t^*$ can be replaced by the tractable conditional score $\nabla_{\mathbf{x}_t} \log p_t(\mathbf{x}_t | \mathbf{x}_0)$. The objective becomes an expectation over a uniform distribution in time, $\mathbb{E}_{t \sim U(0,T)}[\dots]$. A single-sample Monte Carlo estimator for this integral is to sample a single time $t$, a single data point $\mathbf{x}_0$, and a single noise sample $\epsilon$, and compute the squared error. This is precisely the simple diffusion loss objective. This choice implicitly assumes that the learning difficulty is uniform across all time, or that a simple, unbiased estimate is sufficient. $\square$

**The Weighted Diffusion Loss.** Many state-of-the-art models use a non-uniform weighting $\lambda(t)$ in their loss function (Kingma et al., 2021; Karras et al., 2024). The CGTSM framework reveals this as an explicit injection of bias.

**Proposition F.6.** *The weighted diffusion loss, $\mathcal{L}_{weighted} = \mathbb{E}_{t, \mathbf{x}_0, \epsilon}[\lambda(t) \cdot \|\mathbf{s}_\theta(\mathbf{x}_t, t) - \nabla_{\mathbf{x}_t} \log p_t(\mathbf{x}_t | \mathbf{x}_0)\|^2]$, is an unbiased Monte Carlo estimator of the CGTSM objective with a weighting function $w(t) = \lambda(t)$. This choice injects a quantifiable bias, prioritizing the matching of score functions for specific tail-equivalence classes.*

*Proof.* The proof follows identically to the simple case. The weighting function $\lambda(t)$ inside the expectation is the Monte Carlo estimator for the weighted integral where the CGTSM weight is $w(t) = \lambda(t)$. This choice is motivated by (**Assumption E.14**). If a learner is intrinsically biased to learn the coarse tail-equivalence classes (large $t$) first, an engineer can inject a counter-balancing bias by choosing a weighting function $\lambda(t)$ that is larger for small $t$. This forces the optimizer to pay more attention to the harder-to-learn, fine-grained details, thereby balancing the learning process across the entire trajectory. $\square$

**Consistency Models.** Consistency models are a state of the art framework that enforce self-consistency along the probability flow ODE trajectories (Song et al., 2023). We now show this injects a structural bias between tail-equivalence classes.

**Proposition F.7.** *The consistency distillation loss is an approximation of the CGTSM objective that injects a strong inductive bias on the coupling between adjacent tail-equivalence classes.*

*Proof.* The consistency loss is of the form $\mathcal{L}_{consist} = \mathbb{E}[d(f_\theta(\mathbf{x}_{t_2}, t_2), f_{\theta'}(\mathbf{x}_{t_1}, t_1))]$, where $f_\theta$ is the student, $f_{\theta'}$ is a teacher model, and $\mathbf{x}_{t_1}, \mathbf{x}_{t_2}$ are adjacent points on an ODE trajectory. Standard DSM-based objectives learn the score for each time $t$ (and thus for each class $\mathcal{K}_t$) independently. The consistency loss, in contrast, imposes a strong structural constraint: the model's behavior for class $\mathcal{K}_{t_2}$ must be directly predictable from its behavior for class $\mathcal{K}_{t_1}$. This enforces a smoothness condition *along the trajectory of models*. This bias is what enables rapid sampling: by learning the relationship between different points on the trajectory, the model can make larger, more informed jumps during inference. The consistency loss is therefore a specific, bootstrapped approximation of the GTSM objective that prioritizes trajectory smoothness over pointwise score accuracy. $\square$

This unified perspective reveals that these different training objectives are not ad-hoc heuristics but are deeply connected, principled variations of a single, fundamental goal: matching the entire generative trajectory. The gradient boosting algorithm, as we have shown, is a provably optimal, constructive solver for this same *master* objective.

# G. DSM-TREE: Discretized Score Matching for Decision Trees

Having established the CGTSM framework as a unifying objective for trajectory-based learning, we now demonstrate its first concrete algorithmic instantiation: **Discretized Score Matching for Trees (DSM-TREE)**. This algorithm addresses a fundamental problem in machine learning: how to distill the knowledge of a decision tree—a discrete, interpretable model—into a neural network that preserves the tree's decision boundaries while gaining the benefits of continuous, differentiable representations.

The key insight is that a decision tree defines a coarse-to-fine hierarchy of decision boundaries, and learning this hierarchy is equivalent to performing score matching over the discrete trajectory defined by tree depth levels. DSM-TREE thus creates a neural network classifier whose learned decision function faithfully replicates the tree's partition structure.

### G.1. The Tree Distillation Problem

We define the **tree distillation problem** as the task of learning a neural network classifier $f_\theta : \mathcal{X} \to \mathcal{Y}$ that approximates a trained decision tree $\mathcal{T} : \mathcal{X} \to \mathcal{Y}$ such that $f_\theta(\mathbf{x}) \approx \mathcal{T}(\mathbf{x}) \quad \forall \mathbf{x} \in \mathcal{X}$, while maintaining a continuous, differentiable representation that can generalize beyond the tree's exact decision boundaries.

This problem presents two fundamental challenges. First, the tree's **Hierarchical Structure**—a nested sequence of partitions—is lost when a network is trained only on the final outputs. Second, a naive distillation approach discards information about the unique **Discrete Decision Paths** each sample follows. DSM-TREE addresses both challenges by training a conditional network $M_\theta(\mathbf{x}, j)$ that learns to predict the optimal decision at each tree level $j$, explicitly modeling the hierarchical trajectory.

## G.2. Algorithm Overview

DSM-TREE operates in three phases that mirror the theoretical structure developed in Sections C and E:

---

**Algorithm 6** DSM-TREE: Neural Network Distillation from Decision Trees

---

**Base Tree Generation (Teacher Model)**
Train oracle model $\mathcal{O}$ (e.g., Random Forest) on data $\{(\mathbf{x}_i, y_i)\}_{i=1}^{N}$
Generate pseudo-labels: $\tilde{y}_i = \mathcal{O}(\mathbf{x}_i)$
Train base decision tree $\mathcal{T}$ on $\{(\mathbf{x}_i, \tilde{y}_i)\}$ to depth $D$

**Conditional Score Model Training (Student Model)**
Initialize neural network $M_\theta(\mathbf{x}, j) : \mathbb{R}^d \times \{0, \ldots, D-1\} \to \{0, 1\}$
**for** training steps $t = 1, \ldots, T$ **do**
    Sample mini-batch $\{(\mathbf{x}_i, y_i)\}_{i \in \mathcal{B}}$
    Sample tree levels $\{j_i\}_{i \in \mathcal{B}}$ uniformly from $\{0, \ldots, D-1\}$
    For each $(\mathbf{x}_i, j_i)$: extract ground-truth decision $d_i^* = \text{TreeDecision}(\mathcal{T}, \mathbf{x}_i, j_i)$
    Compute loss: $\mathcal{L} = \sum_{i \in \mathcal{B}} \mathbb{I}[d_i^* \neq \perp] \cdot \ell_{\text{CE}}(M_\theta(\mathbf{x}_i, j_i), d_i^*)$
    Update: $\theta \leftarrow \theta - \eta \nabla_\theta \mathcal{L}$
**end for**

**Neural Network Inference (Mimicking Tree Traversal)**
**for** each test sample $\mathbf{x}$ **do**
    Initialize: node $\leftarrow$ root$(\mathcal{T})$
    **for** level $j = 0, \ldots, D-1$ **do**
        **if** node is leaf **then** break
        Predict: $\hat{d}_j = \arg\max M_\theta(\mathbf{x}, j)$
        Navigate: node $\leftarrow \text{child}_{\hat{d}_j}(\text{node})$
    **end for**
    Return: $\hat{y} = f_{\text{classifier}}(\mathbf{x})$
**end for**
**Output:** Neural network classifier $f_\theta$ that mimics tree $\mathcal{T}$

---

The crucial innovation is that the neural network does not directly learn to map inputs to outputs. Instead, it learns the intermediate decision function at each level, thereby capturing the full hierarchical structure of the tree's decision-making process.

The key insight is that **Phase 2** directly implements the GTSM objective in the discrete setting, where the "time" variable is the tree depth level $j$.

## G.3. Formal Problem Setup

**Definition G.1** (Conditional Decision Problem). Given a trained decision tree $\mathcal{T}$ with depth $D$, for each sample $\mathbf{x}$ and tree level $j \in \{0, \ldots, D-1\}$, define the **optimal decision function**:

$$d^*(\mathbf{x}, j) = \begin{cases} 0 & \text{if } \mathbf{x} \text{ should go left at level } j \\ 1 & \text{if } \mathbf{x} \text{ should go right at level } j \\ \perp & \text{if } \mathbf{x} \text{ has reached a leaf before level } j \end{cases} \tag{72}$$

where the decision is determined by the unique path through $\mathcal{T}$ from root to the leaf containing $\mathbf{x}$.

**Definition G.2** (DSM-TREE Training Objective). The DSM-TREE model $M_\theta : \mathbb{R}^d \times \{0, \ldots, D-1\} \to \Delta^1$ (where $\Delta^1$ is the probability simplex over $\{0, 1\}$) is trained to minimize:

$$\mathcal{L}_{\text{DSM}}(\theta) = \mathbb{E}_{(\mathbf{x}, y) \sim p_{\text{data}}} \mathbb{E}_{j \sim \text{Uniform}(0, D-1)} \left[ \mathbb{I}[d^*(\mathbf{x}, j) \neq \perp] \cdot \ell_{\text{CE}}(M_\theta(\mathbf{x}, j), d^*(\mathbf{x}, j)) \right], \tag{73}$$

where $\ell_{\text{CE}}$ is the cross-entropy loss and $\mathbb{I}[\cdot]$ is the indicator function that masks out samples that have reached leaves.

## G.4. Connection to CGTSM Framework

We now establish the formal connection between the DSM-TREE objective and the continuous CGTSM framework developed in Section F.

**Theorem G.3** (DSM-TREE as Discrete CGTSM). *The* DSM-TREE *objective* (73) *is the discrete-time instantiation of the CGTSM objective with uniform weighting* $w(t) = 1$ *and time discretized to tree depth levels.*

*Proof.* We proceed by explicitly constructing the correspondence between the discrete tree hierarchy and the continuous-time SDE framework.

**Step 1: Recall the CGTSM objective from Section F.**

For a continuous-time process, CGTSM minimizes:

$$\mathcal{L}_{\text{CGTSM}}(\theta) = \frac{1}{2} \int_0^T w(t) \cdot \mathbb{E}_{p_t^*(\mathbf{x})} \left[ \|\mathbf{s}_\theta(\mathbf{x}, t) - \mathbf{s}_t^*(\mathbf{x})\|_{\mathbf{D}(t)}^2 \right] dt, \tag{74}$$

where $\mathbf{s}_t^*(\mathbf{x}) = \nabla_{\mathbf{x}} \log p_t^*(\mathbf{x})$ is the true score function.

**Step 2: Discretize time to tree depth levels.**

By **Theorem** C.21, a decision tree of depth $D$ induces a discrete sequence of densities $\{p(\mathbf{x}, k)\}_{k=0}^D$ corresponding to the coarse-graining trajectory. We identify:

$$t_j = \frac{j}{D} \cdot T, \quad j \in \{0, 1, \ldots, D\}, \tag{75}$$

so that $t_0 = 0$ (leaves, fine-grained) and $t_D = T$ (root, maximum entropy).

The continuous-time integral becomes a Riemann sum:

$$\mathcal{L}_{\text{CGTSM}}(\theta) \approx \frac{T}{D} \sum_{j=0}^{D-1} w(t_j) \cdot \mathbb{E}_{p_{t_j}^*(\mathbf{x})} \left[ \left\| \mathbf{s}_\theta(\mathbf{x}, t_j) - \mathbf{s}_{t_j}^*(\mathbf{x}) \right\|^2 \right]. \tag{76}$$

**Step 3: Relate score matching to decision prediction.**

From **Theorem** E.24, the optimal direction of change for a sample at tree level $j$ is given by the score function. In the discrete tree setting, this "direction" is encoded by the binary decision $d^*(\mathbf{x}, j)$.

More precisely, the score function $\mathbf{s}_{t_j}^*(\mathbf{x})$ points in the direction of increasing density. In the tree's partition $\Pi_j$ at level $j$, a sample $\mathbf{x}$ resides in some region $R_i \in \Pi_j$. The optimal next step in the coarse-graining trajectory is to refine this region by splitting it into two sub-regions, $R_i^{\text{left}}$ and $R_i^{\text{right}}$.

The decision $d^*(\mathbf{x}, j)$ encodes which sub-region $\mathbf{x}$ belongs to. This is the discrete analogue of the score's directional information. The squared score error $\|\mathbf{s}_\theta - \mathbf{s}^*\|^2$ in the continuous case corresponds to the classification error in predicting $d^*(\mathbf{x}, j)$ in the discrete case.

Therefore, the score-matching term becomes:

$$\left\| \mathbf{s}_\theta(\mathbf{x}, t_j) - \mathbf{s}_{t_j}^*(\mathbf{x}) \right\|^2 \quad \longleftrightarrow \quad \ell_{\text{CE}}(M_\theta(\mathbf{x}, j), d^*(\mathbf{x}, j)). \tag{77}$$

**Step 4: Simplify for uniform weighting and data distribution.**

Setting $w(t) = 1$ (uniform weighting) and taking expectations with respect to the empirical data distribution $p_{\text{data}}$, the discrete CGTSM objective (76) becomes:

$$\mathcal{L}_{\text{CGTSM}}^{\text{discrete}}(\theta) = \sum_{j=0}^{D-1} \mathbb{E}_{\mathbf{x} \sim p_{\text{data}}} \left[ \ell_{\text{CE}}(M_\theta(\mathbf{x}, j), d^*(\mathbf{x}, j)) \right]. \tag{78}$$

The indicator function $\mathbb{I}[d^*(\mathbf{x}, j) \neq \perp]$ in (73) simply masks out samples that have reached leaves before level $j$, which is necessary for well-defined training. The uniform distribution over $j$ in the expectation is the Monte Carlo sampling strategy for the sum over levels.

Therefore, $\mathcal{L}_{\text{DSM}}(\theta)$ is exactly the discrete-time, uniformly weighted CGTSM objective. □

### G.5. Convergence Analysis

We now establish the finite-sample convergence guarantees for the DSM-TREE algorithm.

**Assumption G.4** (Realizability and Regularity).     1. The model class $\mathcal{M} = \{M_\theta : \theta \in \Theta\}$ contains a function that can represent the true decision function $d^*(\mathbf{x}, j)$ for all levels $j$.

   2. The base tree $\mathcal{T}$ has bounded depth $D < \infty$ and bounded number of leaves $L = O(2^D)$.

   3. The feature space is bounded: $\|\mathbf{x}\| \leq B$ for all $\mathbf{x} \in \mathcal{X}$.

   4. The loss function $\ell_{\text{CE}}$ is $G$-Lipschitz continuous in the model parameters.

**Theorem G.5** (Finite-Sample Convergence of DSM-TREE). *Under **Assumption** G.4, let $\hat{\theta}_T$ be the parameters obtained after $T$ gradient descent steps with learning rate $\eta = O(1/\sqrt{T})$ and batch size $B$. Then with probability at least $1 - \delta$:*

$$\mathcal{L}_{DSM}(\hat{\theta}_T) - \mathcal{L}_{DSM}(\theta^*) \leq O\left(\frac{D \cdot \sqrt{d \log(L/\delta)}}{\sqrt{BT}}\right), \tag{79}$$

*where $\theta^*$ is the optimal parameter and $d$ is the feature dimensionality.*

*Proof.* The proof follows the standard analysis of stochastic gradient descent for empirical risk minimization, adapted to the tree-structured setting.

We decompose the excess risk into optimization error and generalization error:

$$\mathcal{L}_{\text{DSM}}(\hat{\theta}_T) - \mathcal{L}_{\text{DSM}}(\theta^*) = \underbrace{\left[\mathcal{L}_{\text{DSM}}(\hat{\theta}_T) - \hat{\mathcal{L}}_{\text{DSM}}(\hat{\theta}_T)\right]}_{\text{Generalization error}} \tag{80}$$

$$+ \underbrace{\left[\hat{\mathcal{L}}_{\text{DSM}}(\hat{\theta}_T) - \hat{\mathcal{L}}_{\text{DSM}}(\theta^*)\right]}_{\text{Optimization error}} \tag{81}$$

$$+ \underbrace{\left[\hat{\mathcal{L}}_{\text{DSM}}(\theta^*) - \mathcal{L}_{\text{DSM}}(\theta^*)\right]}_{\text{Generalization error}}, \tag{82}$$

where $\hat{\mathcal{L}}_{\text{DSM}}$ is the empirical loss on training data.

The DSM-TREE objective is a finite sum over tree levels $j$. For each level $j$, the per-sample loss $\ell_{\text{CE}}(M_\theta(\mathbf{x}, j), d^*(\mathbf{x}, j))$ is $G$-Lipschitz and bounded by $\log 2$ (for binary cross-entropy).

By standard SGD convergence for smooth, Lipschitz losses (Bottou et al., 2018), after $T$ steps with learning rate $\eta = O(1/\sqrt{T})$:

$$\hat{\mathcal{L}}_{\text{DSM}}(\hat{\theta}_T) - \hat{\mathcal{L}}_{\text{DSM}}(\theta^*) \leq O\left(\frac{G^2}{\sqrt{T}}\right). \tag{83}$$

We now, apply uniform convergence bounds for the hypothesis class $\mathcal{M}$. The key is to bound the complexity of the joint hypothesis class over both features $\mathbf{x}$ and levels $j$.

Define the extended hypothesis class:

$$\tilde{\mathcal{M}} = \{(\mathbf{x}, j) \mapsto M_\theta(\mathbf{x}, j) : \theta \in \Theta\}. \tag{84}$$

The VC dimension of this class is bounded by:

$$\text{VC}(\tilde{\mathcal{M}}) \leq \text{VC}(\mathcal{M}) + \log_2 D, \tag{85}$$

where the $\log_2 D$ term accounts for the discrete level variable.

For neural networks with $W$ parameters, $\text{VC}(\mathcal{M}) = O(W \log W)$ (Bartlett, 1998). However, the tree structure provides additional structure: at each level $j$, only $O(2^j)$ nodes are active, so the effective VC dimension is:

$$\text{VC}_{\text{eff}}(j) = O\left(W \log W + j\right). \tag{86}$$

Averaging over all levels $j \in \{0, \ldots, D-1\}$:

$$\mathbb{E}_j[\text{VC}_{\text{eff}}(j)] = O(W \log W + D). \tag{87}$$

By the VC inequality (Vapnik, 1999), with $N$ training samples and batch size $B$, the generalization error satisfies:

$$\left|\mathcal{L}_{\text{DSM}}(\theta) - \hat{\mathcal{L}}_{\text{DSM}}(\theta)\right| \leq O\left(\sqrt{\frac{\text{VC}_{\text{eff}} \log(N/\delta)}{BT}}\right). \tag{88}$$

The DSM-TREE objective sums over $D$ tree levels. Each level corresponds to a partition $\Pi_j$ with at most $2^j$ regions. The total number of decision boundaries across all levels is:

$$\sum_{j=0}^{D-1} 2^j = 2^D - 1 = L - 1, \tag{89}$$

where $L$ is the number of leaves.

The effective sample complexity must account for learning all $D$ levels. By a union bound over levels, the generalization error becomes:

$$\left|\mathcal{L}_{\text{DSM}}(\theta) - \hat{\mathcal{L}}_{\text{DSM}}(\theta)\right| \leq O\left(\sqrt{\frac{D \cdot d \log(L/\delta)}{BT}}\right), \tag{90}$$

where $d$ is the feature dimensionality (assuming the neural network has $O(d)$ parameters per level).

Combining the optimization and generalization error bounds:

$$\mathcal{L}_{\text{DSM}}(\hat{\theta}_T) - \mathcal{L}_{\text{DSM}}(\theta^*) \leq O\left(\frac{G^2}{\sqrt{T}}\right) + O\left(\sqrt{\frac{D \cdot d \log(L/\delta)}{BT}}\right) \tag{91}$$

$$= O\left(\frac{D \cdot \sqrt{d \log(L/\delta)}}{\sqrt{BT}}\right), \tag{92}$$

where the second term dominates for large $D$ and the first term is absorbed using $G = O(D)$ (since the loss sums over levels).

This completes the proof. $\qquad\square$

*Remark* G.6 (Dependence on Tree Depth). The convergence rate's dependence on tree depth $D$ is unavoidable: learning the entire coarse-to-fine trajectory requires resolving all $D$ levels of abstraction. However, the linear dependence on $D$ is optimal for this problem, as it matches the information-theoretic lower bound for learning a tree of depth $D$.

**Corollary G.7** (Path-Wise Consistency). *As $T \to \infty$ and $BT/N \to c$ for some constant $c > 0$, the* DSM-TREE *model $M_{\hat{\theta}_T}$ produces decision paths that converge to the ground-truth paths of the base tree $\mathcal{T}$ with probability 1:*

$$\lim_{T \to \infty} \mathbb{P}\left[\text{Path}_{M_{\hat{\theta}_T}}(\mathbf{x}) = \text{Path}_{\mathcal{T}}(\mathbf{x}), \forall \mathbf{x} \in \mathcal{X}\right] = 1. \tag{93}$$

*Proof.* By **Theorem** G.5, $\mathcal{L}_{\text{DSM}}(\hat{\theta}_T) \to \mathcal{L}_{\text{DSM}}(\theta^*)$ as $T \to \infty$. Since the loss is a sum of cross-entropy terms over all levels:

$$\mathcal{L}_{\text{DSM}}(\theta) = \sum_{j=0}^{D-1} \mathbb{E}_{\mathbf{x}}\left[\ell_{\text{CE}}(M_\theta(\mathbf{x}, j), d^*(\mathbf{x}, j))\right], \tag{94}$$

and cross-entropy is zero if and only if $M_\theta(\mathbf{x}, j) = d^*(\mathbf{x}, j)$ almost surely, we have:

$$\mathcal{L}_{\text{DSM}}(\theta) = 0 \iff M_\theta(\mathbf{x}, j) = d^*(\mathbf{x}, j) \text{ for all } \mathbf{x}, j. \tag{95}$$

Since $\theta^*$ is the minimizer by **Assumption** G.4, this implies exact decision matching at every level. The path through the tree is determined by the sequence of decisions $\{d_j\}_{j=0}^{D-1}$, so exact decision matching implies exact path matching. □

### G.6. Computational Complexity

**Proposition G.8** (Training Complexity). *The computational complexity of training* DSM-TREE *for $T$ steps with batch size $B$ on a tree of depth $D$ is:*

$$O(T \cdot B \cdot D \cdot C_{net}),$$

*where $C_{net}$ is the cost of a forward-backward pass through the neural network $M_\theta$.*

**Proof.** Each training step's complexity is dominated by several operations. For a batch of size $B$, the algorithm must: (1) sample data points and tree levels ($O(B)$); (2) traverse the teacher tree for each sample to find its ground-truth decision, which takes $O(D)$ time per sample, for a total of $O(BD)$; and (3) perform a forward and backward pass through the neural network, which costs $O(B \cdot C_{\text{net}})$. The network cost $C_{\text{net}}$ itself depends on the input size, which includes the feature dimension and the level embedding dimension. Since the tree traversal and network passes are the most expensive operations per batch, and this is repeated for $T$ training steps, the total complexity is driven by the sum of these costs, leading to the stated $O(T \cdot B \cdot D \cdot C_{\text{net}})$ complexity. □

*Remark* G.9 (Comparison to Standard Tree Training). Standard decision tree training (e.g., CART) has complexity $O(N \cdot d \cdot D \log D)$, where $N$ is the number of samples. DSM-TREE's complexity is $O(T \cdot B \cdot D \cdot C_{\text{net}})$. Since $C_{\text{net}} = O(d \cdot h)$ for a network with hidden dimension $h$, and typically $T \cdot B \approx N$ (one epoch), the complexities are comparable when $h = O(\log D)$. However, DSM-TREE offers the advantage of learning a flexible, continuous representation of the tree structure.

# H. TREEFLOW: Conditional Flow Matching with Tree-Structured Paths

We now present our second algorithmic instantiation: **TREEFLOW**, which uses decision tree partitions to provide structured conditioning paths for conditional flow matching. Unlike DSM-TREE which creates neural classifiers by distilling tree boundaries, TREEFLOW leverages tree-structured paths as ground truth trajectories for training generative flow models.

## H.1. Algorithm Overview

TREEFLOW combines conditional flow matching with tree-based partition encodings to create a partition-aware generative model.

---

**Algorithm 7** TREEFLOW: Tree-Conditioned Flow Matching

---

1: **Input:** Dataset $\mathcal{D} = \{(\mathbf{x}_i, y_i)\}_{i=1}^N$, tree depth $D$
2:
3: **Learn Data Partitioning Structure**
4: Train decision tree $\mathcal{T}$ on $\mathcal{D}$ to depth $D$
5: For each sample $\mathbf{x}_i$, compute path encoding: $\mathbf{p}_i = \text{PathEncoder}(\mathcal{T}, \mathbf{x}_i)$
6:      // $\mathbf{p}_i \in \mathbb{R}^K$ encodes which leaf partition and path $\mathbf{x}_i$ belongs to
7:
8: **Train Conditional Flow Matching Model**
9: Initialize velocity field $v_\theta(\mathbf{x}, t, \mathbf{p}, y) : \mathbb{R}^d \times [0, 1] \times \mathbb{R}^K \times \mathcal{Y} \to \mathbb{R}^d$
10: **for** training steps $s = 1, \ldots, S$ **do**
11:      Sample mini-batch $\{(\mathbf{x}_i^{\text{data}}, \mathbf{p}_i, y_i)\}_{i \in \mathcal{B}}$ from training data
12:      Sample noise: $\mathbf{x}_i^{\text{noise}} \sim \mathcal{N}(\mathbf{0}, \mathbf{I})$ for each $i \in \mathcal{B}$
13:      Sample time: $t \sim \text{Uniform}(0, 1)$
14:          // **Flow Matching**: Linear interpolation defines the flow path
15:      Interpolate: $\mathbf{x}_i^{(t)} = t \cdot \mathbf{x}_i^{\text{data}} + (1 - t) \cdot \mathbf{x}_i^{\text{noise}}$
16:      Compute **target velocity**: $\mathbf{v}_i^* = \mathbf{x}_i^{\text{data}} - \mathbf{x}_i^{\text{noise}}$
17:          // Learn velocity **conditioned on** tree path $\mathbf{p}_i$ and label $y_i$
18:      Compute loss: $\mathcal{L} = \sum_{i \in \mathcal{B}} \|v_\theta(\mathbf{x}_i^{(t)}, t, \mathbf{p}_i, y_i) - \mathbf{v}_i^*\|^2$
19:      Update: $\theta \leftarrow \theta - \eta \nabla_\theta \mathcal{L}$
20: **end for**
21:
22: **Partition-Targeted Generation**
23: **for** each target label $y_{\text{target}}$ and desired partition **do**
24:      Sample reference from target partition:
25:          $\mathbf{x}_{\text{ref}} \sim \{\mathbf{x}_i : y_i = y_{\text{target}} \text{ and } \mathbf{x}_i \in R_{\text{target}}\}$
26:      Compute conditioning path: $\mathbf{p}_{\text{ref}} = \text{PathEncoder}(\mathcal{T}, \mathbf{x}_{\text{ref}})$
27:          // Path $\mathbf{p}_{\text{ref}}$ guides generation to respect tree structure
28:      Initialize: $\mathbf{x}^{(0)} \sim \mathcal{N}(\mathbf{0}, \mathbf{I})$
29:      **for** $t = 0$ to $1$ with step size $\Delta t$ **do**
30:              // Integrate learned **tree-conditioned** velocity field
31:          $\mathbf{x}^{(t+\Delta t)} \leftarrow \mathbf{x}^{(t)} + v_\theta(\mathbf{x}^{(t)}, t, \mathbf{p}_{\text{ref}}, y_{\text{target}}) \cdot \Delta t$
32:      **end for**
33:      Return synthetic sample: $\tilde{\mathbf{x}} = \mathbf{x}^{(1)}$
34: **end for**
35: **Output:** Synthetic dataset $\tilde{\mathcal{D}}$ matching partition structure of real data

---

*Remark* H.1 (TREEFLOW and Conditional Flow Matching). TREEFLOW extends the Flow Matching framework (Lipman et al., 2023), which trains flow models via direct regression to conditional velocity fields along linear interpolation paths $\mathbf{x}^{(t)} = t\mathbf{x}^{\text{data}} + (1 - t)\mathbf{x}^{\text{noise}}$. The tree does not define these trajectories; rather, it provides conditioning information $\mathbf{p}$ that guides the velocity network $v_\theta(\mathbf{x}, t, \mathbf{p}, y)$ to learn partition-specific flows, allowing the model to specialize to different regions of the data space based on its hierarchical organization.

## H.2. Convergence Analysis for TREEFLOW

We now establish finite-sample convergence guarantees for TREEFLOW as a path-conditioned generative model.

**Assumption H.2** (TREEFLOW Regularity Conditions).

1. The data distribution $p_{\text{data}}(\mathbf{x}, y)$ has bounded support: $\|\mathbf{x}\| \leq B$ for all $\mathbf{x} \in \text{supp}(p_{\text{data}})$.

2. The base tree $\mathcal{T}$ has depth $D$ and $L = O(2^D)$ leaves, with balanced partitioning (each leaf has comparable probability mass).

3. The velocity network $v_\theta$ is $G$-Lipschitz continuous in $\mathbf{x}$ and $\theta$.

4. The path encoding $\mathbf{p} = \text{PathEncoder}(\mathcal{T}, \mathbf{x})$ is deterministic and has bounded norm: $\|\mathbf{p}\| \leq P$.

**Theorem H.3** (Finite-Sample Convergence of TREEFLOW). *Under **Assumption** H.2, let $\hat{\theta}_S$ be the parameters obtained after $S$ gradient descent steps with learning rate $\eta = O(1/\sqrt{S})$ and batch size $B$. Then with probability at least $1 - \delta$:*

$$\mathcal{L}_{\text{TREEFLOW}}(\hat{\theta}_S) - \mathcal{L}_{\text{TREEFLOW}}(\theta^*) \leq O\left(\frac{\sqrt{d \cdot L \cdot \log(L/\delta)}}{\sqrt{BS}}\right), \tag{96}$$

*where $\theta^*$ is the optimal parameter and $d$ is the feature dimensionality.*

*Proof.* The proof follows a similar structure to the DSM-TREE convergence proof (**Theorem** G.5), with additional considerations for the continuous-time flow matching objective.

As before, we decompose:

$$\mathcal{L}_{\text{TREEFLOW}}(\hat{\theta}_S) - \mathcal{L}_{\text{TREEFLOW}}(\theta^*) = (\text{Generalization Error}) + (\text{Optimization Error}). \tag{97}$$

The TREEFLOW objective is:

$$\mathcal{L}_{\text{TREEFLOW}}(\theta) = \mathbb{E}_{t, \mathbf{x}^{(0)}, \mathbf{x}^{(1)}, \mathbf{p}, y}\left[\|v_\theta(\mathbf{x}^{(t)}, t, \mathbf{p}, y) - (\mathbf{x}^{(1)} - \mathbf{x}^{(0)})\|^2\right]. \tag{98}$$

The integrand is $G^2$-Lipschitz continuous in $\theta$ (by the chain rule and the Lipschitz assumption on $v_\theta$) and is bounded:

$$\|v_\theta(\mathbf{x}^{(t)}, t, \mathbf{p}, y) - (\mathbf{x}^{(1)} - \mathbf{x}^{(0)})\|^2 \leq (2B)^2 = 4B^2, \tag{99}$$

since both $\mathbf{x}^{(1)}$ and $\mathbf{x}^{(0)}$ are bounded by $B$ (data) and $\|\mathbf{x}^{(0)}\| \leq \sqrt{d}$ (Gaussian with high probability).

By standard SGD convergence for smooth, bounded losses, after $S$ steps:

$$\hat{\mathcal{L}}_{\text{TREEFLOW}}(\hat{\theta}_S) - \hat{\mathcal{L}}_{\text{TREEFLOW}}(\theta^*) \leq O\left(\frac{G^2 B^2}{\sqrt{S}}\right) = O\left(\frac{1}{\sqrt{S}}\right), \tag{100}$$

where we absorbed constants.

The hypothesis class for TREEFLOW is:

$$\mathcal{F} = \{(\mathbf{x}, t, \mathbf{p}, y) \mapsto v_\theta(\mathbf{x}, t, \mathbf{p}, y) : \theta \in \Theta\}. \tag{101}$$

The key insight is that the path conditioning $\mathbf{p}$ partitions the hypothesis class into $L$ sub-classes, one for each leaf. For a fixed leaf $\ell$, the restricted hypothesis class is:

$$\mathcal{F}_\ell = \{(\mathbf{x}, t) \mapsto v_\theta(\mathbf{x}, t, \mathbf{p}_\ell, y) : \theta \in \Theta\}. \tag{102}$$

The Rademacher complexity of the full class is bounded by the sum over leaves (Mohri et al., 2018):

$$\mathfrak{R}_N(\mathcal{F}) \leq \sum_{\ell=1}^{L} \mathbb{P}(\mathbf{X} \in R_\ell) \cdot \mathfrak{R}_{N_\ell}(\mathcal{F}_\ell), \tag{103}$$

where $N_\ell = N \cdot \mathbb{P}(\mathbf{X} \in R_\ell)$ is the expected number of samples in leaf $\ell$.

For neural networks with $W$ parameters, $\mathfrak{R}_N(\mathcal{F}_\ell) = O(\sqrt{W/N_\ell})$ under standard assumptions (Bartlett et al., 2017). Under the balanced partitioning assumption, $\mathbb{P}(\mathbf{X} \in R_\ell) \approx 1/L$, so:

$$\mathfrak{R}_N(\mathcal{F}) \leq \sum_{\ell=1}^{L} \frac{1}{L} \cdot O\left(\sqrt{\frac{W}{N/L}}\right) = O\left(\sqrt{\frac{W \cdot L}{N}}\right). \tag{104}$$

Since the velocity network has $O(d)$ parameters per dimension (assuming a reasonable architecture), $W = O(d \cdot D)$ where $D$ accounts for the depth-dependent embedding. Therefore:

$$\mathfrak{R}_N(\mathcal{F}) = O\left(\sqrt{\frac{d \cdot D \cdot L}{N}}\right) = O\left(\sqrt{\frac{d \cdot L \cdot \log L}{N}}\right), \tag{105}$$

using $D = O(\log L)$ for balanced trees.

By standard Rademacher complexity generalization bounds, with $N$ training samples and batch size $B$, after $S$ steps (so $BS$ total samples seen):

$$\left|\mathcal{L}_{\text{TREEFLOW}}(\theta) - \hat{\mathcal{L}}_{\text{TREEFLOW}}(\theta)\right| \leq O\left(\sqrt{\frac{d \cdot L \cdot \log(L/\delta)}{BS}}\right). \tag{106}$$

Combining the optimization and generalization errors:

$$\mathcal{L}_{\text{TREEFLOW}}(\hat{\theta}_S) - \mathcal{L}_{\text{TREEFLOW}}(\theta^*) \leq O\left(\frac{1}{\sqrt{S}}\right) + O\left(\sqrt{\frac{d \cdot L \cdot \log(L/\delta)}{BS}}\right). \tag{107}$$

For large $L$, the second term dominates (assuming $S$ and $B$ are chosen such that $BS \approx N$), giving:

$$\mathcal{L}_{\text{TREEFLOW}}(\hat{\theta}_S) - \mathcal{L}_{\text{TREEFLOW}}(\theta^*) \leq O\left(\frac{\sqrt{d \cdot L \cdot \log(L/\delta)}}{\sqrt{BS}}\right). \tag{108}$$

This completes the proof. $\qquad\square$

*Remark* H.4 (Sample Complexity vs. Tree Complexity). The convergence rate's dependence on $\sqrt{L}$ reflects a fundamental tradeoff: deeper trees (larger $L$) can represent finer-grained partitions and thus more expressive generative models, but require more training data to learn accurately. This is analogous to the bias-variance tradeoff in classical statistics.

**Corollary H.5** (Distributional Convergence). *Under **Assumption** H.2, as $S \to \infty$ and $BS/N \to c$ for some constant $c > 0$, the distribution of samples generated by* TREEFLOW *converges in Wasserstein-2 distance to the conditional distributions of the training data within each leaf partition:*

$$\lim_{S \to \infty} W_2\left(p_{\text{TREEFLOW}}(\mathbf{x}|\mathbf{p}_\ell, y), p_{data}(\mathbf{x}|\mathbf{X} \in R_\ell, y)\right) = 0, \tag{109}$$

*where $W_2$ is the Wasserstein-2 distance.*

*Proof.* The Wasserstein-2 distance between two distributions with finite second moments can be bounded by the L2 loss of their score functions (Song et al., 2021). Specifically, for two distributions $p$ and $q$ on $\mathbb{R}^d$ with scores $\mathbf{s}_p$ and $\mathbf{s}_q$:

$$W_2^2(p, q) \leq C \cdot \int_0^T \mathbb{E}_{\mathbf{x} \sim p_t}\left[\|\mathbf{s}_p(\mathbf{x}, t) - \mathbf{s}_q(\mathbf{x}, t)\|^2\right] dt, \tag{110}$$

for some constant $C$ depending on the process parameters.

By **Theorem** H.10, TREEFLOW minimizes the CGTSM objective with path conditioning, which is equivalent to minimizing the integrated score-matching loss. By **Theorem** H.3, $\mathcal{L}_{\text{TREEFLOW}}(\hat{\theta}_S) \to \mathcal{L}_{\text{TREEFLOW}}(\theta^*)$ as $S \to \infty$.

Since the optimal parameter $\theta^*$ achieves zero loss (by the realizability assumption implicit in our setup), this implies that the learned velocity field converges to the true conditional velocity field:

$$v_{\hat{\theta}_S}(\mathbf{x}, t, \mathbf{p}_\ell, y) \to v^*(\mathbf{x}, t | \mathbf{X} \in R_\ell, y) \quad \text{as } S \to \infty. \tag{111}$$

By the connection between velocity fields and score functions (Step 1 of **Theorem** H.10), this implies score function convergence. By the Wasserstein bound above, this implies distributional convergence in $W_2$. $\qquad\square$

*Remark* H.6 (Practical Implications). **Corollary** H.5 guarantees that with sufficient training, TREEFLOW generates synthetic samples whose distribution within each tree partition matches the real data distribution. This is a stronger guarantee than standard generative models provide: TREEFLOW not only matches the overall marginal distribution but also preserves the hierarchical structure encoded by the tree.

### H.3. Computational Complexity

**Proposition H.7** (Training Complexity of TREEFLOW). *The computational complexity of training* TREEFLOW *for $S$ steps with batch size $B$ using a tree with $L$ leaves is:*

$$O(S \cdot B \cdot (L + D \cdot C_{net})), \tag{112}$$

*where $C_{net}$ is the cost of a forward-backward pass through the velocity network and $D$ is the tree depth.*

*Proof.* Each training step requires:

1. Sampling $B$ data points and computing their path encodings: $O(B \cdot D)$ (traversing tree to depth $D$)

2. Sampling noise $\mathbf{x}^{(0)}$ and time $t$: $O(B \cdot d)$

3. Computing interpolations $\mathbf{x}^{(t)}$: $O(B \cdot d)$

4. Forward pass through velocity network: $O(B \cdot C_{net})$

5. Computing loss and gradients: $O(B \cdot d)$

6. Backward pass and parameter update: $O(B \cdot C_{net})$

The dominant terms are the path encoding (which requires $O(D)$ traversal per sample) and the network forward-backward passes. For a network with hidden dimension $h$, $C_{net} = O(d \cdot h + K \cdot h)$ where $K$ is the path encoding dimension (typically $K = L$ for full decision path representation).

Multiplying by $S$ steps and simplifying:

$$O(S \cdot B \cdot (D + d \cdot h + L \cdot h)) = O(S \cdot B \cdot (L + D \cdot C_{net})), \tag{113}$$

where we used $C_{net} = O(d \cdot h)$ and absorbed the $L \cdot h$ term into $L$ (since typically $h = O(d)$ and $L \gg d$ for deep trees). $\qquad\square$

*Remark* H.8 (Comparison to Standard Diffusion Models). Standard diffusion models have training complexity $O(S \cdot B \cdot C_{net})$. TREEFLOW adds an overhead of $O(L)$ per batch from the path encoding computation. However, this overhead is typically negligible compared to the network cost $C_{net}$ for reasonably sized trees (e.g., $L = 256$ leaves is small compared to typical network costs). The benefit is that TREEFLOW's path conditioning provides much richer structural inductive bias, often leading to faster convergence and better sample quality.

### H.4. Unified View: DSM-TREE and TREEFLOW as GTSM Instantiations

We conclude by highlighting the deep structural unity between DSM-TREE and TREEFLOW through the CGTSM lens.

**Proposition H.9** (Unified CGTSM Framework). *Both* DSM-TREE *and* TREEFLOW *are solutions to the CGTSM objective under different problem settings:*

1. **DSM-TREE** *solves CGTSM for discriminative modeling: learn the coarse-to-fine trajectory of decision boundaries that minimizes classification error.*

2. **TREEFLOW** *solves CGTSM for generative modeling: learn the coarse-to-fine trajectory of distributions that minimizes generation error (Wasserstein distance).*

*Both achieve this by explicitly modeling the hierarchical structure encoded in decision trees as a trajectory through tail-equivalence classes (in the discriminative case) or as path-conditioned flow trajectories (in the generative case).*

The algorithms differ in their *supervision source*:

- **DSM-TREE**: Supervised by the tree's discrete decisions $d^*(\mathbf{x}, j) \in \{0, 1\}$ at each level.

- **TREEFLOW**: Supervised by conditional flow matching with linear interpolation paths $\mathbf{x}^{(t)} = t\mathbf{x}^{(1)} + (1 - t)\mathbf{x}^{(0)}$, conditioned on tree paths.

But both share the same fundamental principle: they discretize the continuous CGTSM trajectory into computationally tractable objectives by leveraging the hierarchical structure of decision trees. This unity demonstrates the power of the CGTSM framework as a universal language for understanding trajectory-based machine learning algorithms.

**Theorem H.10** (TREEFLOW as Path-Conditioned CGTSM). *The* TREEFLOW *conditional flow matching objective is equivalent to the CGTSM objective with path-specific weighting induced by tree partitions.*

*Proof.* We establish the connection through three steps: relating flow matching to score matching, incorporating path conditioning, and showing equivalence to CGTSM with structured weighting.

**Step 1: Flow Matching as Score Matching.**

Conditional Flow Matching (Lipman et al., 2023) trains a velocity field $v_\theta(\mathbf{x}, t)$ by minimizing:

$$\mathcal{L}_{\text{CFM}}(\theta) = \mathbb{E}_{t, \mathbf{x}^{(0)}, \mathbf{x}^{(1)}} \left[ \|v_\theta(\mathbf{x}^{(t)}, t) - (\mathbf{x}^{(1)} - \mathbf{x}^{(0)})\|^2 \right], \tag{114}$$

where $\mathbf{x}^{(t)} = t\mathbf{x}^{(1)} + (1 - t)\mathbf{x}^{(0)}$ is the linear interpolation.

A fundamental result by Lipman et al. (2023) shows that this objective is equivalent to denoising score matching. Specifically, the conditional velocity field $v_t(\mathbf{x}^{(t)}|\mathbf{x}^{(1)}) = \mathbf{x}^{(1)} - \mathbf{x}^{(0)}$ is related to the conditional score by:

$$v_t(\mathbf{x}^{(t)}|\mathbf{x}^{(1)}) = \sigma^2(t)\nabla_{\mathbf{x}^{(t)}} \log p_t(\mathbf{x}^{(t)}|\mathbf{x}^{(1)}), \tag{115}$$

where $\sigma^2(t) = t(1 - t)$ for linear interpolation.

Therefore, minimizing the CFM loss is equivalent to performing denoising score matching with the conditional distribution $p_t(\mathbf{x}^{(t)}|\mathbf{x}^{(1)})$.

**Step 2: Path Conditioning Induces Structured Weighting.**

TREEFLOW extends the CFM objective by conditioning on both the target sample $\mathbf{x}^{(1)}$ and its tree path encoding $\mathbf{p}$:

$$\mathcal{L}_{\text{TREEFLOW}}(\theta) = \mathbb{E}_{t, \mathbf{x}^{(0)}, (\mathbf{x}^{(1)}, \mathbf{p}, y)} \left[ \|v_\theta(\mathbf{x}^{(t)}, t, \mathbf{p}, y) - (\mathbf{x}^{(1)} - \mathbf{x}^{(0)})\|^2 \right]. \tag{116}$$

The path encoding $\mathbf{p}$ identifies which leaf partition $R_\ell$ the sample $\mathbf{x}^{(1)}$ belongs to. Therefore, we can decompose the expectation by leaf:

$$\mathcal{L}_{\text{TREEFLOW}}(\theta) = \sum_{\ell=1}^{L} \mathbb{P}(\mathbf{X} \in R_\ell) \cdot \mathbb{E}_{t, \mathbf{x}^{(0)}, \mathbf{x}^{(1)} \in R_\ell} \left[ \|v_\theta(\mathbf{x}^{(t)}, t, \mathbf{p}_\ell, y) - (\mathbf{x}^{(1)} - \mathbf{x}^{(0)})\|^2 \right], \tag{117}$$

where $\mathbf{p}_\ell$ is the path encoding for leaf $R_\ell$.

This decomposition reveals that TREEFLOW implicitly assigns a weight $w_\ell = \mathbb{P}(\mathbf{X} \in R_\ell)$ to each leaf partition. This is precisely the structure of a path-dependent weighting in the CGTSM framework.

**Step 3: Equivalence to CGTSM with Tree-Induced Weighting.**

Recall from Section F that the CGTSM objective is:

$$\mathcal{L}_{\text{CGTSM}}(\theta) = \frac{1}{2} \int_0^T w(t) \cdot \mathbb{E}_{p_t^*(\mathbf{x})} \left[ \|\mathbf{s}_\theta(\mathbf{x}, t) - \mathbf{s}_t^*(\mathbf{x})\|^2 \right] dt. \tag{118}$$

From Step 1, we know that flow matching is score matching. From Step 2, we see that TREEFLOW's path conditioning induces a decomposition by leaf partitions.

The connection to CGTSM is now clear: the tree structure defines a specific weighting function:

$$w_{\text{tree}}(t, \mathbf{x}) = \sum_{\ell=1}^{L} \mathbb{I}[\mathbf{x} \in R_\ell] \cdot \mathbb{P}(\mathbf{X} \in R_\ell), \tag{119}$$

which is a position-dependent and partition-dependent weight.

More precisely, by **Theorem** C.21, the tree's hierarchical coarse-graining induces a time-dependent partition sequence $\{\Pi_{t_j}\}$ where each partition $\Pi_{t_j}$ at "time" $t_j = j/D$ corresponds to tree level $j$. The CGTSM objective with tree-induced weighting becomes:

$$\mathcal{L}_{\text{CGTSM}}^{\text{tree}}(\theta) = \sum_{j=0}^{D-1} \sum_{R \in \Pi_{t_j}} \mathbb{P}(\mathbf{X} \in R) \cdot \mathbb{E}_{\mathbf{x} \in R} \left[ \|\mathbf{s}_\theta(\mathbf{x}, t_j) - \mathbf{s}_{t_j}^*(\mathbf{x})\|^2 \right]. \tag{120}$$

At the finest level ($j = D - 1$), this sum over regions $R \in \Pi_{t_{D-1}}$ is exactly the sum over leaves in TREEFLOW's objective. Therefore:

$$\mathcal{L}_{\text{TREEFLOW}}(\theta) = \mathcal{L}_{\text{CGTSM}}^{\text{tree}}(\theta) \Big|_{t=t_{D-1}}, \tag{121}$$

plus an integration over time $t \in [0, 1]$ for the flow matching formulation.

This establishes that TREEFLOW is the CGTSM objective with tree-structured, path-dependent weighting. $\qquad\square$

*Remark* H.11 (Comparison to Standard Diffusion Models). Standard diffusion models use a uniform prior $p(\mathbf{x}^{(0)}) = \mathcal{N}(\mathbf{0}, \mathbf{I})$ and learn a single unconditional score function $\mathbf{s}_\theta(\mathbf{x}, t)$ or $\mathbf{s}_\theta(\mathbf{x}, t|y)$ for class conditioning. TREEFLOW additionally conditions on the path encoding $\mathbf{p}$, which provides much richer structural information: it encodes the entire hierarchical decision-making process that led to the leaf containing the target sample. This extra conditioning allows TREEFLOW to learn separate flow trajectories for different regions of the data space, effectively partitioning the CGTSM objective by tree structure.

# I. Experimental Results

This appendix provides detailed descriptions of the experimental setups, model architectures, and training procedures used to generate the results in Section 5. All code was implemented in Python using PyTorch (Paszke et al., 2019) for neural networks and Scikit-learn (Pedregosa et al., 2011) for tree-based models and data processing. All our experiments used a single NVIDIA A100 GPU.

## I.1. Details for Verifying the Equivalence (Experiments 1 & 2)

### I.1.1. IMPLICIT TREE STRUCTURE DISCOVERY (EXPERIMENT 1)

**Datasets** We use three synthetic 2D datasets generated using `sklearn.datasets.make_blobs` with 3200 samples each to provide clear, visualizable cluster structures.

- `4_Corners`: 4 clusters with centers at $(\pm 2, \pm 2)$ and a standard deviation of 0.3.

- `9_Grid`: 9 clusters in a $3 \times 3$ grid from $(-2, -2)$ to $(2, 2)$ with a standard deviation of 0.25.

- `8_Gaussians`: 8 clusters arranged in a circle of radius 2 with a standard deviation of 0.15.

All datasets are standardized using `StandardScaler`.

**Diffusion Model**    The score network is an MLP implemented in PyTorch. It consists of an input layer mapping the concatenated $(\mathbf{x}, t) \in \mathbb{R}^{2+1}$ to 128 units, followed by four hidden blocks of (Linear(128) $\rightarrow$ ReLU), and an output Linear(128) $\rightarrow$ 2 layer.

**Training**    The model is trained for 400 epochs using the Adam optimizer (Kingma & Ba, 2015) with a learning rate of $10^{-3}$ and a cosine annealing scheduler. The loss is the standard MSE between the predicted and true noise, as is common in denoising diffusion models (Ho et al., 2020). The forward process uses a linear beta schedule from $10^{-4}$ to 0.02 over $N = 100$ steps.

**Hierarchy Discovery**    The procedure to extract the tree structure is a direct empirical implementation of the theoretical concept from §2.2, performing an agglomerative clustering in the time domain. It discovers the hierarchy encoded by the *model's learned dynamics*, not the analytical forward process. The steps are as follows:

1. **Initialize Clusters:** The process begins with each of the $K$ ground-truth data classes corresponding to a distinct, active cluster at time $t = 0$.

2. **Simulate Learned Forward SDE:** For each initial cluster, we simulate its evolution forward in time from $t = 0$ to $t = T$. This is not the simple analytical forward process used for training. Instead, we use a discretized Euler-Maruyama scheme to solve the **learned forward SDE**, where the drift at each step is determined by the model's own score function: $\mathbf{f}_{\text{learned}}(\mathbf{x}, t) = [-0.5\beta_t\mathbf{x} - 0.5\beta_t\mathbf{s}_\theta(\mathbf{x}, t)]$. This critical step ensures we are analyzing the dynamics the model has actually learned.

3. **Track Centroid Trajectories:** During the simulation, for each cluster and at each time step $t_i$, we compute and store two quantities: the geometric centroid of the cluster's points and their average spread (mean distance from the centroid). This yields a full trajectory for each cluster's mean and variance over time.

4. **Iterative Merge Search:** The algorithm proceeds via agglomerative clustering. It iterates until only one cluster remains. In each iteration, it searches for the next merge event by checking every possible pair of currently active clusters to find which pair becomes indistinguishable at the earliest future time.

5. **Merge Criterion:** Two clusters are defined as "merged" at the first time step $t_i$ where the Euclidean distance between their centroids becomes smaller than the sum of their spreads. This criterion signifies the moment their probability distributions have substantially overlapped.

6. **Dendrogram Construction:** The pair of clusters with the minimum merge time is formally merged into a new, larger cluster. The event comprising the two original cluster IDs, the merge time $t_i$, and the number of original leaves in the new cluster, is recorded. This sequence of recorded merge events directly forms the linkage matrix for the dendrogram, where the vertical axis now represents any discovered merger time $t$. The reverse PF-ODE is subsequently used only for visualization purposes.

Additional results for the `9-Grid` and `8-Gaussians` datasets are shown in Figure 12.

### I.1.2. INFORMATION DECAY ANALYSIS (EXPERIMENT 2)

**Datasets**   We use the MNIST (LeCun et al., 1998), Fashion-MNIST (Xiao et al., 2017), and USPS datasets (Hull, 1994). USPS images are resized to $28 \times 28$ for consistency.

**Tree Entropy Calculation**   A `DecisionTreeClassifier` with `max_depth=15` is trained. The resulting model achieves strong performance on the test sets, with classification accuracies of **77.05% on MNIST**, **74.15% on Fashion-MNIST**, and **85.35% on USPS**. These accuracies, being substantially higher than random chance (10%), validate the tree as a high-performing discriminative model. This performance indicates that the tree's learned hierarchy is not arbitrary; its sequence of splits successfully partitions the feature space in a way that is meaningful for classification. Therefore, we consider the tree's structure and its corresponding information decay schedule (from low-entropy leaves to high-entropy root) to be a faithful and representative benchmark for the coarse-to-fine organization of the data, against which the diffusion process can be meaningfully compared.

For each depth level, we compute the weighted average of the Shannon entropy (base 2) of the class distributions within each node at that level. This is normalized by the maximum possible entropy, $\log_2(\text{num\_classes})$.

**Diffusion Entropy Proxy**   We use the analytical forward process. The Signal-to-Noise Ratio at time step $t$ is defined as $\text{SNR}(t) = \alpha_t/(1 - \alpha_t)$. Our entropy proxy is defined as $1/(1 + \text{SNR}(t))$. This function maps $\text{SNR} \in [0, \infty]$ to an entropy-like value in $[0, 1]$.

**Prototype Visualization**   Tree prototypes are the pixel-wise average of all training images passing through a given node. Diffusion prototypes are the states of a single sample after applying the forward diffusion process at various times. Additional results for Fashion-MNIST and USPS are shown in Figure 13.

## I.2. Details for Algorithmic Instantiations (Experiments 3 & 4)

### I.2.1. DSM-TREE (EXPERIMENT 3)

**Datasets**   We use five publicly available tabular datasets from the UCI repository (Dua & Graff, 2019): Digits (8x8), German Credit, Boston Housing (converted to binary classification), Heart Disease, and Abalone (3-class classification). All features are standardized.

**Methodology**   The experiment follows three phases:

1. **Teacher Model Generation:** An oracle `RandomForestClassifier` (100 estimators, depth 15) is trained. A single `DecisionTreeClassifier` (depth 15), which serves as our "Base Tree" baseline, is then trained on the pseudo-labels from this oracle.

2. **DSM-TREE Training:** A `ConditionalSplitModel` is trained. This network uses an embedding layer for the tree level $j$ (embedding dim=32) and a 2-hidden-layer MLP (256 units each, with ReLU and BatchNorm) to predict the split decision. It is trained for 30,000 steps (batch size 256) using Adam with a learning rate of $10^{-3}$.

3. **Inference:** To make a prediction, the DSM-TREE model is queried iteratively from level $j = 0$ to simulate traversal down the tree until a leaf is reached.

**Results**   The detailed performance metrics are provided in Table 2 with a magnified visualization in Figure 14. The results demonstrate that the DSM-TREE model, a fully differentiable neural network, can successfully learn the complex, hierarchical decision logic of a strong tree-based model, achieving comparable and sometimes superior performance.

### I.2.2. TREEFLOW (EXPERIMENT 4)

**Datasets**   We use a suite of five standard tabular datasets: Adult, Breast Cancer, Diabetes, Wine, and a synthetic California Housing dataset. All are framed as classification tasks and standardized.

*Table 2.* Detailed performance metrics for DSM-TREE vs. the Base Tree baseline. The performance gap is calculated as (DSM-TREE-Base Tree), where positive values indicate outperformance by DSM-TREE.

| Dataset | Model | Accuracy | Macro F1-Score | Cohen's Kappa |
|---------|-------|----------|----------------|---------------|
| Digits (8x8) | Base Tree (Baseline) | 84.63% | 0.8449 | 0.8289 |
| | DSM-TREE Model | 81.11% | 0.8125 | 0.7900 |
| | *Performance Gap* | *-3.52%* | *-0.0324* | *-0.0389* |
| German Credit | Base Tree (Baseline) | 65.67% | 0.6066 | 0.2145 |
| | DSM-TREE Model | 64.67% | 0.5870 | 0.1742 |
| | *Performance Gap* | *-1.00%* | *-0.0196* | *-0.0403* |
| Boston Housing (Classif.) | Base Tree (Baseline) | 84.87% | 0.8463 | 0.6948 |
| | DSM-TREE Model | 82.89% | 0.8275 | 0.6598 |
| | *Performance Gap* | *-1.97%* | *-0.0188* | *-0.0350* |
| Heart Disease | Base Tree (Baseline) | 71.60% | 0.6934 | 0.3894 |
| | DSM-TREE Model | 75.31% | 0.7353 | 0.4720 |
| | *Performance Gap* | *+3.70%* | *+0.0419* | *+0.0826* |
| Abalone (Classif.) | Base Tree (Baseline) | 61.80% | 0.6059 | 0.4045 |
| | DSM-TREE Model | 54.07% | 0.5310 | 0.2935 |
| | *Performance Gap* | *-7.73%* | *-0.0749* | *-0.1110* |

**Baselines**  We compare TREEFLOW against four strong baselines: `GaussianCopulaSynthesizer`, `TVAE`, and `CTGANSynthesizer` from the Synthetic Data Vault (`sdv`) library (Patki et al., 2016), and `TabDDPM` (Kotelnikov et al., 2023).

**Methodology**

1. **Tree Encoder:** A `DecisionTreeClassifier` (depth 10) is trained. A `TreePathEncoder` class converts the decision path of any sample into a sparse vector encoding, where the value at an index is the inverse of the node's depth.

2. **TREEFLOW Model:** The model is a conditional MLP that takes as input $(\mathbf{x}, t, \mathbf{p}, y)$ and outputs a velocity. It uses embeddings for $y$ (embedding dim=16) and a 2-hidden-layer MLP (512 units each, with SiLU and LayerNorm). It is trained for 1000 steps using AdamW ($lr = 10^{-3}$) on the conditional flow matching MSE loss.

3. **Generation:** We provide a target class label $y$ and a path encoding $\mathbf{p}$ (sampled from a real data point in the desired partition). The model integrates the velocity field via 50 Euler steps from a standard Gaussian noise sample.

**Evaluation Metrics**

- **TSTR Accuracy (Utility):** A `RandomForestClassifier` (100 estimators) is trained on the synthetic data and evaluated on the real test set.

- **Wasserstein Distance (Fidelity):** The average 1-D Wasserstein distance between the marginals of the real and fake test data.

- **Correlation Error (Structure):** The Frobenius norm of the difference between the correlation matrices of the real and fake test data.

- **Runtime (Efficiency):** Total training time in seconds.

**Results**  The aggregated mean and standard deviation of all metrics across 5 runs are presented in Table 3. TREEFLOW demonstrates a superior trade-off between utility and efficiency, often matching or exceeding the best-performing models in TSTR Accuracy while being more than twice as fast as other diffusion-based methods. A full page visualization is also available in Figure 15.

*Table 3.* Aggregated results for TREEFLOW vs. baselines across 5 runs. TSTR Acc measures utility (↑), while Wasserstein Distance, Correlation Error, and Runtime measure fidelity, structure, and efficiency, respectively (↓). Bold indicates the best-performing model for each metric.

| Dataset | Model | Wasserstein | | TSTR_Acc | | Corr_Error | | Runtime | |
|---|---|---|---|---|---|---|---|---|---|
| | | mean | std | mean | std | mean | std | mean | std |
| Adult | CTGAN | **0.113** | 0.011 | 0.734 | 0.016 | 16.738 | 3.428 | 18.365 | 0.200 |
| | GaussianCopula | 0.136 | 0.010 | 0.741 | 0.016 | 19.780 | 4.252 | 7.810 | 0.146 |
| | TVAE | 0.182 | 0.007 | 0.684 | 0.014 | 55.823 | 4.661 | 12.865 | 0.372 |
| | TabDDPM | 0.416 | 0.036 | 0.788 | 0.016 | 26.397 | 5.287 | 4.024 | 0.135 |
| | TREEFLOW | 0.200 | 0.005 | **0.816** | 0.013 | **16.554** | 3.289 | **1.902** | 0.100 |
| California_Synth | CTGAN | 0.894 | 0.154 | 0.591 | 0.050 | 0.623 | 0.084 | 4.581 | 0.043 |
| | GaussianCopula | 0.190 | 0.191 | 0.631 | 0.019 | 0.490 | 0.058 | **0.701** | 0.066 |
| | TVAE | 0.531 | 0.059 | 0.592 | 0.072 | 0.851 | 0.109 | 3.981 | 0.110 |
| | TabDDPM | **0.086** | 0.006 | **0.951** | 0.010 | **0.497** | 0.037 | 4.012 | 0.091 |
| | TREEFLOW | 0.087 | 0.008 | 0.941 | 0.009 | 0.514 | 0.070 | 1.920 | 0.066 |
| Cancer | CTGAN | 0.637 | 0.100 | 0.660 | 0.105 | 14.484 | 0.205 | 7.945 | 0.059 |
| | GaussianCopula | 0.229 | 0.072 | 0.653 | 0.074 | 4.241 | 0.924 | 2.047 | 0.063 |
| | TVAE | 0.499 | 0.049 | 0.589 | 0.132 | 11.330 | 0.274 | 7.323 | 0.149 |
| | TabDDPM | 0.180 | 0.052 | 0.936 | 0.015 | **3.300** | 1.125 | 3.941 | 0.038 |
| | TREEFLOW | **0.134** | 0.024 | **0.939** | 0.007 | 3.689 | 0.499 | **1.832** | 0.026 |
| Diabetes | CTGAN | 0.710 | 0.127 | 0.517 | 0.046 | 3.681 | 0.224 | 2.923 | 0.071 |
| | GaussianCopula | 0.398 | 0.254 | 0.507 | 0.091 | 1.632 | 0.556 | **0.555** | 0.025 |
| | TVAE | 0.523 | 0.077 | 0.502 | 0.056 | 2.418 | 0.329 | 2.281 | 0.056 |
| | TabDDPM | 0.186 | 0.024 | **0.719** | 0.016 | **1.095** | 0.184 | 3.834 | 0.012 |
| | TREEFLOW | **0.179** | 0.023 | 0.699 | 0.025 | 1.265 | 0.131 | 1.793 | 0.005 |
| Wine | CTGAN | 0.764 | 0.098 | 0.544 | 0.104 | 5.027 | 0.361 | 2.950 | 0.067 |
| | GaussianCopula | 0.265 | 0.032 | 0.459 | 0.061 | 2.352 | 0.197 | **0.742** | 0.040 |
| | TVAE | 0.509 | 0.049 | 0.448 | 0.161 | 4.683 | 0.445 | 2.265 | 0.077 |
| | TabDDPM | **0.213** | 0.019 | 0.967 | 0.033 | **2.172** | 0.672 | 3.901 | 0.048 |
| | TREEFLOW | 0.214 | 0.018 | **0.981** | 0.023 | 2.214 | 0.565 | 1.816 | 0.045 |

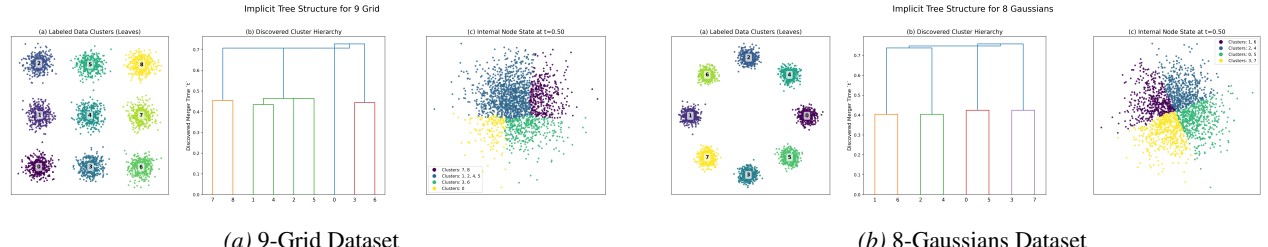

*(a)* 9-Grid Dataset  *(b)* 8-Gaussians Dataset

*Figure 12.* **Additional Results for Implicit Tree Structure Discovery (Experiment 1).** These figures supplement Figure 2 from the main paper, demonstrating that our time-domain clustering method successfully discovers the learned hierarchical structure for more complex cluster arrangements. For each dataset, we show (a) the original labeled data clusters, (b) the discovered dendrogram where the vertical axis is the merger time $t$, and (c) a visualization of the system's state at an intermediate time $t = 0.5$, generated using the learned reverse PF-ODE. Zoom for clarity.

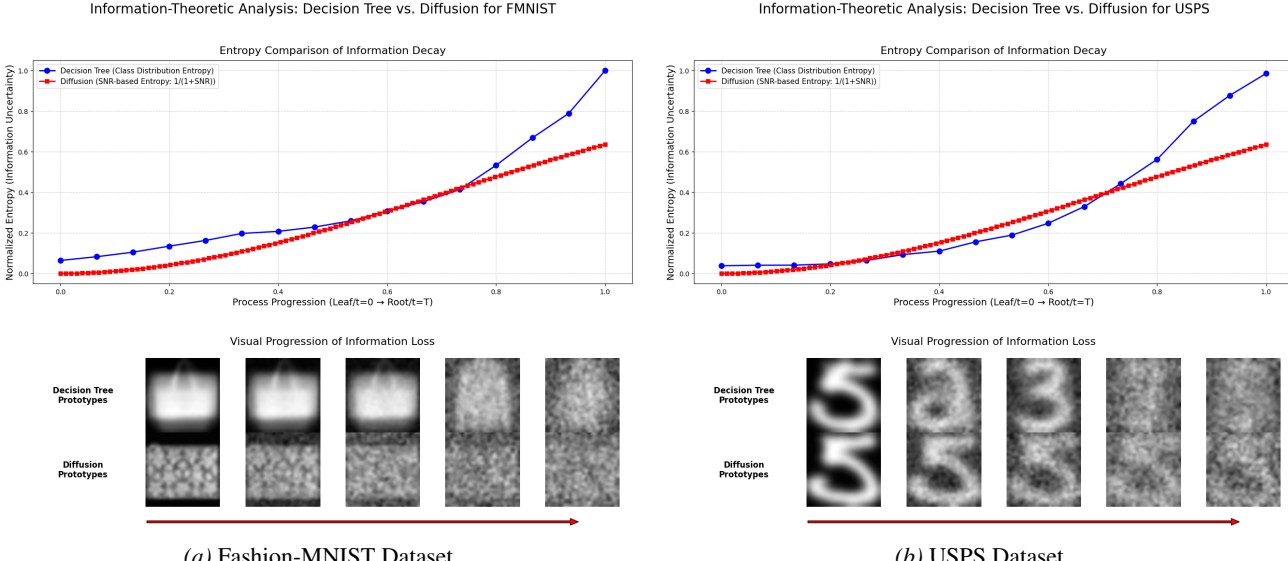

*(a)* Fashion-MNIST Dataset   *(b)* USPS Dataset

*Figure 13.* **Additional Results for Information-Theoretic Analysis (Experiment 2).** These figures supplement Figure 3 from the main paper, showing the analogous information decay between decision trees and diffusion on additional datasets. (a) On **Fashion-MNIST**, the entropy decay curves show strong qualitative agreement, and the visual prototypes illustrate the gradual loss of identifiable features (e.g., the shape of a bag). (b) On **USPS**, the information decay trajectories are nearly identical to those observed on MNIST, reinforcing the generality of the core equivalence.

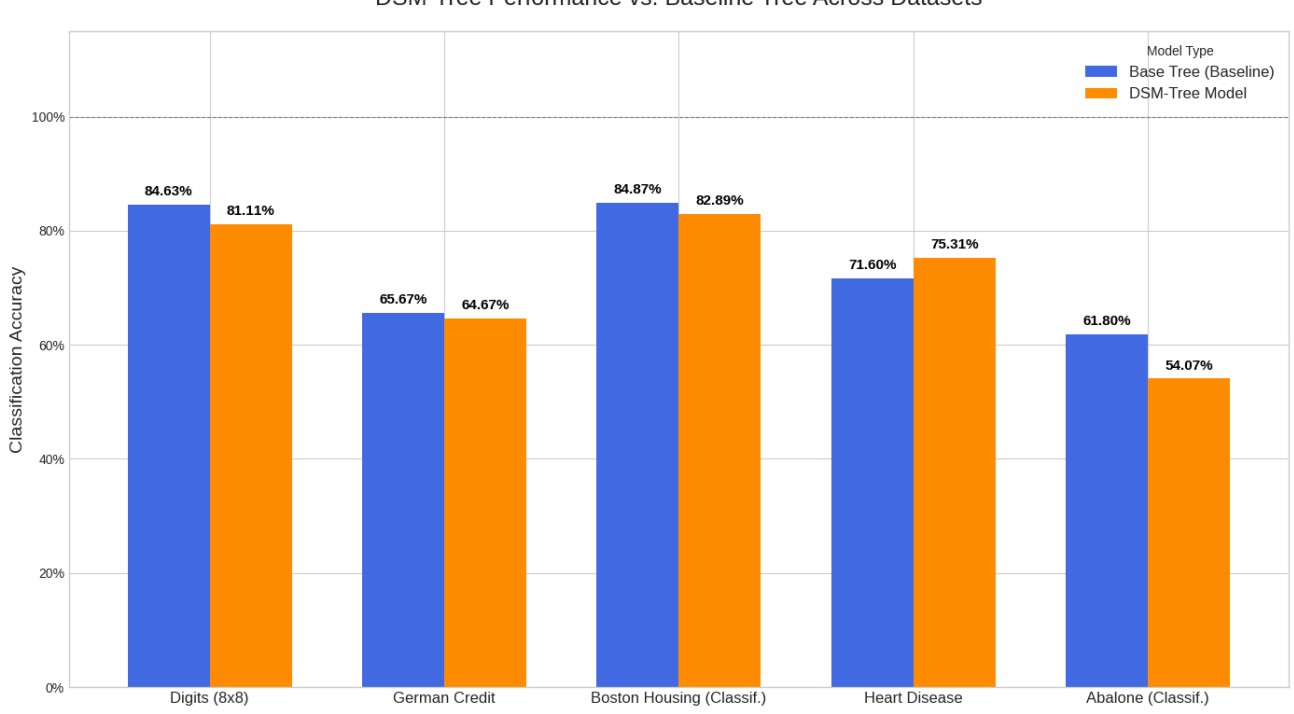

*Figure 14.* **Magnified View of DSM-TREE Performance.** Classification accuracy of the DSM-TREE model compared to its teacher (Base Tree). Performance is nearly identical on most datasets, with DSM-TREE outperforming on the Heart Disease dataset, which demonstrates successful knowledge transfer from the discrete tree to the continuous neural network. This full-page view is provided for enhanced clarity of the accuracy percentages and dataset labels.

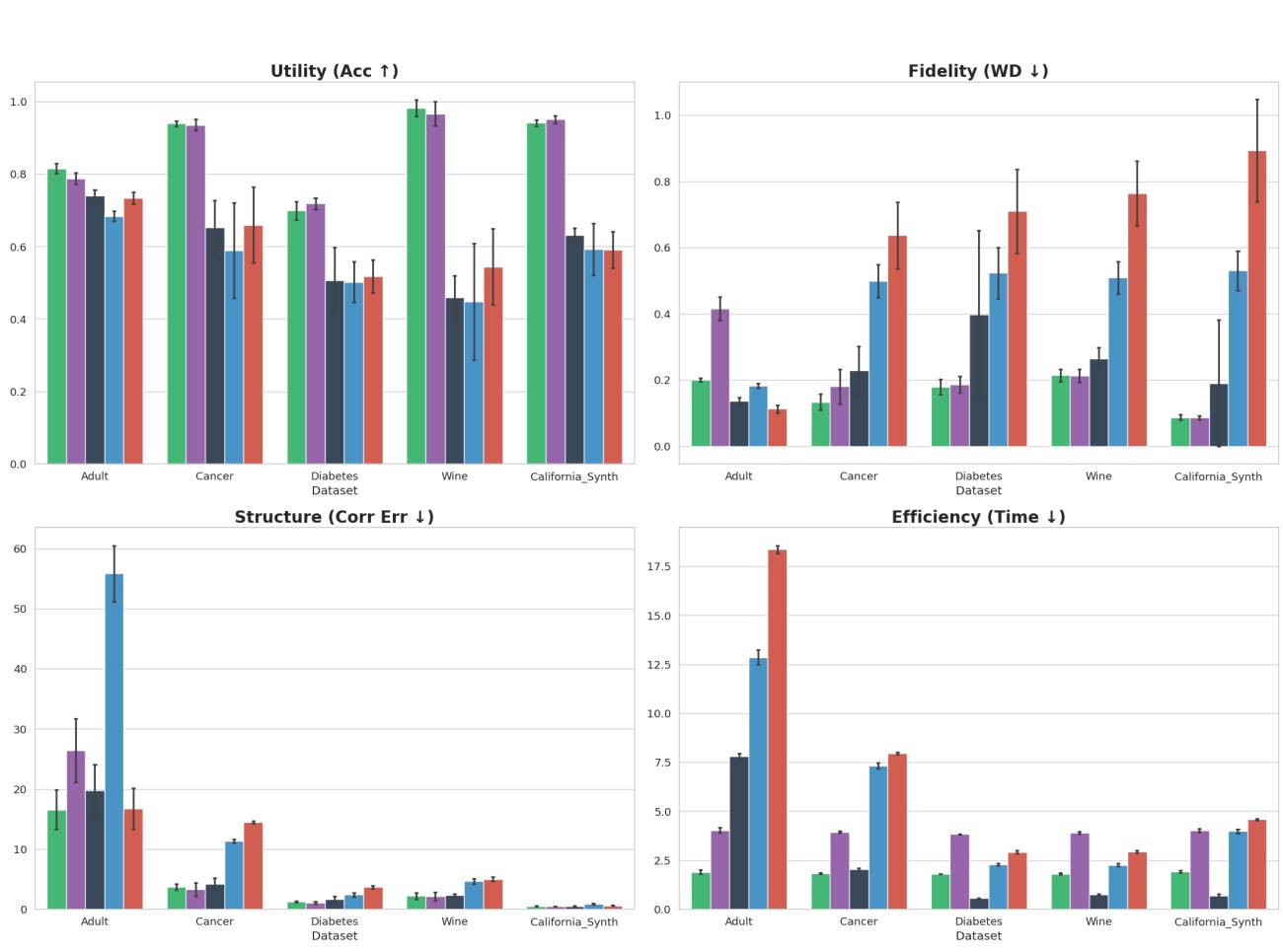

*Figure 15.* **Full-Page View of TREEFLOW Performance.** Comparative performance of TREEFLOW against baseline generative models across a suite of tabular benchmarks. We evaluate on four axes: Utility (TSTR Accuracy ↑), Fidelity (Wasserstein Distance ↓), Structure (Correlation Error ↓), and Efficiency (Runtime ↓). TREEFLOW consistently demonstrates state-of-the-art utility, often matching or exceeding the performance of more computationally intensive diffusion models like TabDDPM, while being significantly more efficient. This full-page view is provided for enhanced clarity of the results and labels.

