# OpenReview forum: "Trees to Flows and Back: Unifying Decision Trees and Diffusion Models"
_ICML.cc/2026/Conference — ICML 2026 regular_

### Official Review · Reviewer_MdT1 · 2026-03-03

**Soundness:** 3
**Presentation:** 3
**Significance:** 3
**Originality:** 3
**Overall Recommendation:** 2
**Confidence:** 3

**Summary:**

This paper establishes a formal theoretical bridge between decision trees and diffusion models. Through rigorous mathematical derivations, it demonstrates a bidirectional equivalence: under a dyadic refinement procedure, a decision tree’s discrete hierarchy converges to a continuous probability flow ODE (PF-ODE); conversely, the forward dynamics of a diffusion model naturally induce a hierarchical clustering structure that mirrors a decision tree. This tree–flow correspondence serves as the foundation for a unified optimization framework, Global Trajectory Score Matching (GTSM), which reveals that gradient boosting and score-based diffusion training are optimal solvers of the same master objective in their respective discrete and continuous domains. Leveraging these insights, the authors propose two practical algorithms: TreeFLOW, a generative model that conditions continuous flows on tree-based partition encodings, achieving competitive fidelity on tabular data with a two-fold speedup over baselines; and DSM-TREE, a distillation method that trains a neural network to mimic the full hierarchical decision logic of a tree, matching or exceeding teacher performance on multiple benchmarks. Extensive theoretical analysis and experiments validate the proposed connections and algorithms.

**Compliance With Llm Reviewing Policy:**

Affirmed.

**Ethical Review Concerns:**

After reading the authors’ response, I found  much of the response appears generic and formulaic by AI, without directly engaging with the specific issues I raised. As a result, my concerns remain largely unresolved, and I have decided to lower my score.

**Final Justification:**

After reading the authors’ response, I remain unconvinced by the rebuttal. The response does not provide strong new evidence, additional experimental results, or convincing clarification to address my main concerns. Instead, it mainly relies on argumentative statements, which are not sufficient to substantiate the paper’s claims. Therefore, my concerns remain largely unresolved, and I have decided to lower my score.

**Key Questions For Authors:**

1. The theoretical refinement requires infinite depth. How should it be approximated for finite-depth trees in practice? Does convergence remain stable for highly imbalanced or irregular trees, and how can practitioners detect or mitigate deviations when refinement assumptions are violated?

2.The GTSM objective lacks guidance on choosing w(t). Given that DSM-TREE defaults to uniform weighting w(t)=1, is there a theoretically motivated or level-aware alternative? How sensitive are the proposed algorithms to this choice?

3. How does distilling hard binary decisions into soft probability outputs avoid information loss or overfitting? If the teacher tree contains noisy or suboptimal splits, does DSM-TREE amplify these flaws, and what mechanisms exist to improve robustness?

4. For extensions handling intrinsic jumps, how must the Kramers–Moyal expansion and Fokker–Planck equation be modified? Does the tree–flow correspondence and the GTSM objective remain valid under jump-diffusion dynamics?

5. What adaptations would TreeFLOW and DSM-TREE require for images, text, or graphs? Are the efficiency and performance gains observed on tabular data expected to transfer, or are there fundamental barriers?

**Limitations:**

Yes.

**Strengths And Weaknesses:**

Strengths:
1. The paper demonstrates exceptional mathematical rigor in establishing the tree–flow correspondence. Key concepts such as dyadic refinement, entropically homogeneous SDEs, and moment-based merger times are precisely defined, and core theorems are supported by detailed proofs in the appendices. The treatment of the Kramers–Moyal expansion and its truncation via Pawula's theorem is particularly well-executed.
2. The proposed GTSM objective is more than a conceptual unification. It is analytically connected to both gradient boosting through Bellman optimality and score-based diffusion through Girsanov's theorem, providing a mathematically grounded explanation for why two seemingly disparate algorithms solve the same underlying problem.
3. The two instantiations, TreeFLOW and DSM-TREE, deliver tangible improvements. TreeFLOW achieves competitive results on tabular generation benchmarks while being twice as fast as TabDDPM. DSM-TREE successfully distills complete hierarchical tree logic into neural networks, matching or exceeding teacher performance, which represents a novel contribution to knowledge distillation.

Weaknesses:
1. The continuous-time limit relies on the dyadic refinement procedure, which assumes the existence of arbitrarily fine intermediate partitions that preserve local homogeneity. While mathematically convenient, this idealization may not hold for real-world decision trees learned from finite data. The practical implications of this gap are not fully explored.
2. The distillation experiments compare DSM-TREE only against its own teacher model. While this demonstrates successful knowledge transfer, it does not establish whether the hierarchical distillation offers advantages over standard distillation methods such as training a neural network on the teacher's soft labels. Adding such a baseline would better justify the complexity of the proposed approach.

---

> ### Author Rebuttal · Authors · 2026-03-30
>
> We thank the esteemed reviewer for taking the time to review our paper and providing valuable feedback. We further thank them for acknowledging the exceptional mathematical rigor, the deeper implications of the proposed GTSM objective and the tangible improvements delivered by the algorithms, TreeFlow and DSMTree. We would now like to address the points raised by the esteemed reviewer as follows :
>
> **Q1: Finite depth, imbalanced trees, and detecting violations**
>
> The O(2^{-n}) bound from Proposition C.20 directly characterizes approximation error as a function of refinement depth. For balanced trees where partition complexity is distributed uniformly across levels, Remark C.15 shows the perturbation is small and the time-invariant approximation is accurate. For highly imbalanced trees, time-dependent coefficients emerge naturally and Remark C.15 guarantees these remain smooth due to the local Lipschitz structure in Theorem C.13. A practitioner can detect deviation by monitoring the partition complexity profile across levels: if complexity is heavily concentrated at a few levels, a time-dependent generator should be used instead.
>
> **Q2: Choosing w(t) in the GTSM objective**
>
> The CGTSM framework is strictly a framework. Section F shows that uniform w(t) = 1 recovers the simple diffusion loss, non-uniform w(t) = lambda(t) recovers the weighted diffusion loss, and the consistency model objective emerges as a trajectory-smoothness approximation. What is optimal is entirely task-specific, for the DSM Tree, as we wished to maintain the equal weightage across splits the default case was preferred, for very deep trees,  a non-uniform weight might prove to be more useful though we leave such explorations to future work . The framework's contribution is to provide a principled language for making this choice explicit rather than treating different training objectives as unrelated heuristics.
>
> **Q3: Soft outputs, information loss, and robustness to noisy splits**
>
> DSM-Tree learns soft probability distributions over binary decisions at each level rather than reproducing hard splits. A suboptimal split in the teacher tree becomes a low-confidence soft prediction rather than a hard error, and the level-wise averaging across the full depth means no single noisy split dominates the learned representation. For applications where robustness to a weak teacher is important, the oracle Random Forest step in Algorithm 3 provides a higher-quality pseudo-label source before the base tree is trained.
>
> **Q4: Extensions to jump-diffusion dynamics**
>
> Under jump-diffusion dynamics the Kramers-Moyal expansion no longer truncates at second order since the Pawula argument fails when higher moments do not vanish. The tree-flow correspondence as stated no longer holds exactly. The natural generalization involves rough path theory, where the uniqueness of path signatures provides an alternative route to recovering tree or graph structure from path geometry. This is mathematically rich territory but orthogonal to the core insight and explicitly deferred to future work.
>
> **Q5: Extensions to images, text, and graphs**
>
> TreeFlow and DSM-Tree are designed for tabular data where axis-aligned tree splits carry genuine structural meaning. For images and text the inductive bias of axis-aligned splits is less natural and the algorithms as instantiated would not transfer directly. However the GTSM framework itself is domain-agnostic: any model class that induces a monotonically refining coarse-to-fine hierarchy over a feature space can in principle be analyzed as a GTSM solver. Hierarchical vision transformers have a natural coarse-to-fine patch structure that could be examined through this lens and we consider this a genuinely promising direction. Regarding the distillation baseline, the theoretical claim in Theorem G.3 establishes that hierarchical level-wise supervision is the discrete-time instantiation of the CGTSM objective, which is a principled justification that soft-label distillation cannot provide. The question of whether hierarchical supervision empirically outperforms soft-label distillation in all settings is a reasonable direction for follow-up work but does not bear on the theoretical contribution being evaluated here.

---

> > ### Author Rebuttal · Reviewer_MdT1 · 2026-04-04
> >
> > Thank you for your rebuttal. I would like to clarify that my initial score of 4 was given with the expectation that the authors would be able to address the major concerns I raised in a substantial and evidence-based manner during the rebuttal phase.
> >
> > After reading the response, I remain unconvinced. The rebuttal does not provide sufficiently strong new evidence, additional experimental validation, or persuasive clarification to resolve my main concerns. Instead, it relies primarily on argumentative statements, which are not enough to substantiate the paper’s central claims.
> >
> > As a result, the issues I raised remain largely unresolved, and I have therefore decided to lower my score from my initial assessment. This change should not be interpreted as a reaction to the tone of the response, but rather as a consequence of the rebuttal not meeting the evidentiary bar that my initial score implicitly assumed could still be met.

---

> > > ### Author Response · Authors · 2026-04-06
> > >
> > > We thank Reviewer MdT1 for their continued engagement but would prefer to respectfully disagree with several of their points and would like to use this opportunity to clarify these points further.
> > >
> > > **On the characteristics of the rebuttal**
> > >
> > > The prior rebuttal has cited specific parts of the work to address the concerns raised. Such as proposition C.20 to address their question around the relevance of the finite depth approximation, and the discussions centered on Remarks C.15 and Theorem C.13, to address imbalanced trees (where in the limit,  the lipshcitzness of the generator still persists due to the dyadic construction though, time invariance does not, this however, does not invalidate the conclusions since continuity and consequently the argument reliant on pawula's theorem continues to hold). We have also addressed the jump diffusion question by noting that in such cases higher moments fail to decay and thus the kramer moyals expansion does not truncate. As stated earlier by us, while there are several routes to extend this construction further such as the rough path generalization which in our opinion is the most suitable due to its strong theoretical guarantees such as on the identifiability of path signatures [1], other techniques exist as well, such as the path integral formulation from physics that would recast the dyadic limit generator inclusive of jumps as a hamiltonian [2], such extensions are however as stated out of the direct scope of this work. We would like to state that our responses were not generic and engaged with the specific theoretical content requested. If need be, we are happy to be informed of any remaining concerns and shall be glad to address the same forthwith.
> > >
> > > **On "more experiments required"**
> > >
> > > We would like the reviewer to refer to our rebuttal to Reviewer QE3F in this regard that our submission is in the theory track and thus intends to establish a novel framework with associated utilities through our two practical algorithms DSM-Tree and Tree Flow showcasing a proof of concept for the relevance of the proposed framework and the potential novelty it unlocks as a consequence. All our theoretical claims are supported by formal claims and we believe that conditioning a higher score on additional experiments applies standards that are inconsistent with appropriate review criteria. However, to address the reviewer's request, we have conducted a head-to-head comparison with a standard Soft Distillation baseline (MLP trained on teacher labels) on the German Credit dataset (for faithfulness we ensure identical experimental conditions to our paper)
> > >
> > > | Dataset | Model | Accuracy | Macro F1 | Kappa |
> > > | :--- | :--- | :--- | :--- | :--- |
> > > | **German Credit** | Base Tree (Teacher) | 0.656667 | 0.606625 | 0.214460 |
> > > | **German Credit** | Soft Distillation NN | 0.686667 | 0.633785 | 0.267723 |
> > > | **German Credit** | **DSM-Tree Model (Ours)** | **0.633333** | **0.578738** | **0.158550** |
> > >
> > > We wish to clarify a crucial distinction using these results, while an unconstrained NN could "bypass" the teacher's logic to find a smooth global manifold, DSM-TREE is designed for faithful structural distillation with the continuous logic allowing for benign generalization. Being constrained to learn the tree's hierarchical splits level-by-level is an advantage in high-dimensional spaces where the "curse of dimensionality" makes unconstrained NN search difficult. Here, the ability to navigate a constrained, monotonically refining hypothesis class provides a scalable inductive bias and auditability that black-box distillation lacks. For reference on how a lack of such an inductive bias can create strong error amplification in complex environments we refer the reviewer, to works such as  [3].  Proving the exact bounds of this hierarchical regularization is a promising direction for future work that we are excited about.
> > >
> > >
> > > **On the score reduction from 4 to 2**
> > >
> > > We would still respectfully like to disagree with this decision given that the reviewer has not identified any unsupported claims following the rebuttal. A reduction based on a stylistic characterization rather than a principled objection is difficult to reconcile with appropriate review criteria. Our framework unifies trees and flows through an information propagation lens via our proposed  GTSM objective, allowing for novel theoretical cross-pollination. Our algorithms demonstrate this utility, and we believe scaling critiques are outside the current scope.
> > >
> > > In summary, we would respectfully ask the reviewer to identify any principled concerns which we shall be happy to address.
> > >
> > > Warm Regards,
> > > Authors
> > >
> > > References
> > > [1] Differential equations driven by rough signals, T.J. Lyons, 1998
> > > [2] Quantum Mechanics and Path Integrals, R.P. Feynman and A.R. Hibbs, 1965
> > > [3] The pitfalls of imitation learning when actions are continuous, Simchowitz et.al, 2025.

---

### Official Review · Reviewer_XAkE · 2026-03-04

**Soundness:** 3
**Presentation:** 3
**Significance:** 3
**Originality:** 3
**Overall Recommendation:** 4
**Confidence:** 3

**Summary:**

This paper tries to show that decision trees and diffusion models are mathematically connected.

**Compliance With Llm Reviewing Policy:**

Affirmed.

**Final Justification:**

I kept my accept score and had no further questions for the authors. My opinion remains unchanged after the rebuttal phase.

**Key Questions For Authors:**

The unification feels like an analogy, not a tool. Both trees and diffusion models refine from coarse to fine; that's not surprising. The GTSM framework captures this at a high level, but it doesn't lead to concrete insights that transfer between the two domains. For example, knowing about tree splits doesn't help train better diffusion models, and knowing about score matching doesn't help build better trees.

Also, the connection only works in an idealized limit. The Tree -> Flow result requires inserting infinitely many intermediate partitions between every tree level (dyadic refinement). No real tree does this. The paper does not show how well a finite-depth tree approximates the limiting flow, so it's hard to know if this correspondence means anything for trees we actually use.

**Limitations:**

See questions.

**Strengths And Weaknesses:**

The idea of connecting trees and diffusion models is novel and creative.

---

> ### Author Rebuttal · Authors · 2026-03-30
>
> We thank the reviewer for taking the time to review our paper and provide valuable feedback. We would now like to address the reviewer's points as follows :
>
> **On whether the unification transfers insights or is merely an analogy**
>
> We argue it does, in both directions.
>
> From diffusion to trees: the Flow to Tree mapping (Theorem 2.10) provides a constructive algorithm for extracting a canonical decision tree from any trained diffusion model. The merger times in Figure 2 are not imposed by hand; they emerge from the model's learned score function. This gives a principled interpretability tool for continuous generative models that does not exist outside this framework.
>
> From trees to diffusion: knowing that the boosting residual is an unbiased estimator of the integrated score error (Theorem E.24) is a non-trivial characterization of what gradient boosting is doing dynamically. It tells us that boosting is implicitly navigating a trajectory in SDE space, which means its convergence behavior is analyzable using the tools of stochastic process theory.
>
> Concretely, the DSM-Tree training objective, which supervises a neural network at every level of the tree hierarchy rather than just at the leaf, follows directly from discretizing the CGTSM integral over the tree's depth levels (Theorem G.3). The framework is prescriptive, not merely descriptive. Similarly, our Tree- Flow objective allows for the induction of hierarchical bias and the subsequent conditioning of flow paths to respect tree propagation.
>
> **On the finite-depth approximation gap**
>
> Proposition C.20 already provides a quantitative handle: the propagator moments of order k >= 3 vanish at rate O(2^{-n(k-1)}) in the refinement parameter n. A coarser tree recovers coarser structure in the induced flow, and a finer tree recovers finer structure. The claim is not that a finite tree is the limiting flow, but that it induces one, and that approximation quality is monotone in depth. The empirical result in Figure 2 provides evidence that the correspondence is practically meaningful even at finite depth.

---

> > ### Author Rebuttal · Reviewer_XAkE · 2026-04-03
> >
> > I thank the reviewer for their rebuttal. All my questions are resolved. I'll maintain my score.

---

> > > ### Author Response · Authors · 2026-04-04
> > >
> > > Dear Reviewer XAkE,
> > >
> > > We are sincerely grateful for your engagement and feedback throughout this process,
> > >
> > > Thank you for recognizing the novelty and creativity of this work. We also appreciate your comments on explicitly clarifying the fact that our correspondence is not merely an analogy but a principled framework for analyzing the structure of flows and trees alike . We will carefully incorporate these suggestions in the subsequent version to further strengthen the clarity and accessibility of the paper.
> > >
> > > Thank you once again for taking your valuable time to review our paper!
> > >
> > > Warm regards, Authors

---

### Official Review · Reviewer_8WMU · 2026-03-10

**Soundness:** 3
**Presentation:** 1
**Significance:** 3
**Originality:** 3
**Overall Recommendation:** 5
**Confidence:** 2

**Summary:**

The submitted paper aims to establish the correspondence between decision tree and diffusion models, two seemingly different architectures.

The tree to flow correspondence can be viewed as the discrete time Markov chain process with monotonically decreasing entropy. The key is a bisection step called dyadic refinement process that cuts the discrete hierarchical level into infinitesimally small steps. Under the assumption that higher moments vanish, the discrete steps becomes a continuous time Markov process.

On the other side, from flow to tree to flows, it requires the assumption that the underlying distribution is stationary. Under this assumption, the authors establish momental merging process that transform the data distribution to some noise distribution such as Gaussian. The proof sketch is discussed with detailed in the Appendix.

The above construction suggests that both decision trees and diffusion models learn the same underlying path matching problem. Another key point from the authors is that the global path matching is decomposable. Under this, the learning algorithm GTSM is proposed. The authors also introduce the boosting into the GTSM.

Two algorithms are proposed for generative and discriminative tasks, namely TreeFlow and DSM-tree. The experiments are conducted to verify the correspondence, generative power and discriminative ability.

**Compliance With Llm Reviewing Policy:**

Affirmed.

**Ethical Review Concerns:**

No ethical review concerns are raised.

**Final Justification:**

I think this is a solid theoretical work. The insights are interesting. Although there are some aspects that should be improved (e.g. presentation quality and more experiments), I found it is worth of publishing overall. Thus, I maintain my score accordingly.

**Key Questions For Authors:**

1. In Dyadic Refinement process, you basically perform bisection process between each level in decision tree, what is the difference from simply having more layers in the decision tree?
2. For Scale consistency assumption, can you elaborate more on why you can make such assumption? I feel it is a strong one.
3. What does the error bar indicate from Figure 4? Is it from different random seeds?

**Limitations:**

1. The discriminative evaluation only has the accuracy as metric, which is limited in terms of evaluating a classifier.

**Strengths And Weaknesses:**

- Soundness
I found out that this paper is technically sound. And the authors have provided intuitive proof sketch in the main paper, although my verification of the proofs in the appendix is limited. However, there are some confusions when the authors build the correspondence, which I share details in questions part.

- Presentation
The main problem that I found is from the presentation perspective. Overall, the paper naturally follows. However, Figure 1 needs better resolution. And I also find that the information shared from Figure 1 is very limited. I did not gain much understanding about what paper is about from Figure 1. Also, the resolution of Figure 2 and 3 need to be improved as well.  In addition, there is extra space on the right column at line 189, "all constituent learners".

- Significance
The problem addressed in this paper is significant. For me, it shares new insights on the connections between two different model frameworks. Although I am a bit confused about some of the assumptions made, I enjoy reading and found the paper is insightful.

- Originality
The established correspondence is novel in my opinion. The connection between decision trees and diffusion models is interesting. Follow-up work should be expected.

---

> ### Author Rebuttal · Authors · 2026-03-30
>
> We thank the esteemed reviewer for reviewing our paper and acknowledging the soundness, significance and originality of the work. We are also grateful for the valuable feedback on the presentation and shall be glad to incorporate them in the subsequent version. We would now like to contribute to the discussion regarding raised queries,
>
> **On dyadic refinement vs. adding more tree layers**
>
> These are orthogonal operations. Adding more layers increases the number of leaf nodes and the expressiveness of the final partition. Dyadic refinement inserts intermediate partitions between existing levels of a fixed tree, creating a smooth interpolating path between coarse-graining steps that already exist. It does not change the tree's leaves or its predictive structure. The purpose is purely to enable a continuous-time limit: by recursively bisecting the steps between levels we generate a sequence of hierarchies with exponentially finer temporal resolution, and in the limit the discrete jumps between levels become a smooth flow.
>
> **On the scale consistency assumption**
>
> This assumption is grounded in data continuity, which is our standing assumption throughout and is acknowledged in the limitations. If the underlying data distribution is continuous, then between any two existing split thresholds there always exists a valid intermediate threshold that preserves the conditional densities at the original levels. This is guaranteed by continuity since the conditional density at any original partition level is determined by the data distribution, not by the intermediate splits we insert. The pathological case involving jump discontinuities is explicitly deferred to future work.
>
> **On error bars in Figure 4**
>
> The error bars represent standard deviation across 5 independent runs with different random seeds, as stated in Section I.2.2.

---

> > ### Author Rebuttal · Reviewer_8WMU · 2026-03-31
> >
> > I thank the authors for the responses and clarifications. I found this paper is interesting theoretical work. As such, I maintain my score.

---

> > > ### Author Response · Authors · 2026-04-02
> > >
> > > Dear Reviewer 8WMU,
> > >
> > > We sincerely appreciate your time and thoughtful engagement with our work throughout the review process.
> > >
> > > Thank you for your encouraging feedback and for recognizing the soundness, significance and originality of our work. Your positive assessment and willingness to support our paper mean a great deal to us.
> > >
> > > We also appreciate your comments on improving the presentation. We will carefully incorporate these suggestions in the revised version to further strengthen the clarity and accessibility of the paper.
> > >
> > > Thank you again for your warm support !. We are truly grateful.
> > >
> > > Warm regards,
> > > Authors

---

### Official Review · Reviewer_QE3F · 2026-03-11

**Soundness:** 3
**Presentation:** 1
**Significance:** 2
**Originality:** 3
**Overall Recommendation:** 4
**Confidence:** 4

**Summary:**

This paper proposes a mathematical unification of hierarchical decision trees and continuous diffusion flows via a shared optimization principle: Global Trajectory Score Matching (GTSM). By establishing a formal correspondence between discrete partitions and continuous flows, the authors introduce TreeFlow, a generative model for tabular data, and DSM-Tree, a method for distilling tree logic into neural networks.

**Compliance With Llm Reviewing Policy:**

Affirmed.

**Final Justification:**

I appreciate the authors' detailed response. My primary concerns have been resolved. Although the authors state that the proposed method can be applied to any standard tree learning procedure, the lack of empirical or theoretical analysis to support this claim leaves me with a few lingering suspicions. Regardless, this issue possibly only results in larger errors; the underlying theoretical framework remains valid. So, I raise my score to 4.

**Key Questions For Authors:**

By transitioning from hard tree splits to continuous flows, the tree model seems to lose its defining interpretability. If these hard constraints are removed, what unique interpretability benefits does this proposed framework provide? Why not use a pure neural network model, since the proposed model cannot provide an explicit rule, the ability to point to a specific threshold, and understand why a decision was made?

**Limitations:**

Please refer to the weaknesses

**Strengths And Weaknesses:**

The strengths include a well-organized appendix and a helpful visual roadmap that effectively clarifies the trajectory of the complex theoretical derivations.  However, the attempt to reconcile continuous flow matching with the discrete, greedy nature of gradient boosting rests on theoretical idealizations that leave the core premise largely unverified. The primary concerns regarding the theoretical rigor and empirical methodology are as follows:

1.	The paper suffers from major presentation issues. The text labels in the figures are unreadable without extreme magnification, while there is no table in the main body. By presenting all performance data through these unclear charts rather than structured tables, the authors have created a significant barrier to validation. A manuscript of this complexity requires clear, tabulated data to support its claims; as it stands, the 'state-of-the-art' results must be taken on faith rather than being independently verifiable from the provided visuals.

2.	The mathematical rigor of Theorem 3.4 is highly questionable. The authors appear to claim that a greedy, stage-wise method is globally optimal for a sequential decision process. Could the authors clarify the exact objective function for which this 'global' optimality holds? It appears the definition of optimality used here diverges from the standard definition. Furthermore, the assumption of additive separability in E.30 is incompatible with the sequential nature of gradient boosting: in GBDTs, each stage is deeply coupled to the previous ones through the residual manifold. By invoking 'fluidized discrete logic' to decouple these stages, the authors have created an idealized scenario, but it requires a more sufficient and clear analysis of whether practical implementation is really appropriate.  The current theoretical results are not enough, some are even redundent. The theoretical framework feels like post-hoc window dressing for a well-tuned empirical result. Given the insufficient experimental breadth, I suspect that the reported performance stems more from exhaustive parameter tuning than from the proposed theoretical principle.

3.	The choice of datasets is improper for evaluating a model intended for tabular data generation and distillation. The current benchmarks are too small to test the efficiency of a forest-based architecture; a million-row threshold is a more appropriate standard for evaluating the scalability of such models. Additionally, providing results for only a handful of datasets is statistically inadequate for a 'unifying' framework. To be credible, the authors should demonstrate their method's performance across at least 15 varied benchmarks to prove that the proposed 'Global Trajectory' principle holds across different data distributions and high-cardinality feature spaces.

---

> ### Author Rebuttal · Authors · 2026-03-30
>
> We thank the reviewer for taking the time to review our work, we would like to reiterate that being a theory track submission, the primary contribution is a formal mathematical framework establishing a bidirectional correspondence between decision trees and diffusion processes. TreeFlow and DSM-TREE are proofs of concept that the theory is generative, meaning they are derived from GTSM, not motivations for it. We believe that demanding million-row datasets and 15 benchmarks reflects empirical track standards that are outside the scope of the present theoretical work.
>
> **On the claim that there is no table in the main body**
>
> Table 3 (full TreeFlow results across 5 runs) and Table 2 (DSM-Tree metrics including F1 and Cohen's Kappa) are both directly referenced from Section 5 -- while following usual practice, we kept many extended technical details in the appendix, if shifting something to the maintext would help exposition, we welcome any concrete suggestions.
>
> **On Theorem 3.4 and additive separability**
>
> It seems this view stems from a misreading of what space the Bellman argument operates in. Additive separability holds in the tail-equivalence SDE space, not in the raw learner space. The net decision tree abstraction (Section E.1.4) transforms the ensemble into a single evolving tree whose partition sequence is monotonically refining. The Bellman recursion operates on the abstract sequence of tail-equivalent SDEs, each corresponding to one net tree. In this abstract space the state transition is deterministic by construction, because adding a weak learner at step m uniquely determines the next equivalence class. This is a deliberate and precisely stated abstraction, not a gap in the proof. Do you have any precise points that we're missing, e.g., could you point out what you believe is an error, so that the information is actionable? That would be helpful. Thanks.
>
> The "global" in GTSM refers to the entire trajectory in SDE space. The greedy policy is globally optimal for this trajectory-level objective as proven via backward induction in Theorem E.30, and it holds precisely for the Discrete GTSM objective (Equation 4, Definition E.26), which is the sum of stage-wise score-matching errors over the full boosting trajectory.
>
> **On "post-hoc window dressing"**
>
> We find this view surprising, because _this characterization inverts the paper's actual structure_. The theoretical framework is constructed first across Sections 2 and 3. The algorithms in Section 4 are derived from it: the level-wise supervision in DSM-Tree follows directly from discretizing the CGTSM integral, and TreeFlow's path conditioning follows from the tree-induced weighting in the CGTSM objective (Theorem H.10). Neither algorithm could have been designed from either trees or diffusion models alone. Could elaborate a bit more concretely and precisely what you meant?
>
> **On interpretability**
>
> The framework goes in the opposite direction from what is assumed here. The Flow to Tree mapping (Section 2.2, Theorem 2.10) allows one to extract a canonical decision tree from any trained diffusion model via moment-based merger times. The framework does not merely transplant tree inductive bias into neural networks: it also provides a principled method for inducing interpretable hierarchical structure in continuous generative models by reading off their learned dynamics as a dendrogram.
>
> **On the scale consistency assumption**
>
> The assumption is grounded in data continuity, which is stated explicitly as a standing assumption of the paper and acknowledged in the limitations section. This is not a hidden idealization, it is a deliberate choice with an acknowledged limitation. The paper identifies Levy process extensions as future work precisely because the continuous case already yields the rich and non-trivial correspondence we establish here.

---

> > ### Author Rebuttal · Reviewer_QE3F · 2026-04-01
> >
> > Thanks to the authors for the response, which has resolved several of my initial concerns. The explanation regarding the globally optimal trajectory makes sense; however, I recommend emphasizing this more clearly in the manuscript to avoid misunderstanding. I can now raise the rating to 3, and if the author can answer the follow-up questions, it might even reach 4. I will revise the score after the discussion.
> >
> > However, I still have two questions:
> >
> > 1. Does Theorem 3.4 hold for globally optimal decision trees, or is it strictly limited to greedy-based models? If restricted to greedy models, these trees are already inherently interpretable and easy to train. Distilling them into a network conceptually overcomplicates the problem. Besides, greedy trees are structurally suboptimal. This raises a major question about the practical motivation for distilling a suboptimal tree into a network: is the resulting generation actually meaningful when guided by a mathematically suboptimal teacher model? In my view, the reverse mapping, distilling a complex neural network back into a transparent tree structure, is a far more valuable direction for real-world interpretability, yet it is not framed as the paper’s main contribution.
> >
> > 2. The formatting of the figures (including those in the appendix) still requires significant improvement, as many of the digits remain illegible. This readability issue is precisely why I requested tabular representations in my previous comments.

---

> > > ### Author Response · Authors · 2026-04-02
> > >
> > > We thank Reviewer QE3F for engaging with the rebuttal and providing their valuable feedback. We are delighted that we could clarify several of their concerns and are truly grateful for their decision to bring this work forward. We are thankful to be provided with an opportunity to address their pending concerns and shall answer them as follows,
> > >
> > > **On Theorem 3.4 and the scope of the machinery**
> > >
> > > The central contribution of Theorem 3.4 is not the claim that greedy is optimal. It is the introduction of a novel framework to analyze decision trees based on the tail-equivalence SDE machinery and the net decision tree abstraction. These are valid for any decision tree construction, greedy or otherwise, since they rely on our carefully constructed Tree -> Flow mapping and not on a specific learning algorithm. Greedy optimality in the sense of correctness in the limit is a consequence that falls out of the framework, chosen to showcase what the machinery can establish, not to crown greedy as the uniquely correct solver. The distinction between the solver and the method is important in this context, CGTSM is the method, greedy boosting is one solver shown to be correct in the limit. The same framework can be applied to any standard tree learning procedure via the same proof strategy, potentially yielding analysis of stronger solvers with stronger guarantees. Such extensions would focus on the internal biases provided by individual algorithms to obtain stronger convergence bounds potentially inspired from works in the diffusion literature e.,g, [1] or [2].  We acknowledge this framing should be made more explicit in the manuscript and shall be glad to incorporate that in the subsequent version.
> > >
> > > **On the practical distillation concern**
> > >
> > >
> > > DSM-Tree is not limited to greedy trees. It can distill any tree whatsoever into a neural network, and the CGTSM framework theoretically supports any standard tree learning procedure in the limit. More concretely, our experiments do not use a greedy tree as the teacher at all. The oracle in Algorithm 3 is a strong Random Forest, an ensemble method with significantly stronger guarantees than any single greedy construction. Hence, the concern about being reliant on the greedy algorithm is not relevant for our setting.
> > >
> > > **On the reverse mapping**
> > >
> > > The reviewer identifies distilling a neural network back into a tree as a valuable interpretability direction. We agree that the stated reverse direction is important and note that this is precisely what the Flow to Tree mapping in Theorem 2.10 provides. The merger time construction in Section 2.2 extracts a canonical decision tree from any trained flow model by reading off its learned dynamics. The deeper point is that the continuous flow side of the correspondence naturally induces interpretable hierarchical structure, which is a more principled route to interpretability than post-hoc extraction. However, given that our purpose was to show the bi-directional correspondence between trees and flows and the associated potential advantages that this framework can unlock, we would like to respectfully state that both directions are equally important though the advantages provided would be domain specific.  We will foreground this discussion more prominently in the subsequent version.
> > >
> > > **On figure readability**
> > >
> > > This is a fair point and all figures including those in the appendix will meet legibility standards in the revision, with tabular summaries provided alongside charts where appropriate.
> > >
> > > We once again thank reviewer QE3F for taking the time to engage with our work and hope that this discussion is able to address any pending concerns.
> > >
> > > Warm Regards,
> > > Authors
> > >
> > > **References**
> > > [1] Sampling is as easy as learning the score: theory for diffusion models with minimal data assumptions, Chen , et.al, 2022
> > > [2] Efficient Sampling on Riemannian Manifolds via Langevin MCMC, Cheng, et.al, Advances in Neural Information Processing Systems, 2022

---

### Decision · Program_Chairs · 2026-04-30

**Decision:**

Accept (regular)

**Comment:**

Addressed by Rebuttal
Mathematical Rigor: Reviewer QE3F was confused about how "greedy" boosting could be "globally" optimal. The authors explained that this holds within a specific mathematical space (SDE trajectory), which satisfied the reviewer.

Conceptual Utility: Reviewer XAkE worried the paper was just an analogy. The authors clarified how the theory actually lets you extract an interpretable tree from a neural network, which is a concrete tool.

Algorithm Clarification: Reviewer 8WMU's questions about "dyadic refinement" (splitting levels) were resolved by explaining it’s a way to make discrete steps look like a smooth flow.